# Bioinformatic Analysis of the CXCR2 Ligands in Cancer Processes

**DOI:** 10.3390/ijms241713287

**Published:** 2023-08-27

**Authors:** Jan Korbecki, Mateusz Bosiacki, Dariusz Chlubek, Irena Baranowska-Bosiacka

**Affiliations:** 1Department of Biochemistry and Medical Chemistry, Pomeranian Medical University in Szczecin, Powstańców Wlkp. 72, 70-111 Szczecin, Poland; jan.korbecki@onet.eu (J.K.); mateusz.bosiacki@pum.edu.pl (M.B.); dchlubek@pum.edu.pl (D.C.); 2Department of Anatomy and Histology, Collegium Medicum, University of Zielona Góra, Zyty 28 St., 65-046 Zielona Góra, Poland; 3Department of Functional Diagnostics and Physical Medicine, Faculty of Health Sciences, Pomeranian Medical University in Szczecin, Żołnierska Str. 54, 71-210 Szczecin, Poland

**Keywords:** chemokine, CXCR2, bioinformatics, pan-cancer analysis, NF-κB, IL-8, Gro-α

## Abstract

Human CXCR2 has seven ligands, i.e., CXCL1, CXCL2, CXCL3, CXCL5, CXCL6, CXCL7, and CXCL8/IL-8—chemokines with nearly identical properties. However, no available study has compared the contribution of all CXCR2 ligands to cancer progression. That is why, in this study, we conducted a bioinformatic analysis using the GEPIA, UALCAN, and TIMER2.0 databases to investigate the role of CXCR2 ligands in 31 different types of cancer, including glioblastoma, melanoma, and colon, esophageal, gastric, kidney, liver, lung, ovarian, pancreatic, and prostate cancer. We focused on the differences in the regulation of expression (using the Tfsitescan and miRDB databases) and analyzed mutation types in CXCR2 ligand genes in cancers (using the cBioPortal). The data showed that the effect of CXCR2 ligands on prognosis depends on the type of cancer. CXCR2 ligands were associated with EMT, angiogenesis, recruiting neutrophils to the tumor microenvironment, and the count of M1 macrophages. The regulation of the expression of each CXCR2 ligand was different and, thus, each analyzed chemokine may have a different function in cancer processes. Our findings suggest that each type of cancer has a unique pattern of CXCR2 ligand involvement in cancer progression, with each ligand having a unique regulation of expression.

## 1. Introduction

Chemokines are chemotactic cytokines that play an important role in the functioning of the immune system [1]. Due to the significant involvement of various immune cells in the processes of tumorigenesis, these cytokines are also a crucial component of the tumor microenvironment [2]. In humans, 43 different chemokines are distinguished and divided into four sub-families [1]. Additionally, nineteen chemokine receptors [1] and four atypical chemokine receptors [3] have been identified.

One of the chemokine receptors is the CXC motif chemokine receptor 2 (CXCR2). The expression of this receptor is found only on neutrophils, basophils, and memory B cells [1]. As a result, CXCR2 ligands act as chemoattractants for neutrophils [1,4] and basophils [1,5]. In the bone marrow, CXCR2 expression is present on hematopoietic stem cells [6] and myeloid progenitors [7,8], indicating that CXCR2 ligands may influence these cell types. The highest levels of CXCR2 expression are observed in the appendix, spleen, and esophagus. CXCR2 expression is also found in the gall bladder, urinary bladder, lung, bone marrow, and skin [9].

CXCR2 serves as a receptor for seven chemokines: CXC motif chemokine ligands (CXCL)1, CXCL2, CXCL3, CXCL5, CXCL6, CXCL7, and CXCL8/IL-8 [1]. It is worth noting that CXCL7 is a protein encoded by the pro-platelet basic protein (PPBP) gene. PPBP is a polypeptide from which the N-terminus is removed following translation, leading to the formation of connective tissue-activating peptide III (CTAP-III), β-thromboglobulin (β-TG), or the shortest CXCL7/NAP-2, depending on the extent of N-terminal cleavage [10].

The most significant difference among these chemokines Is that CXCL6 and CXCL8/IL-8 also activate CXC motif chemokine receptor 1 (CXCR1) at low concentrations [11,12]. In contrast, the remaining CXCR2 ligands also activate CXCR1 at approximately 100 times higher concentration than CXCR2 [11,13]. This is due to the structural differences in the N-terminal regions of both receptors [14]. The amino acid sequences of these receptors exhibit a similarity of 77% [15]. The most prominent distinctions between the two receptors lie in their N-terminal and C-terminal regions. These receptors, along with a pseudogene, form a gene cluster on 2q34-q35 [16], indicating their origination through gene duplication.

While CXCR1 and CXCR2 activate similar signaling pathways [17,18,19,20], their signal transduction processes differ. The mobilization of Ca^2+^ by CXCR2 solely relies on Gα_i_, whereas CXCR1’s signal transduction is only partially dependent on this G protein [19]. Furthermore, focal adhesion kinase (FAK) activation by CXCR2 is adhesion-dependent, unlike CXCR1 signaling [21]. CXCR1, but not CXCR2, induces oxidative burst in neutrophils [19]. Additionally, CXCR1 can activate protein kinase Cε (PKCε), unlike CXCR2 [22]. PKCε subsequently induces the desensitization of CXCR2, suggesting that CXCR1 activation by CXCL8 influences the function of CXCR2 ligands.

All or most CXCR2 ligands are expressed in each organ of the body, although some of them do exhibit organ-specific expression (Table 1) [9]. In the appendix and gall bladder, all CXCR2 ligands show high expression levels, except for *PPBP*. In the spleen, only *CXCL1*, *CXCL6*, and *PPBP* are expressed. The urinary bladder displays high expression of all CXCR2 ligand genes, excluding *CXCL2* and *PPBP*. In the liver, only *CXCL2*, *CXCL3*, and *CXCL8* are expressed. The bone marrow exhibits expression of all CXCR2 ligand genes, except for *CXCL5* and *CXCL6*. *CXCL5* demonstrates tissue-specific expression in the lymph node, while *CXCL3* and *CXCL5* exhibit specific expression in the stomach. *CXCL8* shows specific expression in the esophagus, *CXCL3* in the colon, and *CXCL1* in the small intestine. The lung expresses *CXCL2*, *CXCL3*, and *CXCL5* [9]. Notably, *PPBP* expression is observed exclusively in the bone marrow and spleen.

The expression of CXCR2 ligands is regulated through various mechanisms. One such mechanism involves translational regulation mediated by different transcription factors, particularly nuclear factor κB (NF-κB) [23,24,25]. Another mode of regulation involves changes in mRNA stability facilitated by the binding of specific proteins to mRNA [26,27,28,29]. Additionally, miRNAs also play a significant role in the regulation of CXCR2 ligand expression [30,31,32].

It is important to note that the aforementioned mechanisms do not encompass the entirety of regulatory mechanisms governing the expression and activity of CXCR2 ligands. Proteolytic processing of CXCR2 ligands can also occur, potentially impacting their activation of the CXCR2 receptor [33]. Within the extracellular matrix (ECM), CXCR2 ligands are bound and may be released following ECM degradation [34,35]. Furthermore, CXCR2 ligands can exist as dimers [36,37]. Notably, different heterodimers of CXCR2 ligands appear to activate the CXCR2 receptor in similar ways.

Various amino acid residues of CXCR2 are required for the binding of specific ligands, leading to CXCR2 activation and signal transduction [38]. This indicates that evolutionary changes in the amino acid sequences of CXCR2 ligands have accumulated in conjunction with changes in the CXCR2 receptor.

CXCR2 belongs to the seven transmembrane G protein-coupled receptor (GPCR) family. Consequently, signal transduction occurs through trimeric G proteins, particularly Gα_i_ [17,18,19,39]. This results in the inhibition of adenylyl cyclase, calcium mobilization, and activation of the phosphatidylinositol-4,5-bisphosphate 3-kinase (PI3K)-protein kinase B (PKB) pathway [39]. Additionally, CXCR2 can activate other signaling pathways independent of G proteins. Various cytoplasmic proteins bind to CXCR2 upon receptor activation [40,41], and these proteins play a crucial role in the activation of different signaling cascades. Specifically, they activate PKB, Rac1, and Cdc42 [41,42,43], resulting in actin polymerization and cell migration induced by CXCR2 ligands.

Neutrophils are particularly responsive to CXCR2 ligands [1,4] and during inflammatory responses, the increased expression of CXCR2 ligands leads to tissue infiltration by neutrophils. This mechanism is notably observed in response to bacterial pathogens [44,45], where neutrophils combat bacterial infections. This mechanism is also implicated in various diseases such as rheumatoid arthritis [46], multiple sclerosis [47], periodontal diseases [48], type I diabetes mellitus [49], inflammatory bowel disease [50], and others. Neutrophil recruitment, a significant pathological feature of these diseases, results in tissue damage.

Similarly, in tumors, inflammatory reactions contribute to the formation of the tumor microenvironment [51]. Inflammatory responses are associated with the increased expression of CXCR2 ligands and subsequent activation of the CXCR2 receptor within the tumor. The pathways activated by CXCR2 increase the proliferation of cancer cells and exhibit anti-apoptotic properties [52]. CXCR2 ligands also have pro-angiogenic properties [53], which is related to the expression of CXCR2 on endothelial cells [54]. In addition, CXCR2 ligands cause chemotaxis of cells possessing CXCR2 and, thus, recruit various cells to the tumor niche. In particular, CXCR2 ligands cause the recruitment of neutrophils [55,56] and granulocytic-myeloid-derived suppressor cells (G-MDSC) [57,58] to the tumor niche.

Mammalian chemokine systems have multiple CXCR2 ligands, for instance, seven in humans and six in mice [59]. Despite the fact that all CXCR2 ligands have similar molecular properties and activate CXCR2, changes in their expression are not uniform and depend on the physiological state and the specific molecular process [60,61]. For example, different ligands of CXCR2 play a role in various stages of neutrophil tissue infiltration. Another specific ligand of CXCR2 is responsible for facilitating neutrophil adhesion to vessel walls and their subsequent transmigration [61].

However, the precise significance and functional differences among these individual CXCR2 ligands remain poorly understood. Furthermore, due to the divergence between the CXCR2 ligand systems in mice and humans, our knowledge regarding the specific actions of CXCR2 ligands in humans is still limited [59]. Significantly, there has been no attempt to compare all CXCR2 ligands in terms of different properties in all types of tumors. Also, no one has ever attempted to compare all CXCR2 ligands with regard to their various properties in all types of cancer or addressed the issue of why evolution caused the emergence of a large number of factors with the same properties in a single genome.

An analysis of more than one CXCR2 ligand in a given process is rarely performed due to the costs associated with research and the lack of appropriate research tools. Bioinformatic analysis provides a feasible solution, enabling researchers to analyze the importance of any chosen gene in the tumorigenesis of all major tumors within a relatively short period and with practically zero cost. Thus, the aim of this study was to conduct an analysis of the significance of CXCR2 ligands in tumor processes and to demonstrate the differences in the functions they perform in tumorigenesis.

## 2. Results

### 2.1. The Level of Expression of CXCR2 Ligands in the Tumor Compared to Healthy Tissue Depend on the Type of Cancer

Among the seven CXCR2 ligands, most of them frequently showed increased expression in many tumors (Table 2). *CXCL8* expression was elevated in 14 out of 31 analyzed tumors and decreased in four types of tumors. *CXCL1* expression was elevated in 13 and *CXCL3* in nine types of tumors. *CXCL2* and *PPBP* expression was often decreased in tumors compared to healthy tissue. *CXCL2* was downregulated in eight out of thirty-one analyzed tumors, while this chemokine was only upregulated in four out of thirty-one tumors. *PPBP* was another ligand that frequently showed decreased expression. *PPBP* expression was reduced in four out of thirty-one analyzed tumors and increased in only two tumors. These results demonstrate a diversity of CXCR2 ligand expression patterns in different types of tumors.

### 2.2. Correlation of CXCR2 Ligand Expression with Prognosis

The correlation between CXCR2 ligand expression within tumors and patient prognosis was examined across 31 cancer types. However, it is essential to emphasize that this analysis only reflects correlation, which might indicate either anti-tumor or pro-tumor properties of CXCR2 ligands. It could also imply alternative associations, such as improved prognosis leading to distinct CXCR2 ligand expression, without the ligands participating in tumorigenesis. It is important to note that even the identification of correlation might not necessarily imply a causal relationship.

The analysis of 31 different types of cancer revealed that in eight cancer types, the expression levels of more than one CXCR2 ligand were correlated with prognosis. Within almost all of these types, the correlation was either positive or negative exclusively. The exception was brain lower-grade glioma, where higher CXCL1 expression correlated with worse prognosis, while CXCL2 and CXCL5 expression correlated with better outcomes. The most frequent occurrence (seven out of thirty-one) was the correlation of high CXCL8 expression with poorer prognoses. Conversely, CXCL2 was the most commonly correlated (four out of thirty-one) with better prognoses (Table 3).

### 2.3. Only in 5 out of 20 Types of Cancers, Certain CXCR2 Ligands May Positively Correlate with Lymph Node Metastasis Status

The association between the level of expression of CXCR2 ligands and lymph node metastasis status was analyzed using the UALCAN portal. Of the 31 types of tumors analyzed, 20 were available for analysis, as some tumors do not lead to lymph node metastasis. Among the 20 different types of tumors analyzed, in one tumor, some CXCR2 ligands were associated with worse lymph node metastasis status, which occurred in five types of tumors. However, in eight types of tumors, some CXCR2 ligands were associated with better lymph node metastasis status. Furthermore, in a given tumor, there were no CXCR2 ligands that were associated with both better and worse lymph node metastasis status. This suggests that in a given type of tumor, all CXCR2 ligands are associated with either better or worse lymph node metastasis status.

It is not possible to unequivocally identify which CXCR2 ligand in pan-cancer analysis is consistently associated with better or worse lymph node metastasis status (Table 4). For example, CXCL8 was associated with better lymph node metastasis status in breast invasive carcinoma and head and neck squamous cell carcinoma, whereas the same chemokine was linked with worse lymph node metastasis status in esophageal carcinoma and thyroid carcinoma. CXCL2 was the CXCR2 ligand that was most frequently linked with lymph node metastasis status across cancer types. Higher expression of this chemokine was associated with better lymph node metastasis status in seven cancer types but with worse lymph node metastasis status in two types.

### 2.4. The Expression of CXCL2 and PPBP Often Negatively Correlate with a Proliferation Marker

To investigate the association between proliferation and CXCR2 ligands, the correlation between the expression level of each CXCR2 ligand and the expression level of the proliferation marker Ki-67/*MKI67* was analyzed in 31 types of cancer using the GEPIA portal.

CXCL2 and PPBP were CXCR2 ligands that were most often negatively correlated with the proliferation marker (Table 5). *CXCL2* expression negatively correlated with the proliferation marker in five types of tumors, but positively correlated in nine types of tumors. On the other hand, PPBP expression negatively correlated with proliferation in six types of tumors, while it positively correlated with the proliferation marker in only five types of tumors. The results indicate that CXCR2 ligands may be associated with proliferation, and the type of relationship is specific to each type of tumor. Another conclusion is that CXCL2 and PPBP may be associated with the inhibition of proliferation in the tumor, but only in some types of tumors.

### 2.5. The Correlations between the Expression of CXCR2 Ligands and Three Epithelial-to-Mesenchymal Transition (EMT) Markers Indicate Their Distinct Roles and Relationships in Some Types of Tumors

We analyzed the correlation of three previously studied EMT markers with the expression of CXCR2 ligands to identify new relationships with the EMT process in a given tumor.

In colon adenocarcinoma, *CXCL1*, *CXCL2*, and *CXCL3* negatively correlated with EMT, while *CXCL5*, *CXCL6*, *PPBP*, and *CXCL8* positively correlated with EMT. Similar correlations were observed in esophageal carcinoma, with *CXCL1*, *CXCL2*, *CXCL3*, and *PPBP* negatively correlated with EMT, *CXCL5* and *CXCL6* positively correlated with EMT, and *CXCL8* not correlated with EMT. In rectum adenocarcinoma, *CXCL1* and *CXCL3* negatively correlated with EMT, while *CXCL5*, *CXCL6*, *PPBP*, and *CXCL8* were positively correlated.

In some types of cancer, the expression of CXCR2 ligands negatively correlated with EMT. This was the case in stomach adenocarcinoma, except for chemokines *CXCL2*, *CXCL5*, and *PPBP*, where no correlation was found (Table 6, Table 7 and Table 8).

### 2.6. In Most Cases, the Expression of CXCR2 Ligands Is Not Correlated with the Level of Infiltration of the Tumor Microenvironment by T_reg_ Cells

In order to investigate the relevance of CXCR2 ligands in relation to tumor-associated cells, the correlation between CXCR2 ligands and infiltration levels of CD8^+^ T cells, conventional and plasmacytoid dendritic cells, MDSCs, NK cells, macrophages, neutrophils, and T_reg_ cells were analyzed. CD8^+^ T cells, dendritic cells, and NK cells are immune cells that act against tumors by destroying tumor cells [62,63,64,65,66] but advanced tumors have multiple mechanisms to inhibit their activity. MDSCs and T_reg_ cells are immunosuppressive cells that inhibit the anti-tumor response of the immune system, thereby promoting tumor growth [67,68]. Neutrophils exhibit plasticity and can have both pro- and anti-tumor effects [69,70], and the tumor microenvironment can transform neutrophils into pro-tumor cells.

The expression of CXCR2 ligands was analyzed in relation to the level of T_reg_ cell recruitment using the TIMER2.0 portal. A positive correlation indicated a relationship between CXCR2 ligands and T_reg_ cell recruitment and, therefore, tumor immune evasion. In 14 types of tumors, the expression of at least one CXCR2 ligand was positively correlated with the count of T_reg_ cells in the tumor microenvironment (Table 9). In four types of tumors, the expression of at least one CXCR2 ligand was negatively correlated with the count of T_reg_ cells in the tumor microenvironment.

### 2.7. The Expression of CXCR2 Ligands Always Positively Correlates with the Level of Neutrophil Infiltration

In the majority of analyzed tumors, the expression of at least one CXCR2 ligand positively correlated with the recruitment of neutrophils to the tumor microenvironment (Table 10).

### 2.8. The Impact of CXCR2 Ligands on the Level of MDSC Recruitment Depends on the Type of Tumor

The correlation between CXCR2 ligands and the count of MDSCs in the tumor microenvironment was analyzed. The expression of all analyzed CXCR2 ligands positively or negatively correlated with the count of MDSCs depending on the type of tumor. The expression of no CXCR2 ligand exclusively positively or negatively correlated with the count of MDSCs. However, *CXCL2* was the most frequently negatively correlated CXCR2 ligand with the count of MDSCs. In 14 types of tumors, the expression of *CXCL2* negatively correlated with the count of MDSCs. However, in only five types of tumors, it positively correlated with the count of analyzed cells (Table 11).

### 2.9. The Expression Level of CXCR2 Ligands Negatively Correlates with Tumor Infiltration by CD8^+^ T Cells in Most Cancers

The correlation between the expression of CXCR2 ligands and the level of tumor infiltration by CD8^+^ T cells was analyzed in different types of cancers. These cells act against cancer [62,66]. Therefore, understanding the relationship between CXCR2 ligands and these cells allows for the analysis of the influence of CXCR2 ligands on cancer processes, such as anti-tumor processes.

In 18 out of 30 analyzed types of tumors, the expression level of at least one CXCR2 ligand negatively correlated with the level of tumor infiltration by CD8^+^ T cells, and none of the CXCR2 ligands positively correlated with the count of these cells. In four types of tumors, the expression level of some CXCR2 ligands positively correlated with the count of CD8^+^ T cells (Table 12).

### 2.10. In Most Types of Tumors, None of the CXCR2 Ligands Correlates with Tumor Infiltration by NK Cells

Out of the 30 types of tumors analyzed, only in 11 types, the expression of at least one CXCR2 ligand correlated with the level of tumor infiltration by NK cells. In brain lower grade glioma and diffuse large B-cell lymphoma, it was a positive correlation. In the remaining nine types, the expression level of some CXCR2 ligands negatively correlated with the level of tumor infiltration by NK cells (Table 13).

### 2.11. The Expression Level of CXCR2 Ligands Often Positively Correlates with the Count of Myeloid Dendritic Cells in the Tumor

The correlation between the expression level of CXCR2 ligands and the count of dendritic cells (DCs) in the tumor of 30 different types of cancer was analyzed using the TIMER2.0 portal. It was possible to analyze two subsets of DCs: myeloid and plasmacytoid DCs. Myeloid DCs are currently classified into two subsets: conventional type 1 and conventional type 2 DCs [65]. Therefore, the correlation analysis of tumor infiltration by conventional (myeloid) and plasmacytoid DC was performed in the context of the expression level of CXCR2 ligands.

In 19 types of tumors, the expression of at least one CXCR2 ligand was only positively correlated with the count of conventional (myeloid) DCs in the tumor, while in esophageal carcinoma, diffuse large B-cell lymphoma, cervical squamous cell carcinoma, and endocervical adenocarcinoma, the correlation was negative. In two types of tumors (liver hepatocellular carcinoma and thymoma), some ligands were positively correlated, while others were negatively correlated with the level of infiltration of conventional (myeloid) DCs. In six types of tumors, the expression level of none of the CXCR2 ligands was correlated with the infiltration of conventional (myeloid) DCs in the tumor (Table 14).

### 2.12. In One-Third of Cancer Types, the Expression of CXCR2 Ligands Positively or Negatively Correlates with the Count of Plasmacytoid Dendritic Cells

In 10 types of cancer, the expression level of at least one CXCR2 ligand negatively correlated with tumor infiltration by plasmacytoid DCs, while none of the CXCR2 ligands positively correlated with the analyzed cells. In nine types of cancer, the expression level of at least one CXCR2 ligand positively correlated with tumor infiltration by plasmacytoid DCs, while none of the CXCR2 ligands negatively correlated with the count of DCs. In another eight types of cancer, the expression level of none of the CXCR2 ligands significantly correlated with the count of plasmacytoid DCs in the tumor. In three types of cancer, some of the CXCR2 ligands negatively correlated with the count of plasmacytoid DCs, while others were positively correlated.

The expression level of *CXCL2* was not negatively correlated with the count of plasmacytoid DCs in any of the analyzed types of cancer. It was either positively correlated or not significantly correlated with the analyzed cells (Table 15). For the remaining CXCR2 ligands, depending on the type of cancer, they either positively or negatively correlated or did not significantly correlate with the count of DCs.

### 2.13. In Some Tumors, the Expression of CXCR2 Ligands Negatively Correlates with the Count of Endothelial Cells

The correlation between the expression level of CXCR2 ligands and the number of endothelial cells in the tumor microenvironment was analyzed using the TIMER2.0 database. Endothelial cells constitute the building blocks of blood vessels; hence, the positive correlation of a given ligand with the presence of these cells suggests a potential relationship with angiogenesis.

In 16 out of the 30 types of analyzed tumors, some of the CXCR2 ligands were positively correlated with the number of endothelial cells. Notably, the pattern of ligands positively correlating with these cells varied among different types of tumors. On the other hand, in six types of tumors, the expression level of CXCR2 ligands was only negatively correlated with the number of endothelial cells, indicating an association with the inhibition of angiogenesis and higher expression of these chemokines. In five types of tumors, some of the CXCR2 ligands were negatively correlated, while others were positively correlated with the number of endothelial cells (Table 16).

### 2.14. The Level of CXCR2 Ligand Expression Positively Correlates with the Count of Macrophages in Most Types of Tumors

In 19 types of tumors, which represent almost two-thirds of the investigated types, the expression of some CXCR2 ligands was only positively correlated with the count of macrophages in the tumor microenvironment. The correlation between the expression of individual CXCR2 ligands and macrophage infiltration was specific to each tumor type. In five types of tumors, the expression of some CXCR2 ligands was only negatively correlated with the count of macrophages in the tumor microenvironment. In five types of tumors, some CXCR2 ligands were positively correlated with macrophage infiltration, while others were negatively correlated (Table 17).

### 2.15. In Most Types of Cancer, the Level of Expression of CXCR2 Ligands Positively Correlates with the Count of M1 Macrophages in the Tumor Microenvironment, and Negatively Correlated with the Count of M2 Macrophages

The correlation between CXCR2 ligand expression and the count of M1 and M2 macrophages was assessed using the TIMER2.0 platform. M1 macrophages are pro-inflammatory macrophages with anti-tumor properties [71], whereas M2 macrophages are anti-inflammatory and immunosuppressive, and are involved in promoting tumor growth. Analyzing the correlation between CXCR2 ligand expression and both types of macrophages should provide insight into the nature of the chemokines under investigation, and whether they are associated with the anti-tumor or pro-tumor characteristics of the tumor microenvironment.

The correlation between the two types of macrophages was specific to the tumor type (Table 18 and Table 19). For example, in cervical squamous cell carcinoma and endocervical adenocarcinoma, the expression of CXCR2 ligands was negatively correlated with both the count of M1 and M2 macrophages. In contrast, in bladder urothelial carcinoma, the expression of CXCR2 ligands was positively correlated with the count of M1 macrophages but negatively correlated with the count of M2 macrophages. Finally, in pheochromocytoma and paraganglioma, the expression of CXCR2 ligands was positively correlated with both the count of M1 and M2 macrophages. CXCR2 ligand expression was usually negatively correlated with the count of M2 macrophages. However, CXCL2 was a chemokine that differed from the other CXCR2 ligands. The expression of this chemokine was positively correlated with the count of M2 macrophages in eight types of tumors and negatively correlated in five types of tumors. This indicates that this chemokine has a stronger pro-tumor effect compared to the other CXCR2 ligands. However, the precise effects depend on the tumor type.

### 2.16. In Cancer Diseases, Mutations in CXCR2 Ligand Genes Occur at a Frequency Ranging from 1.1% to 1.3% of All Cancer Cases

Considering all the analyzed cancers, mutations in CXCR2 ligand genes occurred at a frequency ranging from 1.1% to 1.3% of all cancer cases (Figure 1). These were mainly amplifications, whose frequency was similar among all CXCR2 ligands, ranging from 0.83% to 0.89% of cases. In total, 102 cases of amplifications were identified in the 10,783 cases analyzed. Of these, 83 patients had amplifications of all CXCR2 ligand genes. Missense mutations were less frequent, and there were also few cases of deletions, with 12 cases of deletion involving any of the CXCR2 ligand genes. In nine cases, all CXCR2 ligand genes were deleted. There were also individual cases of truncating mutations and splice mutations in CXCR2 ligand genes. One case of a fusion gene, CXCL1-AFP, was detected in head and neck squamous cell carcinoma.

The level of mutations in *CXCR1* and *CXCR2* genes in tumors was 1.4% and 1.5%, respectively. About half of the identified mutations in these genes were missense mutations. Amplifications and deletions occurred at a frequency of 0.33% for each type of mutation. Of the 38 cases of amplification, both receptor genes were simultaneously mutated in 34 cases. In 37 cases of deletion, all cases involved both receptor genes. This indicates that in the majority of cases, amplification and deletion simultaneously affect both CXCR1 and CXCR2 receptor genes.

The frequency of amplifications and deletions of CXCR2 ligand genes varied depending on the type of cancer analyzed. For example, in thyroid carcinoma, kidney renal papillary cell carcinoma, and acute myeloid leukemia, no amplifications or deletions were found. In 20 types of cancer, individual cases of amplification and even rarer deletions occurred. However, in esophageal carcinoma, ovarian serous cystadenocarcinoma, lung squamous cell carcinoma, and breast invasive carcinoma, more than 2% of cases had amplifications of CXCR2 ligand genes (Table 20).

### 2.17. Only Some CXCR2 Ligand Proteins Are Very Similar to Each Other

Above, an analysis was conducted to correlate the expression of CXCR2 ligands with prognosis, proliferation markers, EMT, and the recruitment of cells to the tumor microenvironment. This analysis revealed significant differences among various CXCR2 ligands. Within a given cancer type, many of these ligands exhibit opposing functions and properties in certain analyses. Consequently, in order to determine whether CXCR2 ligands also differ in terms of their sequences to the same extent, a comparison of sequence similarities between CXCR2 ligands was conducted.

BLAST analysis was used to investigate the similarity of CXCR2 ligand proteins. Only some CXCR2 ligand proteins showed high sequence similarity with each other. Specifically, sequence similarity between CXCL1, CXCL2, and CXCL3 was around 85% (Figure 2). Similarly, there was a 76% similarity between CXCL5 and CXCL6. In contrast, the sequence similarity between CXCL7 and CXCL1, CXCL2, and CXCL3 was only about 50%. The sequence similarity between other CXCR2 ligand proteins was below 50%, with CXCL8 being the most divergent from the others.

All CXCR2 ligand proteins have a conserved ELRCXC sequence at the N-terminus. However, the N-terminal fragment before this sequence often differed between CXCR2 ligands, with variation in length before the ELRCXC sequence, such as one amino acid in CXCL7 or nine amino acids in CXCL5. Additionally, the N-terminal sequences also often differed between CXCR2 ligands, for example, between CXCL3 and CXCL8. Differences in amino acid sequence at the C-terminus were also frequent, such as the low sequence similarity observed between CXCL3 and CXCL7.

### 2.18. Regulation of CXCR2 Ligand Transcription May Be Mediated by NF-κB and a Unique Set of Proteins Bound Upstream of the Transcription Start Site

The potential DNA binding proteins from the transcription start site up to 1500 bp upstream of the CXCR2 ligand gene transcription initiation site were analyzed using the Tfsitescan tool. The obtained results indicate very large differences in the regulation of CXCR2 ligand gene expression, but also some similarities. Comparing the 200 bp fragment closest to the transcription start site of *CXCL1* and *CXCL2* promoters, significant differences in identified transcription factors were found, much greater than suggested by the similarity of the amino acid sequences of these two chemokines. The coding sequences of CXCL1 and CXCL2 were 93% similar. Analyzing the 200 bp fragment upstream of the transcription start site of *CXCL1* and *CXCL2* genes, five transcription factors were identified, with NF-κB binding sites present in the promoters of *CXCL1* and *CXCL2*. The second similarity was the EGR1 binding site, although these sites differed partially between the promoters of these chemokines (Table 21). Much larger differences were observed when comparing other chemokines to each other.

The analysis of the 200 bp fragment closest to the transcription start site showed that NF-κB binds to the analyzed sequences of six out of seven CXCR2 ligand promoters. The CXCR2 ligand promoter that does not have an NF-κB binding site is *PPBP*. In the analyzed sequences, Sp1 was also bound to five out of seven CXCR2 ligand promoters. There were significant differences in the analyzed sequences upstream of the transcription start site of CXCR2 ligands. We found examples of identified potential DNA-binding proteins that only bind to the analyzed sequences of one or two CXCR2 ligands, such as c-Myb and *CXCL2*, MZF-1 and *CXCL5*, H4TF1/IKAROS/LYF-1 and *CXCL5*, PuF site in *PPBP*.

The analysis of the fragment from 1500 bp to 200 bp upstream of the transcription start site showed many potential DNA binding proteins that can bind to the analyzed sequences. There were significant differences between the analyzed CXCR2 ligands, but common features could also be identified. PEA-3 bound to the analyzed sequences of six out of seven CXCR2 ligand promoters. No binding site for this protein was identified for *CXCL2*. GATA-1 binds upstream of the transcription start site of five ligands. No binding site for this protein was identified upstream of the transcription start site of *CXCL2* and *PPBP*. However, unique DNA binding proteins for each CXCR2 ligand promoter that bind to the analyzed sequence could be identified, such as FREAC-2 for *CXCL1* and *CXCL3*, Wt1 and HNF-3 for *CXCL2*, NMP-2 for *CXCL2*, TIN-1 for *CXCL3*, HNF-6 for *CXCL6*, HBP1 for *PPBP*, and Gfi-1 for SF-1 for *CXCL8*.

The results indicate that CXCR2 ligand gene expression is regulated by NF-κB. However, there were significant differences in the DNA binding proteins that were bound to the DNA upstream of the transcription start site.

### 2.19. The Expression of Each CXCR2 Ligand Is Regulated by a Unique Set of microRNAs

The regulation of gene expression can occur at the mRNA level. One of the mechanisms for reducing the level of specific mRNAs is microRNA. These are short RNAs that are part of the RNA-induced silencing complex (RISC) [72]. Active RISC with microRNA scans mRNA in search of a complementary sequence to microRNA. When such a sequence is found in mRNA, the transcript is destroyed. Consequently, translation does not occur, and the expression of a specific gene is reduced.

In addition to transcription factor binding to the promoter, the expression of CXCR2 ligands can be regulated by microRNAs. To analyze potential microRNAs regulating CXCR2 ligand expression, miRDB was utilized. Overall, 87 different microRNAs with target scores ranging from 85 to 100 were identified (Appendix A).

Fifty-seven microRNAs were found to regulate the expression of only one CXCR2 ligand, while 26 microRNAs were found to regulate the expression of two ligands (Table 22). In most cases, a microRNA regulates the expression of one CXCR2 ligand with a high probability but with a very low probability for the second CXCR2 ligand. However, nine microRNAs were identified that regulate the expression of two CXCR2 ligands with high probability, including miR-95-5p for CXCL1 and CXCL2, miR-532-5p for CXCL1 and CXCL2, and miR-889-3p for CXCL5 and CXCL8. Three microRNAs were found to regulate the expression of three CXCR2 ligands, and only one microRNA, miR-5692a, regulated the expression of four CXCR2 ligands.

The data obtained showed that the expression of each CXCR2 ligand was regulated by a unique set of microRNAs.

To confirm the results, the 3′-UTR mRNA sequences of CXCR2 ligands were compared using BLAST. The most similar ligands, CXCL1 and CXCL2, were compared, with a coding sequence similarity of 93% and a 3′-UTR similarity of 87%. Similar patterns were observed when comparing CXCL5 and CXCL6. However, the 3′-UTR similarity between other pairs of CXCR2 ligands was too low to analyze using BLAST.

Small differences between the 3′-UTR of CXCL1 and CXCL2 introduce large differences in the regulation of gene expression by microRNAs (Figure 3). When comparing the 3′-UTRs with the most likely binding sites for microRNAs, 16 differences were observed in the nucleotide sequences between CXCL1 and CXCL2. As a result, two binding sites regulate only CXCL1, seven binding sites regulate only CXCL2, and in six binding sites, both chemokines are regulated.

## 3. Discussion

### 3.1. Regulation of CXCR2 Ligand Expression

The regulation of each CXCR2 ligand expression varied significantly. These differences were much greater than what would be expected from differences in amino acid sequence or coding sequence of the CXCR2 ligand genes. The data also showed significant differences in the regulation of CXCR2 ligand expression by microRNA and proteins binding near the CXCR2 ligand gene promoters. These differences explain the differences in the expression of individual CXCR2 ligands in a given tumor type, differences in CXCR2 ligand expression between a tumor and healthy tissue, as well as differences in CXCR2 ligand expression between different types of tumors. They result in the involvement of different CXCR2 ligands in various cancer processes and differences in the patterns of CXCR2 ligand involvement in cancer processes between different types of tumors.

Various types of tumors may differ in terms of microRNA or proteins directly responsible for regulating the expression of a given CXCR2 ligand. Particularly, the overexpression or loss of expression of a regulatory element (such as a miRNA that suppresses expression or a transcription factor binding to enhancers or silencers) results in changes in the expression of genes regulated by these factors. This results in an increase or decrease in the expression of one CXCR2 ligand in one type of tumor but not in another type.

Significant differences in CXCR2 ligand expression also allow for changes in the expression of only specific ligands under the influence of a given factor. This is why CXCR2 ligands can serve different functions in cancer processes within one tumor and why one CXCR2 ligand can serve different functions in two different types of tumors.

### 3.2. Mutation in CXCR2 Ligand Genes

The level of mutation in the CXCR2 ligand genes and the receptors themselves in tumors was low. It is estimated that slightly over 1% of tumor cases have a mutation in a given CXCR2 ligand gene. The most common mutations are amplifications of all CXCR2 ligand genes. However, such a low frequency may not affect the level of CXCR2 ligand expression when analyzing all cases of a given type of tumor. The simultaneous amplification of all CXCR2 ligand genes was related to the proximity of the genes encoding these chemokines. All CXCR2 ligand genes form a gene cluster located at 4q12-q13 [59]. Amplification of this entire gene cluster results in the amplification of all CXCR2 ligand genes.

In tumors, mutations in the CXCR2 ligand receptor genes, *CXCR1* and *CXCR2*, were mainly amplifications and deletions, with each type of mutation occurring in approximately 0.33% of cases. Moreover, the genes for both receptors underwent the same mutation. This is because the *CXCR1* and *CXCR2* genes, together with the pseudogene for these chemokine receptors, form a gene cluster located at 2q34-q35 [16]. Amplification and deletion affect the entire gene cluster.

### 3.3. Correlation between CXCR2 Ligands Expression and Proliferation

In the majority of cases, CXCR2 ligand expression positively correlated with proliferation in tumor tissues, as observed in in vitro experiments. The activation of the CXCR2 receptor leads to proliferation, which is associated with the transactivation of EGFR [73,74]. CXCR2 activation also reduces the expression of p21 and increases the expression of cyclins and cyclin dependent kinases such as cyclin A, cyclin B1, cyclin D1, cyclin E, CDK2, and CDK6, thereby promoting the increased proliferation of cancer cells [52]. In vitro studies on cell lines have confirmed that CXCR2 activation increases proliferation in various types of cancer, including colon cancer [75], esophageal carcinoma [76], gastric cancer [77], malignant melanoma [78], and ovarian cancer [74]. These results confirm the correlation between CXCR2 ligand expression and the proliferation marker Ki-67 in various types of cancer, indicating that CXCR2 ligands increase proliferation in some tumors. However, in some types of cancer, only the expression of certain CXCR2 ligands positively correlated with proliferation, suggesting that only some CXCR2 ligands are associated with or increase proliferation in certain tumors.

Interestingly, CXCR2 activation may not increase or may even inhibit proliferation in certain types of cancer. CXCR2 ligand expression was negatively correlated with proliferation in lung squamous cell carcinoma and brain tumors (lower grade glioma and glioblastoma multiforme). In addition, the expression of some CXCR2 ligands was negatively correlated with proliferation, while others were positively correlated with the examined tumor process. CXCL2 and CXCL7 were the CXCR2 ligands whose expression was most commonly negatively correlated with proliferation. It should be noted that CXCL7 was not specifically analyzed, unlike PPBP, which generates CTAP-III, β-TG, or CXCL7/NAP-2 after the removal of its N-terminus [10]. The negative correlation between CXCR2 ligand expression and proliferation may be due to the direct inhibition of proliferation by CXCR2 ligands or the involvement of these chemokines in anti-tumor processes that result in proliferation inhibition.

In cholangiocarcinoma, in vitro experiments revealed that CXCL1 inhibits the proliferation of OCUG-1 and HuCCT1 tumor cells [79]. Similarly, CXCL2 curbs the proliferation of hepatocellular carcinoma tumor cells (HCCLM3 and MHCC97H) [80]. Moreover, research on various clones of the A549 line (lung adenocarcinoma) demonstrates that CXCL8/IL-8 can impede tumor cell proliferation [81].

Our paper highlighted the absence of a correlation between CXCR2 ligand expression and proliferation in certain types of tumors, including cholangiocarcinoma. Existing in vitro studies indicate that CXCR2 ligands suppress the proliferation of cholangiocarcinoma cancer cells [79]. The observed lack of correlation might stem from tumor development mechanisms, whereby anti-tumor processes in advanced cancers occur less frequently than predicted by in vitro studies. The negative correlation between CXCR2 ligand expression and proliferation may result from the direct inhibition of proliferation by CXCR2 ligands, or from the involvement of the analyzed chemokines in anti-tumor processes that ultimately lead to the inhibition of proliferation.

There are three possible mechanisms by which CXCR2 ligands can inhibit proliferation. The first mechanism involves atypical chemokine receptors, such as atypical chemokine receptor 1 (ACKR1)/Duffy antigen receptor for chemokines (DARC). This receptor can bind CXCR2 ligands, as well as other chemokines, such as CC chemokine ligand 5 (CCL5)/Regulated on Activation Normal T cell Expressed and Secreted (RANTES) [82,83]. The binding of chemokines to this receptor may inhibit the activity of CXCR2 if both receptors are located on the same cancer cell [84]. Thus, the CXCR2 ligands described here increase proliferation through CXCR2, although they can also inhibit this process by binding to ACKR1/DARC.

The second possible explanation is the interaction between CXCR2 ligands. CXCR2 ligands can form homodimers and heterodimers [35,36,37]. Some heterodimers of CXCR2 ligands activate CXCR2 at lower concentrations compared to monomers [37]. Therefore, it can be assumed that the heterodimers of CXCR2 ligands can activate CXCR2 more or less strong compared to a single CXCR2 ligand. An increase in the expression of one CXCR2 ligand may result in a change in the activity of other CXCR2 ligands, which may lead to the inhibition of proliferation. However, the research on the effects of CXCR2 ligand heterodimers is still limited.

The third possible explanation is the involvement of certain CXCR2 ligands in a process that affects proliferation. In breast invasive carcinoma, among the CXCR2 ligands, only CXCL2 inhibits proliferation, and this chemokine was the only one among those analyzed that was negatively correlated with the count of macrophages. In liver hepatocellular carcinoma and lung adenocarcinoma, CXCL2 and PPBP negatively correlated with proliferation, and were the only CXCR2 ligands that negatively correlated with the count of MDSCs. A similar relationship can be observed in thymoma. It is likely that some CXCR2 ligands reduce the recruitment of macrophages and MDSCs to the tumor microenvironment, which are cells that promote tumor cell proliferation.

### 3.4. Correlation between the Expression of CXCR2 Ligands and EMT

In a majority of tumor types, there was a positive correlation between the expression of CXCR2 ligands and EMT. This observation was in line with the expected role of CXCR2 ligands, which are chemotactic cytokines belonging to the chemokine family.

A defining property of chemokines in the immune system is their ability to influence cell migration. Within tumors, CXCR2 ligands are capable of inducing EMT, a phenomenon that has been demonstrated in vitro with breast cancer cells (CXCL8/IL-8) [85], colon cancer cells (CXCL5) [86], hepatocellular carcinoma cells (CXCL5) [87], ovarian cancer cells (CXCL8/IL-8) [88], pancreatic cancer cells (CXCL5) [89], and prostate cancer cells (CXCL1) [90].

It is worth noting that the expression of CXCR2 ligands is also increased by EMT [91,92], which is a result of NF-κB activation. In the case of CXCL1 and CXCL2, EMT leads to the increased expression of these two chemokines through Snail binding to their promoters [91]. Both factors, the induction of EMT by CXCR2 ligands and the increased expression of CXCR2 ligands due to EMT, result in a positive correlation between CXCR2 ligand expression and EMT markers. This relationship was also established in the current investigation.

It is important to note that nearly all available in vitro and in vivo studies that have investigated the impact of CXCR2 ligands on EMT focus on the induction of EMT following exposure of cancer cells to a specific CXCR2 ligand. Some in vivo studies assessed the intensity of EMT following overexpression of a given CXCR2 ligand within the tumor. The results of these experiments showed that CXCR2 ligands induce EMT, which could occur locally in small regions of the tumor or in individual cancer cases. However, this research model may not fully capture the significance of CXCR2 ligands in tumors.

To highlight the importance of individual CXCR2 ligands in EMT, we analyzed the correlation between the expression of the analyzed chemokines and the expression of EMT markers in a given tumor. In six types of tumors, the expression of some CXCR2 ligands negatively correlated, while that of others positively correlated with EMT in a single type of tumor. For example, in colon adenocarcinoma, the expression of CXCL1, CXCL2, and CXCL3 negatively correlated with EMT, whereas the expression levels of CXCL5, CXCL6, PPBP, and CXCL8 positively correlated with EMT. In another example, cervical squamous cell carcinoma and endocervical adenocarcinoma exhibited a negative correlation between the expression of CXCL1, CXCL6, and CXCL8 and EMT, whereas the expression of CXCL3, CXCL5, and PPBP positively correlated with EMT.

These findings demonstrate that higher expression of some CXCR2 ligands in different types of tumors may be associated with EMT, but the lower expression of other CXCR2 ligands may also be associated with the same process within the same type of tumor. Additionally, in many types of tumors, the expression of CXCR2 ligands positively correlated with markers of the epithelial and mesenchymal phenotype.

The causes of the occurrence of CXCR2 ligands with opposite correlations with EMT, or the presence of chemokines that are positively correlated with both phenotypes, are not fully understood, and the differences in the actions of CXCR2 ligands remain poorly characterized. It is possible that different CXCR2 ligands are involved in different stages of a given process. For instance, in tissue infiltration by neutrophils, CXCL1/KC and CXCL2/MIP-2 have been shown to be responsible for the chemotaxis and transendothelial migration of neutrophils, respectively, in a mouse model [61]. Similarly, in some tumors, different CXCR2 ligands may be responsible for different stages or elements of EMT.

Moreover, the interaction between individual CXCR2 ligands may offer a possible explanation for the fact that the expression of some CXCR2 ligands shows opposite correlations with EMT. CXCR2 ligands form homodimers and heterodimers with one another [35,36,37]. Some of the heterodimers of these chemokines may activate the CXCR2 receptor with better or worse parameters [37]. Consequently, the increased expression of one of the CXCR2 ligands may cause a change in the actions of the remaining CXCR2 ligands, which may result in the inhibition of EMT. However, the research on the effects of CXCR2 ligand heterodimers is limited, and more investigations are needed to elucidate their role in regulating EMT.

The expression of none of the CXCR2 ligands correlated with EMT in adrenocortical carcinoma and cholangiocarcinoma. An in vitro study has shown, CXCR2 ligands, particularly CXCL1, inhibit the migration of cholangiocarcinoma cells [79]. However, in cholangiocarcinoma, lymphatic endothelial cells secrete CXCL5, which induces EMT in the tumor cells [93]. Similarly, in stomach adenocarcinoma, CXCL5 has been shown to induce EMT in tumor cells according to an available in vitro study [94]. Nevertheless, no significant correlation was found in this study between the expression of CXCR2 ligands and EMT in cholangiocarcinoma. These properties of CXCR2 ligands may not have a significant impact on EMT in this particular tumor. Other factors may be responsible for EMT in cholangiocarcinoma.

Moreover, in some types of tumors, the expression level of CXCR2 ligands negatively correlated with EMT. This correlation has been observed in stomach adenocarcinoma with respect to CXCL1, CXCL3, CXCL6, and CXCL8. CXCR2 ligands may directly induce EMT, but they may also act on some tumor processes that can inhibit EMT, resulting in a negative correlation between the expression of some CXCR2 ligands and EMT.

### 3.5. Correlation between CXCR2 Ligand Expression and Lymph Node Metastasis Status

In most cases, the expression of CXCR2 ligands was not linked to the status of lymph node metastasis. The occurrence of lymph node metastasis is likely to be influenced by other unrelated factors, including more than 40 various chemokines other than CXCR2 ligands [2], prostaglandins [95], growth factors [96], and many other factors.

Moreover, a better lymph node metastasis status was often observed with higher expression of certain CXCR2 ligands in some types of tumors. This can be explained by the negative correlation between the level of CXCR2 ligand expression and EMT in stomach adenocarcinoma and rectum adenocarcinoma. In breast invasive carcinoma, lung squamous cell carcinoma, prostate adenocarcinoma, and kidney renal papillary cell carcinoma, CXCR2 ligands negatively correlated with EMT but positively correlated with lymph node metastasis. Metastasis is a multi-step process, and EMT is only one of the stages. Some CXCR2 ligands in some types of tumors may act on other stages of lymph node metastasis. They may affect the pre-metastatic niche, which prevents metastasis, but this requires further investigation.

In isolated cases, the higher expression of certain CXCR2 ligands was associated with lymph node metastasis. This association can be explained by EMT of tumor cells induced by some CXCR2 ligands, leading to the migration of tumor cells from the primary tumor and consequent lymph node metastasis. In this work, lymph node metastasis in skin cutaneous melanoma was associated with CXCL2, in liver hepatocellular carcinoma with CXCL5, in kidney renal clear cell carcinoma with PPBP, and in esophageal carcinoma with CXCL8/IL-8. In the case of thyroid carcinoma, CXCR2 ligand expression was associated with lymph node metastasis, but it appeared to be negatively correlated with EMT. CXCR2 ligands in thyroid carcinoma probably cause lymph node metastasis, but not during the induction of EMT. In patients with papillary thyroid cancer, lymph node metastasis is correlated with the neutrophil-to-lymphocyte ratio [97]. CXCR2 ligands increase the count of neutrophils in the blood by causing the egress of these cells from the bone marrow [98]. This suggests that CXCR2 ligands may increase the likelihood of lymph node metastasis by increasing the count of neutrophils.

### 3.6. Correlation between CXCR2 Ligand Expression and Angiogenesis

Endothelial cells express CXCR2 [54,99], making CXCR2 ligands pro-angiogenic [53]. They induce tube formation and proliferation of endothelial cells, as demonstrated by the study on bovine adrenal gland capillary endothelial cells [53], human umbilical vein endothelial cells (HUVEC) [100,101,102,103], human dermal microvascular endothelial cells [54,101], human lung microvascular endothelial cells [54], and human brain microvascular endothelial cells [103]. These findings partly confirm the correlation between CXCR2 ligand expression and the count of endothelial cells in tumors. In 21 types of cancer, the expression of at least one CXCR2 ligand positively correlated with the count of endothelial cells in the tumor. However, in five of these types of cancer, the expression of certain CXCR2 ligands negatively correlated with the count of endothelial cells in the tumor.

In six types of cancer, the expression of at least one CXCR2 ligand negatively correlated with the count of endothelial cells in the tumor, in the absence of CXCR2 ligands whose expression positively correlated with these cells. This may be due to several factors. In some models, CXCR2 ligands may exhibit anti-angiogenic properties [104], suggesting that in some types of cancer, CXCR2 ligands act as anti-angiogenic agents, whereas in others, they act as pro-angiogenic agents. Another explanation may involve the participation of CXCR2 ligands in an anti-tumor process that inhibits angiogenesis. Although CXCR2 ligand expression may not increase angiogenesis, the inhibition of angiogenesis may cause an increase in CXCR2 ligand expression.

In almost all types of cancer, two significant correlations occurred simultaneously. If there was a negative correlation between the expression of a particular CXCR2 ligand and the count of endothelial cells, there was also a negative correlation between the expression of the same CXCR2 ligand and the count of M2 macrophages.

Monocytes recruited to the tumor niche differentiate into M1 or M2 macrophages. M2 macrophages are cells that participate in cancer processes by secreting vascular endothelial growth factor (VEGF) and many other pro-angiogenic factors [105,106], making them pro-angiogenic, unlike M1 macrophages [105,107]. M1 macrophages show greater CXCR2 ligand expression compared to M2 macrophages [108]. Thus, CXCR2 ligands may participate in cancer processes that involve a decrease in the count of M2 macrophages and an increase in the count of M1 macrophages, leading to the inhibition of angiogenesis. As a result, there is a negative correlation between CXCR2 ligand expression and the count of endothelial cells.

In three types of cancer, the expression of any CXCR2 ligand was not correlated with the count of endothelial cells in the tumor mass. CXCR2 ligands are just one of several possible pro-angiogenic factors in the tumor microenvironment [107,109]. The lack of correlation with angiogenesis in a given type of cancer suggests that CXCR2 ligands may not be responsible for angiogenesis, but rather other pro-angiogenic factors.

PPBP may be the ligand most associated with angiogenesis across the largest number of cancer types (17 out of 30). In this regard, the correlation between PPBP expression and the count of endothelial cells in different types of cancer has been investigated. PPBP is a polypeptide that has its N-terminus removed after translation [10]. Depending on how much of the N-terminus is removed, CTAP-III, β-TG, or the shortest CXCL7/NAP-2 is produced. All three proteins have pro-angiogenic properties [53]. Nonetheless, the significance of these three proteins in tumorigenesis and tumor angiogenesis remains poorly understood [53,110]. They may play an important role in angiogenesis among CXCR2 ligands, as shown by the analysis of the correlation between PPBP expression and the count of endothelial cells. However, further research is required in this area.

### 3.7. Correlation between CXCR2 Ligand Expression and T_reg_ Cell Recruitment

The data obtained from the TIMER2.0 portal indicate that, in most cases, the level of CXCR2 ligand expression was not related to T_reg_ cell recruitment. However, there were frequent cases of positive correlation between the expression of CXCR2 ligands and the count of T_reg_ cells in the tumor. Single cases of negative correlation have also been demonstrated.

The common lack of correlation between CXCR2 ligand expression and T_reg_ cell recruitment suggests that the CXCR2 ligands are not associated with these cells. T_reg_ cells are mainly recruited to the tumor microenvironment by chemokines such as CCL1, CCL17, CCL22, CCL28, CXCL9, CXCL10, and CXCL11 [111]. In most cases, CXCR2 ligands have little impact compared to the aforementioned chemokines.

In 34 cases, there was a positive correlation between the expression of a given CXCR2 ligand and the count of T_reg_ in the tumor. This indicates that in some cases, CXCR2 ligands are associated with an increase in the count of T_reg_ cells in the tumor microenvironment. CXCR2 ligands can directly cause the recruitment of T_reg_ cells to the tumor microenvironment. The study on malignant pleural effusion in patients with non-small cell lung cancer has shown that CXCL1 is responsible for the recruitment of T_reg_ cells [112]. Similarly, under the influence of IL-6, the expression of CXCR1 in T_reg_ cells increases [113]. This allows these cells to be recruited by CXCR2 ligands that are also CXCR1 ligands, such as CXCL6 and CXCL8/IL-8. Another reason for the positive correlation between CXCR2 ligand expression and the count of T_reg_ cells is the production of CXCR2 ligands, particularly CXCL8/IL-8, by these cells [114]. CXCR2 ligands can also indirectly increase the count of T_reg_ cells in the tumor microenvironment. The aforementioned chemokines can also recruit naïve CD4^+^ T cells to the tumor microenvironment [115]. These cells, under the influence of the immunosuppressive tumor microenvironment, transform into T_reg_ cells. CXCR2 ligands can also contribute to the differentiation of naïve CD4^+^ T cells into T_reg_ cells [115]. CXCR2 ligands also enhance the immunosuppressive function of T_reg_ cells.

CXCR2 ligands can also increase the count of T_reg_ cells indirectly through neutrophils. CXCR2 ligands are chemotactic factors for neutrophils [1,4,116] and thus cause the recruitment of these cells to the tumor microenvironment. Neutrophils in the tumor secrete CCL17, which directly causes the recruitment of T_reg_ cells to the tumor microenvironment [117]. Another mechanism involves the participation of TGF-β. CXCL8 through CXCR1 can also increase the count of T_reg_ cells in the tumor microenvironment by increasing the expression of TGF-β [118].

Rarely, a negative correlation between CXCR2 ligand expression and T_reg_ cell count in the tumor was observed using the xCell algorithm. However, other algorithms estimating the count of T_reg_ cells did not confirm this negative correlation. If, in some types of tumors, a negative correlation exists between T_reg_ cell count and CXCR2 ligand expression, it can be explained by the association of CXCR2 ligands with pro-inflammatory responses. This has been thoroughly discussed in the subchapter on the correlation of CXCR2 ligands with the count and type of macrophages. In this study, CXCR2 ligand expression was frequently positively correlated with the count of M1 macrophages and negatively correlated with the count of M2 macrophages. M2 macrophages secrete IL-10, which increases the count of T_reg_ cells [119]. CXCR2 ligand expression negatively correlated with the count of M2 macrophages, and therefore, it may also be negatively correlated with the count of T_reg_ cells.

The obtained data demonstrate that the level of CXCR2 ligand expression is most commonly not associated with the count of T_reg_ cells in the tumor microenvironment, but in some cases, it may be related to an increase in the count of these cells in the tumor.

### 3.8. Correlation between CXCR2 Ligand Expression and Neutrophil Recruitment

The obtained data suggest that in nearly all types of cancer, the expression of CXCR2 ligands was positively correlated with the level of neutrophil recruitment to the tumor microenvironment. This is consistent with expectations, as neutrophils are immune system cells that express high levels of receptors for CXCR2 ligands [1,4,116]. Therefore, the increase in CXCR2 ligand levels leads to the infiltration of tissue by neutrophils [4]. Similarly, in tumors, an increase in CXCR2 ligand expression throughout or locally in a small area of the tumor leads to neutrophil recruitment to the tumor microenvironment. However, in some types of cancer, only certain CXCR2 ligands are positively correlated with neutrophil recruitment, suggesting that in these types of cancer only some CXCR2 ligands are responsible for neutrophil recruitment.

In some types of cancer, the expression of none of the CXCR2 ligands significantly correlated with the count of neutrophils. These chemokines are not the only chemotactic factors for neutrophils responsible for their recruitment to the tumor microenvironment. An example of another factor with such properties is leukotriene B_4_ (LTB_4_) [120], a lipid mediator that is formed from arachidonic acid via the 5-lipoxygenase (5-LOX) pathway. The first enzyme of this pathway and the enzyme that regulates the entire pathway is 5-LOX/*ALOX5* [121]. Various factors in the tumor microenvironment cause neutrophil recruitment to the tumor microenvironment, but only some of them are responsible for this process in a given type of cancer. These factors vary between different types of cancer.

### 3.9. Correlation between CXCR2 Ligand Expression and the Recruitment of MDSCs

Depending on the type of cancer, the expression of CXCR2 ligands can be positively or negatively correlated with the count of MDSCs in the tumor. There may also often be no significant correlation between the studied chemokines and the count of MDSCs. However, CXCL2 expression was most commonly negatively correlated with the count of MDSCs. This highlights the diversity of the cancer processes involving CXCR2 ligands, which are dependent on the cancer type.

The positive correlation between CXCR2 ligand expression and the count of MDSCs can be explained by the involvement of these chemokines in the recruitment of MDSCs to the tumor microenvironment. MDSCs can be divided into G-MDSCs and monocytic-myeloid-derived suppressor cells (M-MDSCs) [67]. CXCR2 ligands directly recruit G-MDSCs to the tumor microenvironment, which is related to their high CXCR2 expression [58,122,123,124]. In contrast, M-MDSCs have much lower CXCR2 expression and are not recruited to the tumor microenvironment by CXCR2 ligands, but instead by CCL2 [7,122].

Nevertheless, CXCR2 ligands may indirectly increase the count of M-MDSCs in the tumor. Tumors often show increased CXCR2 ligand expression relative to healthy tissue, leading to elevated levels of CXCR2 ligands in the blood. CXCR2 ligands can cause the expansion of M-MDSCs in the bone marrow, which is associated with their effects on granulocyte and macrophage progenitor cells (GMP) [7,8]. The increase in the count of M-MDSCs in the bone marrow leads to an increase in their count in the blood, which in turn increases the recruitment of M-MDSCs to the tumor microenvironment by other factors. Additionally, MDSCs can be a source of CXCR2 ligands in the tumor microenvironment [125], contributing to the positive correlation between CXCR2 ligand expression and the count of MDSCs in the tumor.

Not all types of tumors showed a positive correlation between the expression of CXCR2 ligands and the count of MDSCs. In 10 out of 30 analyzed types of tumors, the expression of at least one CXCR2 ligand was negatively correlated with the count of MDSCs, in the absence of any positively correlated ligands. This may be related to the involvement of CXCR2 ligands in tumor processes indirectly associated with MDSCs.

Overall, the processes in the tumor microenvironment can be divided into pro-tumor and anti-tumor processes. MDSCs are cells involved in pro-tumor processes, as they cause tumor immune evasion [67]. M2 macrophages are also cells that contribute to this effect [71]. Conversely, anti-tumor reactions are carried out by M1 macrophages [71] and DCs [64,65], which mutually exclude each other. In this study, we have shown that the expression of CXCR2 ligands was often positively correlated with DCs and M1 macrophages, while negatively correlated with MDSCs and M2 macrophages, consistent with previous experimental studies. CXCR2 ligands promote the migration of DCs and, thus, may exert anti-tumor effects [126]. M1 macrophages secrete CXCR2 ligands [108], which can lead to a positive correlation between the count of these cells and the level of CXCR2 ligand expression in the tumor. As M1 macrophages have an anti-tumor and pro-inflammatory character, which is the opposite of the function of MDSCs, in some cases, the expression of CXCR2 ligands may be negatively correlated with the count of MDSCs in the tumor microenvironment.

Moreover, the expression of some CXCR2 ligands was positively correlated in some tumor types, while in others, it was negatively correlated with the count of MDSCs in the tumor. This was observed in six types of tumors. It is likely that individual CXCR2 ligands are involved in the two types of processes described in this discussion, which lead to either positive or negative correlation. Another possible cause may be the interaction between individual CXCR2 ligands. They form homodimers and heterodimers, which activate CXCR2 [35,36,37]. One CXCR2 ligand may completely alter the function of other CXCR2 ligands, although this property is poorly understood.

### 3.10. Correlation between CXCR2 Ligand Expression and CD8^+^ T Cell Infiltration

In the majority of tumors, the expression level of CXCR2 ligands negatively correlated with the count of CD8^+^ T cells present in the tumor. This correlation may be attributed to the functions of CXCR2 ligands in the tumor microenvironment, specifically in promoting tumor immune evasion. CXCR2 ligands increase the count of G-MDSCs in the tumor by recruiting these cells to the tumor niche [58,122,123,124]. CXCR2 ligands also increase the count of T_reg_ cells in the tumor [112,115,127], and enhance the immunosuppressive properties of these cells [115].

Another property of CXCR2 ligands is their involvement in the recruitment of neutrophils to the tumor niche [128]. These cells are transformed into immunosuppressive cells under the influence of factors in the tumor microenvironment. CXCR2 ligands also increase the expression of PD-L1, particularly in macrophages [128]. These processes hinder the anti-tumor response of the immune system, leading to a reduction in the count of CD8^+^ T cells in the tumor and CD8^+^ T cell exhaustion. Therefore, experimental evidence has shown that CXCR2 ligands reduce the count of CD8^+^ T cells in the tumor [58,128,129]. This is consistent with the obtained results of the negative correlation between CXCR2 ligand expression and the count of CD8^+^ T cells in tumors of many types.

In individual types of tumors, only some of the CXCR2 ligands were negatively correlated with the count of the aforementioned cells. This suggests that in a given type of tumor, only some CXCR2 ligands are associated with a reduction in the count of CD8^+^ T cells. Additionally, the involvement of specific CXCR2 ligands in this process was dependent on the type of tumor.

In a few cases, the expression level of certain CXCR2 ligands positively correlated with the count of CD8^+^ T cells in the tumor. This was observed for CXCL1 and CXCL6 in breast invasive carcinoma, PPBP in kidney chromophobe, CXCL3 in kidney renal papillary cell carcinoma, CXCL6 in pancreatic adenocarcinoma, CXCL6 in thyroid carcinoma, and CXCL3 in thymoma. In almost all of these types of tumors, the expression of the aforementioned CXCR2 ligands was also positively correlated with DC, which, similar to CD8^+^ T cells, are anti-tumor cells [64,65]. Furthermore, in kidney renal papillary cell carcinoma and thyroid carcinoma, the expression of CXCR2 ligands negatively correlated with the count of MDSCs, which are pro-tumor cells that decrease and inhibit the function of CD8^+^ T cells [58,67]. These CXCR2 ligands likely participate in an anti-tumor process in these types of tumors, which involves the recruitment of anti-tumor cells.

### 3.11. Correlation between CXCR2 Ligand Expression and NK Cell Infiltration

In the vast majority of tumors, the level of CXCR2 ligand expression was not significantly correlated with the level of tumor infiltration by NK cells. This is consistent with current scientific knowledge, as NK cells have low expression of CXCR1 and CXCR2 [1,130] and are, therefore, not recruited to the tumor microenvironment by CXCR2 ligands. Adoptive cell therapy is being developed to increase CXCR2 expression in NK cells [131,132]. These modified cells efficiently infiltrate the tumor where high levels of CXCR2 ligands are expressed.

In rare cases, CXCR2 ligand expression negatively correlated with the level of NK cell infiltration in the tumor. However, other algorithms available on the TIMER2.0 portal often do not confirm this. CXCR2 ligands induce the recruitment of immunosuppressive cells, including MDSCs [133], leading to the inhibition of the anticancer functions of NK cells. Furthermore, CXCR2 ligands, particularly CXCL8/IL-8, can inhibit the cytotoxic function of NK cells [134]. The involvement of CXCR2 ligands in immunosuppressive processes is associated with a decrease in the count of NK cells in the tumor microenvironment [133]. This explains the cases of negative correlation between the level of CXCR2 ligand expression and the infiltration of the tumor by NK cells. However, these processes may be rare in tumors and occur only in specific types of tumors and specific CXCR2 ligands.

In two types of tumors, the expression of certain CXCR2 ligands positively correlated with the level of infiltration by NK cells. These tumors are lower grade glioma (in the case of CXCL3 and CXCL5) and diffuse large B-cell lymphoma (in the case of CXCL2 and CXCL3). Although there is no research available regarding their association with NK cells in these tumors, it can be assumed that certain CXCR2 ligands participate in an anticancer process that stimulates the immune response and the infiltration of the tumor by NK cells.

The detailed results are shown in Appendix A of the supplement, which summarizes all the discussed results for lower grade glioma, a type of tumor in which CXCR2 ligands negatively correlated with the count of MDSCs, cells that inhibit the anti-tumor response [67]. Additionally, CXCR2 ligands positively correlated with the count of NK cells which indicates anticancer properties. The results suggest that CXCR2 ligands may be linked to an enhancement of the immune response in this type of tumor.

In the case of diffuse large B-cell lymphoma, CXCR2 ligands positively correlated with NK cells are also positively correlated with a proliferation marker. Diffuse large B-cell lymphoma is a tumor located in the lymph node. The lymph node has low but not zero expression of CXCL2 and CXCL3 [9]. CXCR2 ligands, particularly CXCL1, are significant in NK cell development and immune surveillance [135]. Furthermore, NK cells are activated in the lymph nodes [136]. Based on this information, it is necessary to investigate whether diffuse large B-cell lymphoma has a pro-tumor pathway that acts on NK cells through CXCR2 ligands.

### 3.12. Correlation between CXCR2 Ligand Expression and Infiltration by Dendritic Cells (DCs)

The expression of CXCR2 ligands was often positively correlated with the count of conventional (formerly called myeloid) DCs, with occasional cases of negative correlation. In the case of plasmacytoid DCs, there were similar frequencies of positive and negative correlations between the count of these cells and the level of CXCR2 ligand expression, although in a much larger number of cases, no significant correlation was found.

The importance of CXCR2 ligands for DCs has been poorly studied. Studies on GM-CSF-induced DCs from PBMCs with IL-4 or IL-13 exposure have shown that such DCs express CXCR1 and CXCR2 [137,138,139,140], although CXCL2 and CXCL8/IL-8 do not cause the migration of these cells [138,139]. However, further studies on monocyte-derived DCs have shown that CXCL8/IL-8 can cause the migration of these cells [140,141], which may be recruited to the tumor microenvironment by CXCR2 ligands [140,141]. It should be noted that monocyte-derived DCs are a distinct subset of DCs that differ from conventional and plasmacytoid DCs [65]. Additionally, CXCR2 ligands, particularly CXCL8/IL-8, can also cause the migration of spleen DCs [126]. Therefore, CXCR2 ligands, whose expression is often elevated in tumors, directly cause the infiltration of the tumor microenvironment by DCs [126], especially conventional type 2 DCs [127], which may explain the positive correlation between CXCR2 ligand expression and the count of conventional (myeloid) DCs.

There was also often a negative correlation between the level of CXCR2 ligand expression and the count of DCs, more frequently observed in plasmacytoid DCs than in conventional DCs. This correlation may be related to the immunosuppressive functions of CXCR2 ligands in the tumor microenvironment. CXCR2 ligands can increase the count of MDSCs [7,8,58,123,124] and T_reg_ cells [112,115] in the tumor microenvironment, cells which are responsible for tumor immune evasion [67,68] and can indirectly contribute to reducing the infiltration by DCs and other cytotoxic cells, such as NK cells and CD8^+^ T cells. This may explain the negative correlation between CXCR2 ligand expression and the count of DCs, NK cells, and CD8^+^ T cells in analyzing the same tumor and the same CXCR2 ligand.

The relationship between CXCR2 ligands and infiltration of the tumor microenvironment by DCs has not been well understood. This study showed that the level of CXCR2 expression was often positively correlated with infiltration of the tumor by DCs, particularly with conventional DCs. Therefore, precise studies are required to understand the impact of CXCR2 ligands on these cells in the context of tumor processes. The results of such research will contribute to a better understanding of the anti-tumor processes in patients and may lead to the development of new methods in cancer immunotherapy.

### 3.13. Correlation between CXCR2 Ligand Expression and Macrophage Infiltration and Polarization

The expression of CXCR2 ligands positively correlated with the count of macrophages in tumors, with few cases showing a negative correlation. CXCR2 ligand expression was usually positively correlated with the count of M1 macrophages and negatively correlated with the count of M2 macrophages. Additionally, CXCL2 expression was often positively correlated with the count of M2 macrophages in tumors.

CXCR2 ligands can act on monocytes, which can be classified into three subsets: classical, intermediate, and nonclassical. Classical monocytes express CXCR2 and can be recruited by CXCR2 ligands [142]. However, classical monocytes are more immunosuppressive than nonclassical monocytes, as they secrete more CCL2 and CCL5 but less IL-1β and TNF-α in response to LPS [142]. Through this mechanism, CXCR2 ligands can recruit monocytes and increase the count of macrophages, particularly M2 macrophages. However, monocytes that differentiate into macrophages in the tumor microenvironment are mainly recruited by CCR2 ligands, such as CCL2/MCP-1 [143]. Therefore, the direct involvement of CXCR2 ligands in monocyte recruitment and macrophage increase is minimal. Nevertheless, CXCR2 ligands can indirectly cause monocyte recruitment and, thereby, increase the count of macrophages in the tumor microenvironment. CXCR2 ligands directly recruit neutrophils [1,4,116], which secrete CCL2 and cause monocyte recruitment to the tumor niche [117]. This may explain the positive correlation between CXCR2 ligand expression level and the count of macrophages in various types of tumors.

Experiments on pancreatic ductal adenocarcinoma have shown that the recruitment of neutrophils and monocytes is in equilibrium in this type of cancer [55]. Decreased recruitment of one type of cell results in increased recruitment of the other. However, the correlation analysis of CXCR2 ligands with different cells in pancreatic adenocarcinoma in this study showed that the expression of most CXCR2 ligands was not significantly correlated with the count of macrophages in this type of cancer. The expression of CXCR2 ligands positively correlated with the count of neutrophils, but only CXCL5 showed a simultaneous positive correlation with the count of neutrophils and macrophages. The analysis of other types of tumors in this study did not show any opposite correlation between the count of neutrophils and macrophages and the level of CXCR2 ligand expression.

M1 macrophages are characterized by higher expression of CXCR2 ligands compared to M2 macrophages [108]. This explains the positive correlation between the expression of CXCR2 ligands and the number of M1 macrophages in the tumor, as well as the negative correlation with the number of M2 macrophages.

M1 macrophages are anti-tumor macrophages that inhibit tumor growth [71]. In contrast, M2 macrophages are pro-tumor cells. Therefore, CXCR2 ligands can be markers of the anti-tumor response of the immune system. These processes lead to the infiltration of the tumor by DC, NK cells, and CD8^+^ T cells. Therefore, there may be a positive correlation between the level of expression of CXCR2 ligands in tumors and the level of infiltration of the tumor by DC, NK cells, and CD8^+^ T cells.

The expression of CXCR2 ligands negatively correlates with the count of M1 macrophages, which may be caused by a negative correlation between the expression of the analyzed chemokines and the total count of macrophages. Another explanation could be the increase in the count of immunosuppressive cells, such as MDSCs and T_reg_ cells, through CXCR2 ligands. However, by analyzing the tables in the supplement, it can be inferred that the negative correlation with the count of M1 macrophages is associated with a negative correlation with the total count of macrophages.

In some cases, there was a positive correlation between the expression of CXCR2 ligands and the count of M2 macrophages. This may be due to the recruitment of classical monocytes by CXCR2 ligands [142]. These cells differentiate into macrophages that are more polarized towards the M2 phenotype compared to non-classical monocytes. Another reason for the positive correlation may be the enhancing effect of CXCR2 ligands on macrophage polarization [144]. A decrease in the expression of CXCR2 ligands or blocking the CXCR2 receptor can lead to the repolarization of these macrophages into M1 macrophages. Therefore, this can lead to a positive correlation between the expression of CXCR2 ligands and the count of M2 macrophages. However, by analyzing the tables in the supplement, there was no simultaneous positive correlation with the count of M2 macrophages and a negative correlation between the count of M1 macrophages and the expression of CXCR2 ligands.

The expression of CXCL2 was often negatively correlated with the count of MDSCs and positively correlated with the count of M2 macrophages. In the case of this chemokine, in almost every type of tumor, if there was a positive correlation between CXCL2 expression and the count of M2 macrophages, there was also a negative correlation between the expression of this chemokine and the count of MDSCs. This may suggest a compensatory mechanism for the effect of CXCL2 on M2 macrophages and MDSCs.

### 3.14. Expression of CXCR2 Ligands and Prognosis

The differences in survival between patients with higher and lower expression of a particular gene indicate that the gene plays a significant role in either anti-tumor processes (in the case of better prognosis with higher expression) or pro-tumor processes (in the case of worse prognosis with higher expression). However, the impact on survival alone says very little about the mechanisms by which the gene participates in tumor processes. The analyses conducted in this study demonstrated the correlation of CXCR2 ligands with important tumor processes such as proliferation, angiogenesis, EMT, and lymph node metastasis, the recruitment of various cells to the tumor microenvironment including T_reg_ cells, neutrophils, MDSCs, CD8^+^ T cells, NK cells, DCs, and macrophages, and an analysis of macrophage polarization. Thus, it is possible to link the impact of CXCR2 ligands on survival with the tumor mechanisms in which the analyzed chemokines participate. To facilitate this, tables have been prepared showing the participation of CXCR2 ligands in tumor processes in specific tumors. These tables are included in the Appendix A to this study (Appendix A). By examining these tables, it is possible to understand the mechanisms in which CXCR2 ligands participate and their impact on survival.

One example of this is the association of higher CXCL2 expression with worse prognosis in patients with lung squamous cell carcinoma. According to Appendix A, CXCL2 in this tumor positively correlated with pro-tumor processes such as EMT, the recruitment of T_reg_ cells, neutrophils, and angiogenesis. CXCL2 expression was also positively correlated with the count of M2 macrophages. Due to the participation of CXCL2 in these processes, it may result in worse prognosis in patients with lung squamous cell carcinoma.

Another example of analysis is CXCL2 and CXCL5 in brain lower grade glioma. All information on this type of tumor has been grouped in Appendix A. The higher expression of these two chemokines was associated with better prognosis. CXCL2 and CXCL5 negatively correlated with proliferation, angiogenesis, recruitment of MDSCs, and positively correlated with tumor infiltration by NK cells. This suggests a potential mechanism of participation of these two chemokines, resulting in improved prognosis in cases of higher expression of CXCL2 or CXCL5.

However, this study includes only bioinformatic analyses. Although it provides significant insights, it only shows a correlation between the level of expression of a particular CXCR2 ligand and a particular process in a given tumor. Therefore, it may be considered a preliminary study whose conclusions should be confirmed experimentally to demonstrate the real impact of CXCR2 ligands on specific tumor processes.

### 3.15. Summary

All seven CXCR2 ligands exhibited remarkably similar characteristics in the realm of cancer processes. However, a deeper analysis of these individual ligands across 31 different tumor types revealed the emergence of distinct and novel properties.

## 4. Materials and Methods

### 4.1. GEPIA

This study used bioinformatic tools to analyze CXCR2 ligands in cancer diseases. One of these tools was the Gene Expression Profiling Interactive Analysis (GEPIA) database (http://gepia.cancer-pku.cn/detail.php (accessed on 30 November 2022)) [145]. This database provides analysis of raw mRNA gene expression data from the Cancer Genome Atlas (TCGA) [146], which includes almost 10,000 samples from 33 types of tumors (Table 23). Additionally, these data were enriched with gene expression analysis from over 8000 normal healthy tissue samples from the Genotype-Tissue Expression (GTEx) database [147,148]. GEPIA allows for the analysis of differential gene expression between any of the 33 types of tumors and healthy tissue [145]. In this study, 31 out of 33 types of tumors were analyzed, as the GEPIA database does not provide data for uveal melanoma and mesothelioma.

Differences in CXCR2 ligand expression between tumors and healthy tissue were analyzed. The study also investigated the association between CXCR2 ligand expression and patient overall survival. The quartiles with the highest and lowest expression levels of each CXCR2 ligand were compared to determine which ligands had anti- or pro-tumor properties. We also examined the correlation between CXCR2 ligands and various markers of tumorigenesis, including proliferation. The study focused on the correlation between each CXCR2 ligand and Ki-67/*MKI67*, a marker of proliferation found only in proliferating cells, to assess their impact on proliferation [149,150]. We also analyzed the correlation between CXCR2 ligands and vimentin/VIM, N-cadherin/CDH2, and E-cadherin/CDH1, markers which are associated with the mesenchymal and epithelial phenotypes [151,152,153].

Statistical analyses of differences in the expression of CXCR2 ligands between tumors and healthy tissues were conducted using the GEPIA platform. Similarly, the statistical analysis of the results of the impact of CXCR2 ligand expression on patient prognosis was carried out using the GEPIA. A log-rank test, also known as the Mantel–Cox test, was employed to evaluate the statistical significance of patient prognosis. Results with *p* < 0.05 were considered statistically significant. Spearman’s rank correlation coefficient was employed to evaluate the statistical significance of correlations, with results at *p* < 0.05 considered statistically significant.

To perform the analysis on the GEPIA portal, the following data were entered for comparing healthy tissue with tumors:-Cancer type designation;-Targeted gene;-Analysis using GTEx data;-|Log2FC| Cutoff: 1;-*p*-value Cutoff: 0.05.

To analyze the impact of CXCR2 ligand expression levels on prognosis, the following data were entered or selected:-Cancer type designation;-Targeted gene;-Overall survival as the prognosis;-Group cutoff: quartile. If the group size was too small for analysis, median was selected instead.

### 4.2. UALCAN

Another portal used in the analysis of CXCR2 ligand expression in cancer was the University of Alabama at Birmingham Cancer data analysis (UALCAN) portal (http://ualcan.path.uab.edu (accessed on 30 November 2022)) [154,155]. The raw data in UALCAN were derived from TCGA [146]. This portal provided an analysis of the expression of over 20,000 proteins in 33 different types of cancer. The UALCAN does not use gene expression data analysis in healthy tissue from the GTEx, resulting in less accurate differences between gene expression in tumors and normal tissue. Therefore, the expression of CXCR2 ligands was analyzed using the GEPIA.

However, the UALCAN allows for a greater variety of gene data analyses, and, thus, it was used to examine the association between CXCR2 ligands and lymph node metastasis status. The statistical analysis of the obtained results was conducted using the UALCAN, with a significance threshold of *p* < 0.05.

To analyze the correlation between CXCR2 ligand expression levels and lymph node metastasis status, the following data were entered:-TCGA database was used;-Cancer type designation;-Targeted gene.

### 4.3. TIMER2.0

The tumor microenvironment was composed not only of cancer cells but also of tumor-associated cells that participate in tumor processes. Therefore, an essential research direction for any factor is to examine its correlation with tumor-associated cells. To investigate the correlation between the expression of the analyzed CXCR2 ligands and the level of tumor-associated cell recruitment, the Tumor Immune Estimation Resource (TIMER)2.0 portal was used (http://timer.cistrome.org/ (accessed on 30 November 2022)) [156,157,158]. The TIMER2.0 allowed for the analysis of the correlation between the level of tumor infiltration by different types of immune cells, endothelial cells, and fibroblasts with the level of expression of the selected gene or mutation of a given gene in different types of tumors. The TIMER2.0 uses raw data from the TCGA [146], which have been analyzed by various algorithms estimating the count of tumor-associated cells. One such algorithm is the TIMER algorithm [156], which only allows for the analysis of six types of cells. This was far too few for the needs of this study. Therefore, the xCell algorithm [159] was used, which allows for the analysis of 33 types of cells in the tumor microenvironment and was, therefore, the best choice of all available algorithms on the TIMER2.0 portal [160]. However, the method for estimating the count of cells in the tumor microenvironment was indirect and based on the level of expression of gene signatures of specific cells. Individual algorithms differed from each other. Therefore, in the tables presented in this study, information was provided on whether the result was consistent with other algorithms available on the TIMER2.0 website, such as TIMER [156,157], MCP-counter [161], CIBERSORT [162], QUANTISEQ [163], and EPIC [164]. Since there were no available data on MDSC recruitment analysis performed by the xCell algorithm on the TIMER2.0 website, the Tumor Immune Dysfunction and Exclusion (TIDE) algorithm was used to analyze the correlation between the expression of CXCR2 ligands and the level of MDSC recruitment [165].

The statistical significance of the correlation was estimated on the TIMER2.0 platform using Spearman’s Rank correlation coefficient. Results with *p* < 0.05 were considered statistically significant.

To analyze the correlation between different cell types and expression levels, the following data were entered:-Targeted gene;-Type of analyzed cells.

### 4.4. cBioPortal

To investigate the potential involvement of CXCR2 ligands in tumor-associated processes, changes in their expression levels in tumor tissues relative to healthy tissues may be attributed to gene amplification or deletion. Thus, the frequency of CXCR2 ligand mutations was evaluated in various cancers using the cBioPortal (http://www.cbioportal.org/ (accessed on 5 December 2022)) [166,167]. This platform offers access to 363 studies, comprising nearly 183,000 tumor samples, and enables the determination of mutation types in specific genes across selected cancer types.

To assess the mutation frequency in CXCR2 ligand genes in 31 different cancer types, the TCGA PanCancer Atlas studies were selected [168,169,170,171,172,173]. These studies provide the count of gene amplifications and the specific mutation types in CXCR2 ligand genes. A total of 31 studies were utilized, comprising 10,726 cases of cancer.

### 4.5. BLAST

The similarity of CXCR2 ligand proteins and transcripts was analyzed using the basic local alignment search tool (BLAST) [174,175,176,177]. The National Center for Biotechnology Information (NCBI) website was used for this purpose, which provides the BLAST sequence analysis tool (https://blast.ncbi.nlm.nih.gov/blast/Blast.cgi (accessed on 12 December 2022)) [178]. The protein sequences of CXCR2 ligands were obtained from the Protein Database located on the NCBI portal (https://www.ncbi.nlm.nih.gov/protein (accessed on 12 December 2022)), which contains amino acid sequences from various databases, such as GenBank. The following CXCR2 ligand protein sequences were selected [179,180]:CXCL1—GenBank: EAX05693.1;CXCL2—GenBank: EAX05701.1;CXCL3—GenBank: EAX05698.1;CXCL5—GenBank: EAX05696.1;CXCL6—GenBank: AAH13744.1;CXCL7—GenBank: AAH28217.1;CXCL8—GenBank: AAH13615.1.

The transcript sequences of CXCR2 ligands were obtained from the Nucleotide Database at the NCBI portal (https://www.ncbi.nlm.nih.gov/nuccore (accessed on 12 December 2022)), which contains mRNA nucleotide sequences from various databases. The used sequences were taken from Genome Reference Consortium Human Build 38 patch release 14 (GRCh38.p14) [181,182], the human reference genome. The following CXCR2 ligand transcript sequences were selected:*CXCL1*—NCBI: NM_001511.4;*CXCL2*—NCBI: NM_002089.4;*CXCL3*—NCBI: NM_002090.3;*CXCL5*—NCBI: NM_002994.5;*CXCL6*—NCBI: NM_002993.4;*PPBP*—NCBI: NM_002704.3;*CXCL8*—NCBI: NM_000584.4.

### 4.6. Tfsitescan

One of the most important methods of gene expression regulation is transcriptional regulation through the binding of various transcription factors to the promoter. To analyze potential transcription factors regulating the expression of CXCR2 ligands, the Tfsitescan tool at the IFTI-MIRAGE website was used (http://www.ifti.org/cgi-bin/ifti/Tfsitescan.pl (accessed on 12 December 2022)) [183]. The Tfsitescan allows for the identification of potential transcription factors that may bind to a given nucleotide sequence. It searches for sequences to which transcription factors bind in promoters or sequences significant in gene expression regulation that have been experimentally demonstrated.

To identify potential transcription factors, the following data were entered:-1500 bp sequences upstream of the transcription start site of CXCR2 ligand genes;-“IFTI Tfsites” option was selected;-Tfsites query parameters: mammalian sites.

To use this tool, a nucleotide sequence of the promoter was required. It was possible to analyze nucleotide sequences of a maximum length of nearly 1500 bp. To obtain the appropriate promoter sequences for CXCR2 ligand genes, the Gene Database located on the NCBI portal (https://www.ncbi.nlm.nih.gov/gene/ (accessed on 12 December 2022)) was used. The following identification numbers of CXCR2 ligand genes were used to search for appropriate sequences:*CXCL1*—Gene ID: 2919;*CXCL2*—Gene ID: 2920;*CXCL3*—Gene ID: 2921;*CXCL5*—Gene ID: 6374;*CXCL6*—Gene ID: 6372;*PPBP*—Gene ID: 5473;*CXCL8*—Gene ID: 3576.

Nucleotide sequences were selected from the transcription start point to 1500 bp upstream of this point. The obtained sequences were from GRCh38.p14 [181,182], which is the human reference genome. When analyzing the results obtained using the Tfsitescan, we took into account proteins that bind to the introduced sequence in probability was <0.05.

### 4.7. miRDB

To predict possible microRNAs regulating the expression of CXCR2 ligands, the miRDB portal was used (https://mirdb.org (accessed on 9 December 2022)) [184,185]. This portal allows for the prediction of potential microRNAs regulating the expression of a given gene in 5 species, including humans, mice, and rats. The algorithm used by the miRDB portal ranks potential microRNAs by target score, ranging from 1 to 100 [184]. According to the miRDB help tab, results with a target score above 80 are likely to occur in reality. Therefore, in this study, we considered potential microRNAs with a target score above 80.

The miRDB portal was utilized to search for potential miRNAs that regulate the expression of CXCR2 ligands. The search parameters specified the human miRNAs to be considered. Additionally, the desired genes (*CXCL1*, *CXCL2*, *CXCL3*, *CXCL5*, *CXCL6*, *PPBP*, *CXCL8*) were included in the search options, adhering to the miRDB help tab which required a target above 80.

## 5. Conclusions

Based on the results of this study, the following conclusions can be drawn:The level of expression of CXCR2 ligands in the tumor compared to healthy tissue depends on the type of cancer, but is often elevated, particularly in the case of CXCL1 and CXCL8/IL-8. The expression of CXCL2 is often decreased in the tumor compared to healthy tissue.The regulation of the expression of each CXCR2 ligand is different; therefore, each analyzed chemokine may have a different function in cancer processes.Depending on the type of cancer, different CXCR2 ligands are positively or negatively correlated with intense proliferation. PPBP/CXCL7 may have anti-proliferative properties.The level of expression of CXCR2 ligands is associated with cancer cell migration in the EMT process.CXCR2 ligands are often associated with a better lymph node metastasis status. In rare instances, the increased expression of CXCR2 ligands in the tumor is linked to a poorer lymph node metastasis status.Depending on the type of cancer, different CXCR2 ligands are associated with intense angiogenesis. In some cancers, CXCR2 ligands may be associated with the inhibition of angiogenesis.CXCR2 ligands are responsible for recruiting neutrophils to the tumor microenvironment, although not all CXCR2 ligands may be responsible for this process in a given tumor.In individual cases, CXCR2 ligands are associated with the count of T_reg_ cells in the tumor. Usually, the expression of CXCR2 ligands is not associated with the count of T_reg_ cells in the tumor microenvironment.Depending on the type of cancer, CXCR2 ligand expression is positively or negatively correlated with the count of MDSCs in the tumor. However, the expression of CXCL2 is most commonly negatively correlated with the count of these cells.CXCR2 ligands are often associated with a decrease in the count of CD8^+^ T cells in the tumor. In sporadic cases, the expression of CXCR2 ligands is associated with an increase in tumor infiltration by CD8^+^ T cells.CXCR2 ligand expression is not associated with the count of NK cells. In rare cases, there is a negative correlation associated with the immunosuppressive properties of these chemokines.The expression of CXCR2 ligands is associated with the count of conventional DCs in the tumor. CXCR2 ligands probably cause infiltration of the tumor microenvironment by conventional DCs, which increases the patient’s anti-tumor response.CXCR2 ligands are associated with M1 macrophages, which are anti-tumor cells. Therefore, CXCR2 ligands may be a marker of the immune system’s anti-tumor response, including infiltration of the tumor microenvironment by DCs, NK cells, and CD8^+^ T cells. However, CXCL2 may be associated with M2 macrophages and pro-tumor reactions.Mutations in CXCR2 ligand genes are rare in cancer. If they do occur, it is most often the amplification of the entire gene cluster in which all CXCR2 ligand genes are located.The effect of CXCR2 ligands on prognosis depends on the type of cancer.

## Figures and Tables

**Figure 1 ijms-24-13287-f001:**
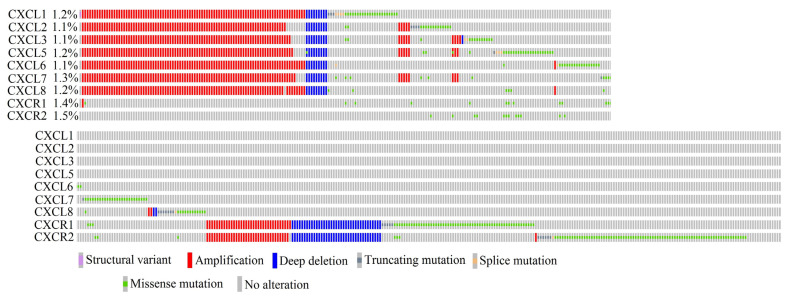
Cases of mutations in CXCR2 ligand genes and CXCR1 and CXCR2 receptors. Mutations are listed according to type and location in the corresponding gene, from CXCL1 to CXCL8, CXCR1, and CXCR2. Each case is represented as a rectangle with a different color or label. Vertically aligned rectangles are from the same patient but show different analyzed genes.

**Figure 2 ijms-24-13287-f002:**
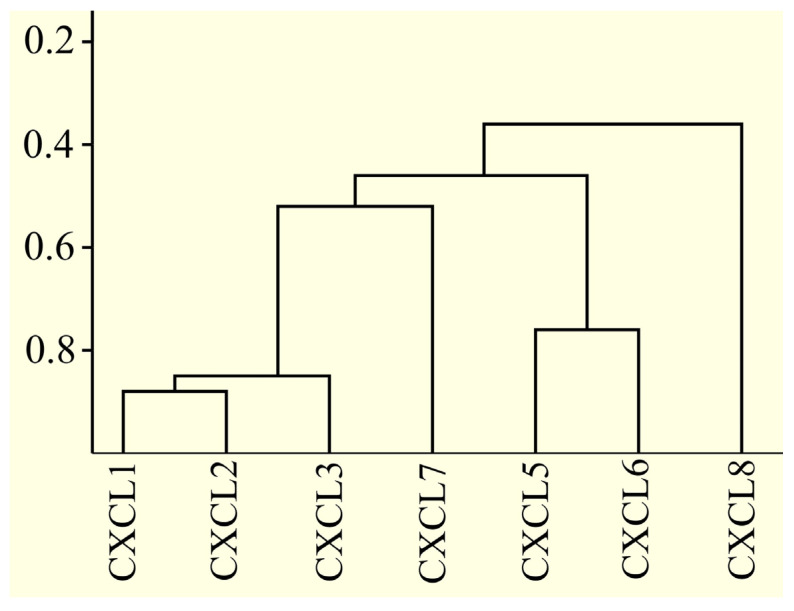
Sequence similarity of CXCR2 ligand proteins. Only some CXCR2 ligand proteins exhibited high sequence similarity. The dendrogram was constructed based on the BLAST similarity analysis of the entire amino acid sequence of each of the two CXCR2 ligand proteins.

**Figure 3 ijms-24-13287-f003:**
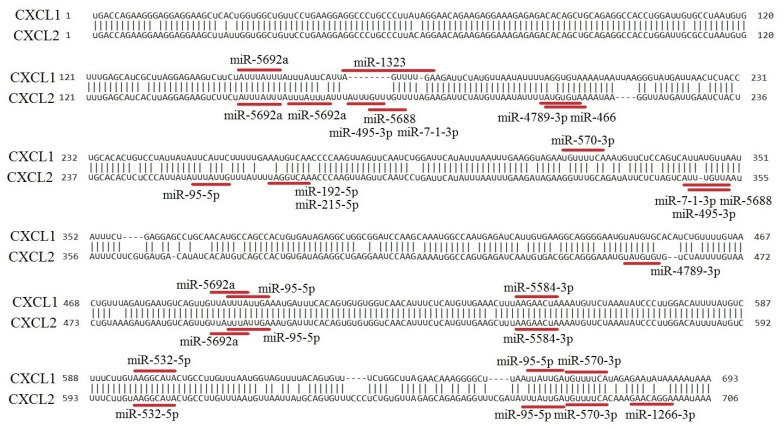
Comparison of the 3′-UTR sequences of CXCL1 and CXCL2, indicating the most likely microRNA binding sites.

**Table 1 ijms-24-13287-t001:** Location of the highest expression of each CXCR2 ligand.

CXCR2 Ligand	Location of the Highest Expression
*CXCL1*	Appendix, bone marrow, gall bladder, small intestine, urinary bladder
*CXCL2*	Appendix, bone marrow, gall bladder, liver, lung
*CXCL3*	Appendix, bone marrow, colon, gall bladder, liver, lung, stomach, urinary bladder
*CXCL5*	Appendix, gall bladder, lung, lymph node, stomach, urinary bladder
*CXCL6*	Appendix, gall bladder, urinary bladder
*PPBP*	Bone marrow, spleen
*CXCL8*	Appendix, bone marrow, esophagus, gall bladder, liver, urinary bladder

**Table 2 ijms-24-13287-t002:** Differences in expression of CXCR2 ligands relative to healthy tissue in selected cancers.

Name of the Cancer	*CXCL1*	*CXCL2*	*CXCL3*	*CXCL5*	*CXCL6*	*PPBP*	*CXCL8*	*CXCR1*	*CXCR2*
Adrenocortical carcinoma	=	↓	=	=	=	=	=	=	=
Bladder urothelial carcinoma	=	=	=	=	=	=	=	=	↓
Breast invasive carcinoma	↓	↓	↓	=	=	=	=	=	=
Cervical squamous cell carcinoma and endocervical adenocarcinoma	↑	=	↑	=	=	=	↑	=	=
Cholangiocarcinoma	↑	↓	↑	↑	↑	=	↑	=	=
Colon adenocarcinoma	↑	↑	↑	↑	=	=	↑	=	=
Lymphoid neoplasm diffuse large B-cell lymphoma	=	=	=	=	=	↓	↓	↓	↓
Esophageal carcinoma	↑	↑	↑	↑	↑	=	↑	=	↓
Glioblastoma multiforme	=	↑	↑	=	=	=	↑	=	=
Head and neck squamous cell carcinoma	↑	=	=	=	=	=	↑	=	↓
Kidney chromophobe	↓	↓	=	=	=	=	↓	=	=
Kidney renal clear cell carcinoma	=	=	=	=	=	=	=	=	=
Kidney renal papillary cell carcinoma	↑	=	=	=	↑	=	↑	=	=
Acute myeloid leukemia	=	=	↓	=	=	↑	=	=	↑
Brain lower grade glioma	=	=	=	=	=	=	=	=	=
Liver hepatocellular carcinoma	=	↓	=	=	=	=	=	=	=
Lung adenocarcinoma	=	↓	↓	=	=	↓	=	↓	↓
Lung squamous cell carcinoma	=	↓	↓	↓	↑	↓	=	↓	↓
Ovarian serous cystadenocarcinoma	↑	=	=	=	=	=	↑	=	=
Pancreatic adenocarcinoma	↑	=	↑	↑	↑	=	↑	=	=
Pheochromocytoma and Paraganglioma	↓	=	=	=	=	=	↓	=	=
Prostate adenocarcinoma	=	↓	=	=	=	=	=	=	=
Rectum adenocarcinoma	↑	↑	↑	↑	=	↑	↑	=	=
Sarcoma	=	=	=	=	=	=	=	=	=
Skin cutaneous melanoma	↑	=	=	=	=	=	↑	=	↓
Stomach adenocarcinoma	↑	=	↑	↑	↑	=	↑	=	=
Testicular germ cell tumors	=	=	=	=	=	=	=	=	=
Thyroid carcinoma	=	=	=	=	=	=	=	=	=
Thymoma	↓	=	=	=	=	↓	↓	↓	↓
Uterine corpus endometrial carcinoma	↑	=	↑	=	=	=	↑	=	=
Uterine carcinosarcoma	↑	=	=	=	=	=	↑	=	=

↑, red background—expression higher in tumor than in healthy tissue; ↓, blue background—expression lower in tumor than in healthy tissue; =, gray background—expression in tumor does not differ from healthy tissue.

**Table 3 ijms-24-13287-t003:** Association of CXCR2 ligand expression levels with prognosis for patients with various cancers.

Name of the Cancer	*CXCL1*	*CXCL2*	*CXCL3*	*CXCL5*	*CXCL6*	*PPBP*	*CXCL8*	*CXCR1*	*CXCR2*
Adrenocortical carcinoma	=	=	=	=	N/A	↓ *p* = 0.061	↓	=	=
Bladder urothelial carcinoma	=	=	=	=	=	=	=	=	=
Breast invasive carcinoma	↑	↑	↑ *p* = 0.056	↑ *p* = 0.08	↑	N/A	=	=	=
Cervical squamous cell carcinoma and endocervical adenocarcinoma	↓	↓	↓	↓	↓ *p* = 0.056	↓ *p* = 0.088	↓	↓	=
Cholangiocarcinoma	=	=	=	=	=	=	=	=	=
Colon adenocarcinoma	=	↑	↑ *p* = 0.094	=	=	=	=	=	=
Lymphoid neoplasm diffuse large B-cell lymphoma	=	=	↓	=	↓ *p* = 0.061	=	=	=	=
Esophageal carcinoma	=	=	=	↓ *p* = 0.092	=	=	↓	=	=
Glioblastoma multiforme	↓	↓	=	↓	↓ 0.096	=	=	=	=
Head and neck squamous cell carcinoma	↓ 0.076	=	=	=	=	=	=	=	=
Kidney chromophobe	=	=	=	↓	=	=	↓	=	=
Kidney renal clear cell carcinoma	↓	↓	↓	↓	↓	=	↓	↑	↑
Kidney renal papillary cell carcinoma	=	↑ *p* = 0.072	=	↑ *p* = 0.078	=	=	=	=	=
Acute myeloid leukemia	↓	↓ *p* = 0.055	=	↓	=	↓ *p* = 0.099	=	=	=
Brain lower grade glioma	↓	↑	↑ *p* = 0.096	↑	=	=	=	↓ *p* = 0.075	↓
Liver hepatocellular carcinoma	↓	=	↓	↓	↓	=	↓	=	=
Lung adenocarcinoma	=	=	=	=	↓ *p* = 0.076	=	↓	=	=
Lung squamous cell carcinoma	=	↓	=	=	=	=	=	=	=
Ovarian serous cystadenocarcinoma	=	=	=	↑ *p* = 0.083	=	=	=	=	=
Pancreatic adenocarcinoma	=	=	=	↓	=	=	=	=	=
Pheochromocytoma and paraganglioma	=	=	=	=	N/A	=	=	=	=
Prostate adenocarcinoma	=	=	=	=	=	N/A	=	=	=
Rectum adenocarcinoma	=	=	=	=	=	=	=	=	=
Sarcoma	=	↑	=	=	=	↓ *p* = 0.091	=	=	=
Skin cutaneous melanoma	=	=	=	=	=	↓ *p* = 0.064	=	=	↓
Stomach adenocarcinoma	=	=	↑ *p* = 0.055	=	=	=	=	=	=
Testicular germ cell tumors	=	=	=	=	=	N/A	=	=	=
Thyroid carcinoma	=	=	=	=	=	↓	=	=	=
Thymoma	=	=	=	=	=	=	=	=	=
Uterine corpus endometrial carcinoma	=	=	=	=	=	=	=	=	=
Uterine carcinosarcoma	=	=	=	=	=	=	=	=	=

↓, red background—higher CXCR2 ligand expression is associated with poorer prognosis; ↓, orange background—trend toward worse prognosis (*p* < 0.10) with higher CXCR2 ligand expression; ↑, blue background—higher CXCR2 ligand expression is associated with better prognosis; ↑, light blue background—tendency for better prognosis (*p* < 0.10) with higher CXCR2 ligand expression; =, gray background—expression level of a given CXCR2 ligand is not associated with prognosis.

**Table 4 ijms-24-13287-t004:** Association of CXCR2 ligand expression levels in tumor with lymph node metastasis status.

Name of the Cancer	*CXCL1*	*CXCL2*	*CXCL3*	*CXCL5*	*CXCL6*	*PPBP*	*CXCL8*	*CXCR1*	*CXCR2*
Adrenocortical carcinoma	=	↓	=	=	=	↓ *p* = 0.069	=	=	=
Bladder urothelial carcinoma	=	=	=	=	=	=	=	=	=
Breast invasive carcinoma	??	↓	↓	??	??	=	↓	=	=
Cervical squamous cell carcinoma	=	=	=	=	=	=	=	=	=
Cholangiocarcinoma	=	=	=	=	=	=	=	=	=
Colon adenocarcinoma	=	=	=	=	=	=	=	=	=
Lymphoid neoplasm diffuse large B-cell lymphoma	N/A	N/A	N/A	N/A	N/A	N/A	N/A	N/A	N/A
Esophageal carcinoma	=	=	??	=	=	=	↑	=	=
Glioblastoma multiforme	N/A	N/A	N/A	N/A	N/A	N/A	N/A	N/A	N/A
Head and neck squamous cell carcinoma	↓	↓	↓ *p* = 0.082	??	↓	=	↓	↓	↓
Kidney chromophobe	=	=	=	??	??	=	=	=	=
Kidney renal clear cell carcinoma	=	=	=	=	=	↑	=	=	=
Kidney renal papillary cell carcinoma	=	=	=	=	↓	=	=	=	=
Acute myeloid leukemia	N/A	N/A	N/A	N/A	N/A	N/A	N/A	N/A	N/A
Brain lower grade glioma	N/A	N/A	N/A	N/A	N/A	N/A	N/A	N/A	N/A
Liver hepatocellular carcinoma	=	=	=	↑	=	N/A	=	=	=
Lung adenocarcinoma	=	=	=	=	=	=	=	=	=
Lung squamous cell carcinoma	=	↓	↓	=	=	??	=	??	↓
Ovarian serous cystadenocarcinoma	N/A	N/A	N/A	N/A	N/A	N/A	N/A	N/A	N/A
Pancreatic adenocarcinoma	=	=	=	=	=	=	=	=	=
Pheochromocytoma and paraganglioma	N/A	N/A	N/A	N/A	N/A	N/A	N/A	N/A	N/A
Prostate adenocarcinoma	=	↓	↓ *p* = 0.059	↓	=	=	=	↓	↓
Rectum adenocarcinoma	↓	↓	↓	=	=	=	=	=	=
Sarcoma	N/A	N/A	N/A	N/A	N/A	N/A	N/A	N/A	N/A
Skin cutaneous melanoma	=	↑	=	=	=	=	=	=	↑
Stomach adenocarcinoma	↓	↓	↓	=	=	=	=	=	=
Testicular germ cell tumors	N/A	N/A	N/A	N/A	N/A	N/A	N/A	N/A	N/A
Thyroid carcinoma	↑ *p* = 0.094	↑	↑ *p* = 0.082	=	=	=	↑	=	=
Thymoma	N/A	N/A	N/A	N/A	N/A	N/A	N/A	N/A	N/A
Uterine corpus endometrial carcinoma	N/A	N/A	N/A	N/A	N/A	N/A	N/A	N/A	N/A
Uterine carcinosarcoma	N/A	N/A	N/A	N/A	N/A	N/A	N/A	N/A	N/A

↓, blue background—higher expression of CXCR2 ligand in is in tumors with lower lymph node metastasis status; ↓, light blue background—higher expression of CXCR2 ligand in is in tumors with a tendency (*p* < 0.1) to lower lymph node metastasis status; ↑, orange background—higher expression of CXCR2 ligand in is in tumors with a tendency (*p* < 0.1) to worse lymph node metastasis status; ↑, red background—higher expression of CXCR2 ligand in is in tumors with worse lymph node metastasis status; =, gray background—the expression level of a given CXCR2 ligand in a tumor is not associated with lymph node metastasis status; ??, gray background—with increasing pathologic lymph node status, there is an alternating increase and decrease in the expression of a specific CXCR2 ligand.

**Table 5 ijms-24-13287-t005:** Correlation of CXCR2 ligand expression level with proliferation marker *Ki-67/MKI67*.

Name of the Cancer	*CXCL1*	*CXCL2*	*CXCL3*	*CXCL5*	*CXCL6*	*PPBP*	*CXCL8*	*CXCR1*	*CXCR2*
Adrenocortical carcinoma	0.03	0.02	0.08	−0.08	0.02	0.15	0.39	0.11	0.24
Bladder urothelial carcinoma	0.24	0.19	0.26	0.32	0.19	0.12	0.24	0.05	−0.17
Breast invasive carcinoma	0.13	−0.08	0.11	0.23	0.09	0.05	0.26	0.00	0.01
Cervical squamous cell carcinoma and endocervical adenocarcinoma	0.16	0.03	0.06	0.09	0.19	0.01	0.24	0.10	0.18
Cholangiocarcinoma	0.12	0.01	0.28	0.23	0.06	0.17	0.24	0.30	0.35
Colon adenocarcinoma	0.14	0.17	0.18	0.18	0.05	0.04	0.04	0.05	0.04
Lymphoid neoplasm diffuse large B-cell lymphoma	0.10	0.33	0.33	0.12	0.03	0.19	0.26	0.13	0.30
Esophageal carcinoma	0.19	0.16	0.20	0.18	0.13	−0.02	0.17	−0.07	−0.14
Glioblastoma multiforme	−0.17	−0.26	−0.20	−0.20	−0.19	−0.19	−0.29	−0.04	−0.07
Head and neck squamous cell carcinoma	0.09	0.07	0.18	0.14	0.12	−0.09	0.19	0.02	0.02
Kidney chromophobe	0.28	−0.17	0.15	0.27	0.24	0.09	0.29	−0.04	0.12
Kidney renal clear cell carcinoma	0.05	0.03	0.10	0.19	0.03	0.06	0.14	0.08	0.19
Kidney renal papillary cell carcinoma	−0.06	−0.05	−0.02	0.00	−0.07	0.00	0.14	−0.03	0.19
Acute myeloid leukemia	0.04	−0.09	−0.05	−0.01	−0.02	0.08	0.00	0.11	0.29
Brain lower grade glioma	−0.09	−0.13	−0.16	−0.24	−0.10	−0.07	−0.02	0.04	0.19
Liver hepatocellular carcinoma	0.20	−0.12	0.30	0.29	0.15	−0.11	0.20	0.00	0.09
Lung adenocarcinoma	0.08	−0.09	0.08	0.16	0.10	−0.11	0.23	−0.07	−0.08
Lung squamous cell carcinoma	−0.08	−0.16	−0.10	−0.09	−0.13	−0.14	−0.16	−0.16	−0.10
Ovarian serous cystadenocarcinoma	0.07	0.07	0.09	0.13	0.15	0.07	0.14	0.16	0.26
Pancreatic adenocarcinoma	−0.02	−0.09	0.13	0.18	−0.08	0.01	0.10	0.02	−0.04
Pheochromocytoma and paraganglioma	−0.03	−0.10	0.03	−0.07	0.09	0.02	0.05	0.14	0.16
Prostate adenocarcinoma	−0.01	−0.01	0.05	0.05	0.00	0.14	0.21	0.08	0.03
Rectum adenocarcinoma	0.31	0.23	0.33	0.28	0.25	0.20	0.22	0.27	0.28
Sarcoma	0.08	0.00	0.16	0.19	0.05	0.02	0.21	−0.13	0.05
Skin cutaneous melanoma	0.11	0.15	0.15	0.14	0.11	0.02	0.21	0.05	0.11
Stomach adenocarcinoma	0.26	0.21	0.29	0.18	0.07	0.16	0.26	0.08	0.02
Testicular germ cell tumors	0.07	−0.06	0.11	0.05	0.08	−0.03	−0.06	−0.15	0.00
Thyroid carcinoma	0.41	0.34	0.37	0.43	0.28	0.24	0.39	0.12	0.41
Thymoma	−0.25	0.28	0.71	−0.07	−0.02	−0.25	0.32	−0.23	0.11
Uterine corpus endometrial carcinoma	0.00	−0.06	−0.02	0.08	0.08	−0.03	0.11	−0.12	0.05
Uterine carcinosarcoma	−0.02	0.06	0.08	0.33	−0.04	0.04	0.18	−0.03	0.09

Red background—expression of CXCR2 ligand is positively correlated with Ki-67/*MKI67* expression; blue background—CXCR2 ligand expression is negatively correlated with Ki-67/*MKI67* expression; gray background—CXCR2 ligand expression is not significantly correlated with Ki-67/*MKI67* expression.

**Table 6 ijms-24-13287-t006:** Correlation of CXCR2 ligand expression levels with EMT marker: vimentin/*VIM*.

Name of the Cancer	*CXCL1*	*CXCL2*	*CXCL3*	*CXCL5*	*CXCL6*	*PPBP*	*CXCL8*	*CXCR1*	*CXCR2*
Adrenocortical carcinoma	−0.01	0.03	0.08	0.02	0.14	0.09	0.11	0.02	0.07
Bladder urothelial carcinoma	0.25	0.52	0.41	0.41	0.24	0.20	0.19	0.39	−0.15
Breast invasive carcinoma	0.36	0.40	0.39	0.38	0.27	0.23	0.33	0.18	0.31
Cervical squamous cell carcinoma and endocervical adenocarcinoma	−0.05	0.13	0.05	0.25	0.10	0.06	−0.01	0.11	0.01
Cholangiocarcinoma	0.18	0.09	0.23	0.26	0.29	0.11	0.24	0.24	0.20
Colon adenocarcinoma	0.05	−0.07	−0.10	0.27	0.34	0.25	0.40	0.47	0.40
Lymphoid neoplasm diffuse large B-cell lymphoma	0.29	0.41	0.49	0.50	0.10	0.08	0.26	0.43	0.64
Esophageal carcinoma	−0.05	0.02	−0.08	0.16	0.16	0.01	0.06	0.09	−0.08
Glioblastoma multiforme	0.26	0.18	0.36	0.31	0.24	0.23	0.26	0.14	0.15
Head and neck squamous cell carcinoma	0.01	0.25	0.25	0.21	0.13	−0.03	0.02	0.17	−0.07
Kidney chromophobe	0.43	0.48	0.47	0.35	0.18	0.35	0.44	0.32	0.36
Kidney renal clear cell carcinoma	0.09	0.11	0.11	0.25	0.15	0.12	0.13	0.10	0.14
Kidney renal papillary cell carcinoma	0.32	0.20	0.31	0.29	0.35	0.25	0.42	0.25	0.30
Acute myeloid leukemia	0.02	−0.06	−0.11	−0.14	−0.07	−0.15	0.02	0.08	0.16
Brain lower grade glioma	0.04	0.17	0.11	−0.09	0.18	0.02	0.16	0.11	0.46
Liver hepatocellular carcinoma	0.43	0.18	0.39	0.34	0.46	0.17	0.44	0.29	0.42
Lung adenocarcinoma	0.19	0.11	0.14	0.14	0.21	0.27	0.13	0.17	0.30
Lung squamous cell carcinoma	0.06	0.29	0.27	0.39	0.07	0.33	0.06	0.24	0.09
Ovarian serous cystadenocarcinoma	0.15	0.16	0.13	0.19	0.21	0.21	0.17	0.31	0.30
Pancreatic adenocarcinoma	0.21	0.24	0.03	0.16	0.19	0.38	0.34	0.31	0.42
Pheochromocytoma and paraganglioma	0.09	0.17	0.20	0.04	0.04	0.14	0.17	0.08	0.06
Prostate adenocarcinoma	0.26	0.29	0.25	0.24	0.33	0.22	0.27	0.32	0.36
Rectum adenocarcinoma	0.04	−0.04	−0.13	0.33	0.20	0.26	0.40	0.47	0.34
Sarcoma	0.18	0.05	0.25	0.17	0.14	0.15	0.26	0.08	0.05
Skin cutaneous melanoma	0.10	−0.05	0.01	0.03	0.02	0.01	0.10	−0.04	−0.02
Stomach adenocarcinoma	−0.05	−0.07	−0.17	0.01	0.08	−0.02	0.02	0.16	0.16
Testicular germ cell tumors	0.54	0.57	0.58	0.41	0.45	0.23	0.48	0.21	0.31
Thyroid carcinoma	0.02	0.21	0.16	0.04	0.09	0.11	0.13	0.22	0.13
Thymoma	0.46	0.17	0.31	0.52	0.56	−0.12	0.32	0.19	0.30
Uterine corpus endometrial carcinoma	0.17	0.17	0.18	0.18	0.11	0.01	0.17	0.16	0.22
Uterine carcinosarcoma	0.17	0.30	0.26	0.35	0.07	0.19	0.32	0.26	0.11

Red background—expression of CXCR2 ligand is positively correlated with vimentin/*VIM* expression; blue background—CXCR2 ligand expression is negatively correlated with vimentin/*VIM* expression; gray background—CXCR2 ligand expression is not significantly correlated with vimentin/*VIM* expression.

**Table 7 ijms-24-13287-t007:** Correlation of CXCR2 ligand expression levels with EMT marker: N-cadherin/*CDH2*.

Name of the Cancer	*CXCL1*	*CXCL2*	*CXCL3*	*CXCL5*	*CXCL6*	*PPBP*	*CXCL8*	*CXCR1*	*CXCR2*
Adrenocortical carcinoma	0.11	0.20	0.11	0.00	0.16	−0.02	0.18	−0.14	−0.10
Bladder urothelial carcinoma	0.19	0.30	0.26	0.33	0.25	0.10	0.17	0.22	−0.13
Breast invasive carcinoma	0.10	−0.01	0.10	0.15	0.04	0.11	0.33	0.09	0.19
Cervical squamous cell carcinoma and endocervical adenocarcinoma	−0.03	0.19	0.19	0.36	0.02	0.22	0.06	0.11	−0.13
Cholangiocarcinoma	−0.10	−0.05	−0.07	−0.10	−0.02	0.20	0.11	0.00	0.02
Colon adenocarcinoma	−0.15	−0.21	−0.25	0.11	0.16	0.20	0.22	0.28	0.25
Lymphoid neoplasm diffuse large B-cell lymphoma	0.15	0.24	0.30	0.39	0.42	0.06	0.24	0.18	0.28
Esophageal carcinoma	−0.25	−0.17	−0.33	0.05	0.01	−0.17	−0.01	−0.15	−0.10
Glioblastoma multiforme	0.12	0.10	0.16	0.12	0.01	0.10	0.04	0.10	0.17
Head and neck squamous cell carcinoma	−0.04	0.14	0.19	0.21	0.09	0.01	0.07	0.04	−0.08
Kidney chromophobe	0.47	0.20	0.31	0.44	0.29	0.08	0.48	0.05	0.09
Kidney renal clear cell carcinoma	0.03	−0.01	0.00	0.08	0.08	0.12	0.14	0.31	0.42
Kidney renal papillary cell carcinoma	0.15	0.05	0.09	0.27	0.32	0.19	0.22	0.11	0.24
Acute myeloid leukemia	0.03	0.15	0.13	0.17	0.02	0.02	−0.05	−0.02	−0.25
Brain lower grade glioma	−0.04	−0.04	−0.02	0.09	0.06	−0.05	0.11	0.08	0.27
Liver hepatocellular carcinoma	0.15	0.10	0.20	0.08	0.14	0.00	0.20	0.15	0.28
Lung adenocarcinoma	0.16	−0.03	0.10	0.24	0.27	0.04	0.27	0.16	0.13
Lung squamous cell carcinoma	−0.01	0.07	0.10	0.15	0.02	0.17	0.02	0.11	0.07
Ovarian serous cystadenocarcinoma	−0.06	0.00	−0.03	0.12	0.06	0.10	0.07	0.11	0.03
Pancreatic adenocarcinoma	0.13	0.04	−0.07	0.13	0.16	0.25	0.25	0.15	0.24
Pheochromocytoma and paraganglioma	0.20	0.13	0.23	0.28	0.16	0.07	0.24	0.16	0.30
Prostate adenocarcinoma	0.13	0.16	0.17	0.14	0.21	0.14	0.18	0.23	0.29
Rectum adenocarcinoma	−0.23	−0.20	−0.35	0.13	0.07	0.21	0.20	0.23	0.10
Sarcoma	−0.13	−0.23	0.03	0.10	−0.05	0.04	0.07	−0.20	0.04
Skin cutaneous melanoma	−0.03	0.11	0.12	0.11	0.10	0.05	0.15	−0.08	0.04
Stomach adenocarcinoma	−0.13	−0.08	−0.27	−0.04	0.00	0.07	0.02	0.05	0.03
Testicular germ cell tumors	0.07	0.01	0.13	0.05	0.26	−0.04	−0.14	−0.14	−0.02
Thyroid carcinoma	−0.21	−0.09	−0.16	−0.28	−0.13	−0.06	−0.14	0.10	−0.07
Thymoma	0.46	0.03	0.04	0.48	0.53	0.00	−0.02	0.03	0.10
Uterine corpus endometrial carcinoma	−0.04	−0.07	0.01	0.12	0.15	0.03	0.03	−0.03	0.01
Uterine carcinosarcoma	0.24	0.28	0.24	0.23	−0.08	0.31	0.22	0.13	0.19

Red background—expression of CXCR2 ligand is positively correlated with N-cadherin/*CDH2* expression; blue background—CXCR2 ligand expression is negatively correlated with N-cadherin/*CDH2* expression; gray background—CXCR2 ligand expression is not significantly correlated with N-cadherin/*CDH2* expression.

**Table 8 ijms-24-13287-t008:** Correlation of CXCR2 ligand expression levels with EMT marker: *E-cadherin/CDH1*.

Name of the Cancer	*CXCL1*	*CXCL2*	*CXCL3*	*CXCL5*	*CXCL6*	*PPBP*	*CXCL8*	*CXCR1*	*CXCR2*
Adrenocortical carcinoma	−0.10	0.02	0.09	0.11	0.02	−0.03	0.14	0.22	0.14
Bladder urothelial carcinoma	−0.06	−0.18	−0.13	−0.07	0.08	−0.10	0.05	−0.09	0.29
Breast invasive carcinoma	−0.23	−0.30	−0.25	−0.13	−0.13	−0.07	−0.05	0.09	0.03
Cervical squamous cell carcinoma and endocervical adenocarcinoma	0.18	0.01	0.02	0.05	0.22	0.06	0.19	0.08	0.14
Cholangiocarcinoma	−0.06	−0.08	−0.08	−0.17	−0.29	−0.09	0.06	−0.31	−0.21
Colon adenocarcinoma	−0.11	−0.04	−0.07	−0.04	−0.01	−0.04	−0.10	0.06	0.13
Lymphoid neoplasm diffuse large B-cell lymphoma	0.00	0.15	0.23	0.24	−0.03	0.18	−0.09	0.42	0.46
Esophageal carcinoma	0.07	−0.04	0.04	−0.03	0.05	0.01	0.05	0.03	0.13
Glioblastoma multiforme	−0.16	−0.15	−0.25	−0.28	−0.29	−0.19	−0.27	−0.12	−0.05
Head and neck squamous cell carcinoma	−0.03	−0.09	−0.04	0.19	0.13	0.13	0.17	0.15	0.34
Kidney chromophobe	0.09	0.04	−0.03	0.13	0.02	−0.11	0.13	0.07	0.20
Kidney renal clear cell carcinoma	−0.10	−0.13	−0.11	−0.08	−0.03	−0.10	−0.07	0.06	0.17
Kidney renal papillary cell carcinoma	−0.04	0.03	0.02	−0.05	0.00	0.04	−0.06	0.01	0.12
Acute myeloid leukemia	0.23	0.13	0.17	0.36	0.07	0.39	0.00	0.21	−0.01
Brain lower grade glioma	−0.13	0.23	0.22	0.17	0.10	−0.04	−0.05	0.03	0.15
Liver hepatocellular carcinoma	0.20	0.11	0.15	0.13	0.21	−0.03	0.14	0.05	0.16
Lung adenocarcinoma	0.00	−0.02	−0.06	−0.03	0.02	0.09	0.01	−0.07	0.02
Lung squamous cell carcinoma	−0.09	−0.13	−0.20	−0.15	−0.07	−0.08	−0.15	−0.10	0.04
Ovarian serous cystadenocarcinoma	0.04	−0.05	−0.02	0.10	0.09	0.00	0.08	0.28	0.35
Pancreatic adenocarcinoma	0.06	−0.09	0.18	0.14	0.12	0.06	0.21	0.01	0.01
Pheochromocytoma and paraganglioma	0.19	0.13	0.22	0.16	0.11	0.12	0.09	0.13	0.11
Prostate adenocarcinoma	0.08	0.04	0.06	0.19	0.16	0.05	0.23	0.23	0.27
Rectum adenocarcinoma	0.03	−0.07	0.11	−0.07	0.02	0.04	−0.07	0.02	0.07
Sarcoma	−0.32	−0.02	−0.18	−0.10	−0.19	−0.18	−0.23	−0.21	−0.09
Skin cutaneous melanoma	0.22	−0.11	−0.10	−0.06	0.02	−0.07	−0.07	0.06	0.06
Stomach adenocarcinoma	0.21	0.09	0.19	0.08	0.15	0.09	0.20	0.15	0.14
Testicular germ cell tumors	0.20	0.21	0.16	0.49	0.05	0.44	0.37	0.40	0.24
Thyroid carcinoma	0.05	0.10	0.11	0.12	0.06	0.12	0.15	0.10	0.11
Thymoma	0.38	−0.01	−0.37	0.33	0.28	0.18	0.11	0.22	0.32
Uterine corpus endometrial carcinoma	0.20	0.17	0.19	0.29	0.21	0.15	0.38	0.25	0.38
Uterine carcinosarcoma	0.55	0.20	0.30	0.19	0.13	0.09	0.32	0.22	0.18

Red background—expression of CXCR2 ligand is positively correlated with E-cadherin/*CDH1* expression; blue background—CXCR2 ligand expression is negatively correlated with E-cadherin/*CDH1* expression; gray background—CXCR2 ligand expression is not significantly correlated with E-cadherin/*CDH1* expression.

**Table 9 ijms-24-13287-t009:** Correlation of CXCR2 ligand expression levels with T_reg_ recruitment to tumor niche.

Name of the Cancer	*CXCL1*	*CXCL2*	*CXCL3*	*CXCL5*	*CXCL6*	*PPBP*	*CXCL8*	*CXCR1*	*CXCR2*
Adrenocortical carcinoma	−0.14	−0.03	0.18	0.13	0.01	−0.23	−0.10	−0.22	0.12
Bladder urothelial carcinoma	0.18	0.14	0.18	0.16	0.10	−0.01	0.19	0.12	0.02
Breast invasive carcinoma	0.017	−0.01	0.01	0.02	−0.02	−0.05	0.18 *	−0.02	0.12
Cervical squamous cell carcinoma and endocervical adenocarcinoma	−0.02	−0.03	−0.04	0.05	0.16 *	−0.01	0.03	−0.06	0.03
Cholangiocarcinoma	−0.20	−0.05	−0.15	−0.29	−0.17	−0.24	−0.15	−0.14	0.00
Colon adenocarcinoma	0.07	0.07	0.08	0.10	0.14	0.03	0.08	0.17	0.23
Lymphoid neoplasm diffuse large B-cell lymphoma	−0.30	−0.27	−0.13	−0.01	−0.22	−0.17	−0.28	−0.21	0.10
Esophageal carcinoma	−0.06	−0.08 *	−0.08 *	−0.07	0.00	−0.08	0.03	−0.09	0.05
Glioblastoma multiforme	0.12	0.12	0.11	0.21	0.10	−0.16	−0.01	0.06	0.08
Head and neck squamous cell carcinoma	0.016	0.02	0.05	0.01	−0.07	0.03	0.02	0.11 *	0.09 *
Kidney chromophobe	0.00	0.06	0.10	−0.01	−0.14	−0.23	−0.02	−0.15	−0.03
Kidney renal clear cell carcinoma	−0.01	−0.05	−0.03	−0.01	0.05	−0.05	−0.02	−0.04	−0.04
Kidney renal papillary cell carcinoma	−0.13 *	−0.02	−0.11	−0.23 *	−0.19 *	−0.04	−0.07	−0.10 *	−0.05 *
Acute myeloid leukemia	N/A	N/A	N/A	N/A	N/A	N/A	N/A	N/A	N/A
Brain lower grade glioma	0.05	−0.01 *	−0.02	0.04 *	−0.03	−0.08	0.07	0.11	−0.02
Liver hepatocellular carcinoma	−0.02 *	0.10	−0.01 *	−0.12 *	−0.04 *	−0.09	0.07 *	0.05	0.06
Lung adenocarcinoma	0.04	0.08	0.07	0.05	−0.06	−0.01	0.11 *	0.07	0.13
Lung squamous cell carcinoma	0.02	0.14	0.13	0.10	0.02	−0.02	0.08	0.09	0.02
Ovarian serous cystadenocarcinoma	−0.03	−0.02	−0.04	−0.11	−0.02	−0.05	−0.02	0.01	0.05
Pancreatic adenocarcinoma	−0.14 *	−0.05 *	0.01	0.10	−0.14	0.14 *	0.02 *	0.13	0.17
Pheochromocytoma and paraganglioma	0.13	0.08	0.15	0.15 *	0.04	0.05	0.20	0.11	0.04
Prostate adenocarcinoma	−0.05	−0.02	−0.02	−0.08	−0.03	−0.04	0.10	0.08	0.01
Rectum adenocarcinoma	0.14	0.24	0.21 *	0.18	0.16	0.16	0.15	0.11 *	0.20
Sarcoma	0.08	0.07 *	0.11	−0.02	0.06	0.11	0.13	0.10	0.05
Skin cutaneous melanoma	0.16 *	0.05	0.09	−0.04	−0.08	−0.02	0.22	−0.10 *	−0.08
Stomach adenocarcinoma	0.13 *	0.05	0.13 *	0.06	0.02	−0.01	0.15 *	0.11	0.18
Testicular germ cell tumors	−0.25 *	−0.15 *	−0.06 *	−0.39 *	−0.13 *	−0.34 *	−0.20 *	−0.26 *	−0.18 *
Thyroid carcinoma	0.27	0.21	0.27	0.40	0.11	0.23	0.36	0.00	0.26
Thymoma	0.20	0.16	0.16	0.17 *	0.19	0.05	0.20 *	−0.02	0.22
Uterine corpus endometrial carcinoma	0.08	0.17	0.17	−0.02	0.09	−0.03	0.05	0.03	−0.06 *
Uterine carcinosarcoma	−0.08 *	−0.05	−0.07 *	−0.18	0.12	−0.29 *	−0.20	0.07	−0.03

Red background—expression of CXCR2 ligand is positively correlated with T_reg_ count; blue background—CXCR2 ligand expression is negatively correlated with T_reg_ count; gray background—CXCR2 ligand expression is not significantly correlated with T_reg_ count; *—correlation analyzed by other algorithms indicated a different result.

**Table 10 ijms-24-13287-t010:** Correlation of the expression level of CXCR2 ligands with the level of neutrophil recruitment to the tumor niche.

Name of the Cancer	*CXCL1*	*CXCL2*	*CXCL3*	*CXCL5*	*CXCL6*	*PPBP*	*CXCL8*	*CXCR1*	*CXCR2*
Adrenocortical carcinoma	0.13	0.00	0.15	0.15	−0.08	0.22 *	0.14	0.14 *	0.21 *
Bladder urothelial carcinoma	0.23	0.18	0.20	0.21	0.19	0.23	0.30	0.24	0.15
Breast invasive carcinoma	0.12	0.09	0.11	0.09	0.13	0.13	0.09 *	0.10 *	0.13
Cervical squamous cell carcinoma and endocervical adenocarcinoma	0.26	0.19	0.24	0.15	0.22	0.23	0.32	0.38	0.24
Cholangiocarcinoma	0.05	−0.22	0.12	−0.08	−0.02	0.32	0.16	0.14	0.12
Colon adenocarcinoma	0.27	0.25	0.25	0.35	0.36	0.18	0.43	0.58	0.54
Lymphoid neoplasm diffuse large B-cell lymphoma	0.38	0.31	0.31	0.27	0.11	−0.03	0.13	0.26	0.13
Esophageal carcinoma	0.31	0.16	0.23	0.19	0.24	0.30	0.29	0.34	0.35
Glioblastoma multiforme	0.09 *	0.07 *	0.11 *	0.10 *	0.12	0.16	0.11	0.17 *	0.18
Head and neck squamous cell carcinoma	0.32	0.20	0.19	0.24	0.26	0.32	0.33	0.40	0.43
Kidney chromophobe	0.02	0.01	0.22 *	−0.05	−0.17	0.05	−0.01	−0.07	−0.15
Kidney renal clear cell carcinoma	0.10	0.14	0.17	0.00	0.04 *	0.20	0.21	0.29	0.25
Kidney renal papillary cell carcinoma	0.04	−0.06	0.07	0.09 *	0.12 *	0.12	0.12	0.18	0.15
Acute myeloid leukemia	N/A	N/A	N/A	N/A	N/A	N/A	N/A	N/A	N/A
Brain lower grade glioma	0.12	0.18 *	0.20	0.14 *	0.09	0.08	0.18	0.20	0.29
Liver hepatocellular carcinoma	0.11	0.12	0.12	0.18	0.12	0.07	0.13	0.18	0.21
Lung adenocarcinoma	0.13	0.18	0.19	0.19	0.10	0.23	0.21	0.40	0.33
Lung squamous cell carcinoma	0.31	0.22	0.30	0.15	0.19	0.19	0.40	0.48	0.42
Ovarian serous cystadenocarcinoma	0.08	0.02	0.10	−0.01	0.09	0.06	0.09	0.12 *	0.10
Pancreatic adenocarcinoma	0.17	0.18	0.15	0.28	−0.01 *	0.05 *	0.23	0.26	0.25
Pheochromocytoma and paraganglioma	0.18	0.20	0.17	0.22	0.18	0.30	0.21	0.36	0.36
Prostate adenocarcinoma	0.06 *	0.07 *	0.08 *	0.03 *	0.07	0.08	0.08 *	0.12	0.11
Rectum adenocarcinoma	0.16 *	0.16	0.19	0.21	0.26	0.03	0.19 *	0.40	0.36
Sarcoma	0.16	0.22	0.04 *	0.02	0.09	0.16	0.17	0.13	0.14
Skin cutaneous melanoma	0.09 *	0.05 *	0.09	0.09	0.06	0.02	0.13	0.06	0.09 *
Stomach adenocarcinoma	0.27	0.26	0.25	0.32	0.27	0.26	0.40	0.56	0.53
Testicular germ cell tumors	0.02	0.05	0.11	0.25 *	−0.05	0.04	0.24	0.09	0.17
Thyroid carcinoma	−0.08 *	−0.02 *	−0.03 *	−0.05 *	−0.05	0.06 *	0.00	0.07	−0.03 *
Thymoma	0.35	−0.09	−0.02	0.29	0.36 *	0.05	0.15 *	0.11	0.25
Uterine corpus endometrial carcinoma	0.15	0.19 *	0.14	0.21 *	0.19	0.11	0.30	0.37	0.18 *
Uterine carcinosarcoma	−0.02 *	0.16	0.06 *	−0.16	0.07	0.16	0.10	−0.04 *	0.10

Red background—expression of CXCR2 ligand is positively correlated with the count of neutrophils; gray background—CXCR2 ligand expression is not significantly correlated with the count of neutrophils; *—correlation analyzed by other algorithms indicated a different result.

**Table 11 ijms-24-13287-t011:** Correlation of CXCR2 ligand expression levels with the level of tumor infiltration by MDSCs.

Name of the Cancer	*CXCL1*	*CXCL2*	*CXCL3*	*CXCL5*	*CXCL6*	*PPBP*	*CXCL8*	*CXCR1*	*CXCR2*
Adrenocortical carcinoma	0.16	0.07	0.08	−0.13	0.09	−0.02	0.36	0.15	0.02
Bladder urothelial carcinoma	0.23	0.13	0.20	0.27	0.17	0.18	0.24	−0.05	−0.26
Breast invasive carcinoma	0.19	0.04	0.14	0.21	0.11	0.05	0.19	−0.19	−0.41
Cervical squamous cell carcinoma and endocervical adenocarcinoma	0.19	0.29	0.26	0.23	0.04	0.17	0.36	0.07	−0.35
Cholangiocarcinoma	0.38	0.26	0.32	0.44	0.39	0.15	0.34	0.32	0.27
Colon adenocarcinoma	0.02	0.07	0.06	0.00	−0.10	0.02	−0.09	−0.37	−0.34
Lymphoid neoplasm diffuse large B-cell lymphoma	−0.03	0.10	0.00	0.04	0.15	0.38	0.16	0.06	−0.19
Esophageal carcinoma	−0.01	−0.24	−0.26	0.16	0.10	−0.04	0.15	−0.21	−0.21
Glioblastoma multiforme	0.00	−0.14	−0.03	−0.18	0.05	−0.08	−0.01	−0.16	−0.46
Head and neck squamous cell carcinoma	0.14	0.18	0.24	0.23	0.13	0.08	0.25	−0.05	−0.34
Kidney chromophobe	−0.18	−0.32	−0.29	−0.13	0.04	−0.04	−0.15	−0.11	−0.15
Kidney renal clear cell carcinoma	0.08	0.09	0.05	0.09	0.05	0.04	0.00	−0.18	−0.32
Kidney renal papillary cell carcinoma	−0.15	−0.24	−0.16	−0.11	−0.08	−0.07	−0.15	−0.19	−0.24
Acute myeloid leukemia	N/A	N/A	N/A	N/A	N/A	N/A	N/A	N/A	N/A
Brain lower grade glioma	0.01	−0.24	−0.28	−0.25	−0.15	0.00	−0.09	−0.05	−0.27
Liver hepatocellular carcinoma	0.22	−0.19	0.28	0.33	0.23	−0.11	0.20	−0.04	−0.02
Lung adenocarcinoma	0.10	−0.12	0.05	0.13	0.12	−0.07	0.32	−0.16	−0.37
Lung squamous cell carcinoma	0.20	−0.16	0.00	−0.05	0.15	−0.09	0.03	−0.19	−0.29
Ovarian serous cystadenocarcinoma	−0.06	−0.06	−0.05	0.05	−0.03	0.11	0.00	−0.23	−0.42
Pancreatic adenocarcinoma	−0.05	−0.17	−0.05	0.00	−0.14	−0.04	−0.02	−0.24	−0.44
Pheochromocytoma and paraganglioma	−0.49	−0.43	−0.38	−0.33	−0.13	−0.36	−0.40	−0.27	−0.32
Prostate adenocarcinoma	−0.24	−0.23	−0.21	−0.22	−0.21	−0.06	−0.14	−0.32	−0.40
Rectum adenocarcinoma	0.07	0.10	0.09	0.22	0.17	0.14	0.02	−0.24	−0.20
Sarcoma	−0.20	−0.33	−0.08	0.06	−0.13	0.03	−0.02	−0.29	−0.32
Skin cutaneous melanoma	0.20	0.13	0.25	0.23	0.23	0.20	0.29	0.08	−0.01
Stomach adenocarcinoma	0.18	0.22	0.16	0.07	0.14	0.26	0.16	−0.17	−0.23
Testicular germ cell tumors	−0.15	−0.22	−0.16	0.15	−0.24	0.19	−0.05	0.10	−0.02
Thyroid carcinoma	−0.21	−0.12	−0.14	−0.18	−0.22	−0.02	−0.16	−0.17	−0.44
Thymoma	0.21	−0.08	0.12	0.03	0.14	−0.24	0.12	−0.25	−0.03
Uterine corpus endometrial carcinoma	−0.02	−0.23	−0.12	−0.14	0.00	−0.02	0.01	−0.35	−0.40
Uterine carcinosarcoma	−0.05	−0.18	−0.10	0.03	−0.22	0.11	0.06	−0.30	−0.29

Red background—expression of CXCR2 ligand is positively correlated with the count of MDSCs; blue background—CXCR2 ligand expression is negatively correlated with the count of MDSCs; gray background—CXCR2 ligand expression is not significantly correlated with the count of MDSCs.

**Table 12 ijms-24-13287-t012:** Correlation of CXCR2 ligand expression levels with the level of tumor infiltration by CD8^+^ T cells.

Name of the Cancer	*CXCL1*	*CXCL2*	*CXCL3*	*CXCL5*	*CXCL6*	*PPBP*	*CXCL8*	*CXCR1*	*CXCR2*
Adrenocortical carcinoma	−0.05	0.00	0.06	0.11	0.21	0.15	−0.04	−0.09	0.06
Bladder urothelial carcinoma	−0.11 *	−0.05	−0.09	−0.14 *	−0.10	−0.11 *	−0.11 *	−0.06	0.08 *
Breast invasive carcinoma	0.12	0.01	0.07	0.06	0.12	−0.04	−0.04	−0.02	0.06
Cervical squamous cell carcinoma and endocervical adenocarcinoma	−0.17	−0.13 *	−0.15 *	−0.14 *	−0.05	−0.15 *	−0.29	−0.14 *	0.04
Cholangiocarcinoma	−0.28	−0.27	−0.19	−0.13	−0.18	−0.07	−0.37 *	−0.32	−0.17
Colon adenocarcinoma	−0.03	−0.02	−0.02	−0.16	−0.08	−0.22	−0.10	0.04	0.07
Lymphoid neoplasm diffuse large B-cell lymphoma	−0.05	−0.02	−0.03	−0.13	−0.38 *	−0.34 *	−0.36 *	0.10	0.08
Esophageal carcinoma	−0.13	−0.05	−0.09	−0.15	−0.22 *	−0.09	−0.15 *	−0.08	−0.05
Glioblastoma multiforme	−0.08	−0.17	−0.17	−0.09	−0.07	0.01	−0.20 *	−0.15	−0.13
Head and neck squamous cell carcinoma	−0.10 *	−0.09	−0.10	−0.24	−0.17	−0.19	−0.25	−0.01	0.13 *
Kidney chromophobe	0.08	−0.01	0.06	−0.13	0.04	0.26 *	0.21	−0.03	0.04
Kidney renal clear cell carcinoma	−0.11	−0.07	−0.09	−0.06	−0.15	−0.06	−0.15	−0.10 *	−0.05
Kidney renal papillary cell carcinoma	0.06	0.09	0.13	0.10	0.07	0.00	0.02 *	0.05	0.08 *
Acute myeloid leukemia	N/A	N/A	N/A	N/A	N/A	N/A	N/A	N/A	N/A
Brain lower grade glioma	−0.04	0.05	0.10 *	0.00	0.05	−0.04	−0.04	−0.03	0.01
Liver hepatocellular carcinoma	−0.15 *	−0.05	−0.02 *	−0.14 *	−0.21 *	−0.08	−0.18 *	−0.04	−0.06
Lung adenocarcinoma	−0.12 *	−0.01	−0.09	−0.17	−0.08	−0.23	−0.21	−0.10 *	−0.11 *
Lung squamous cell carcinoma	−0.26	−0.09	−0.11 *	−0.07	−0.17	−0.11 *	−0.15	−0.14 *	−0.05
Ovarian serous cystadenocarcinoma	−0.10	−0.15	−0.11	−0.08	−0.03	−0.26	−0.13 *	−0.10	0.00
Pancreatic adenocarcinoma	0.09	0.12	0.00	−0.05	0.16	−0.01	0.04	0.21	0.31
Pheochromocytoma and paraganglioma	−0.06	−0.14	−0.06	−0.12	−0.06	−0.09	−0.15 *	−0.09	−0.08
Prostate adenocarcinoma	0.02	−0.04	0.01	0.04	0.00 *	−0.04	−0.05 *	−0.07	−0.04 *
Rectum adenocarcinoma	−0.01	0.14	−0.02	−0.02	−0.04	0.02	0.07	0.17	0.13
Sarcoma	0.05	0.11	0.10	0.00	0.02	−0.06	−0.03	−0.07	0.00
Skin cutaneous melanoma	−0.16	0.00	−0.11	−0.12	−0.19	−0.20	−0.14	−0.16	−0.10
Stomach adenocarcinoma	−0.18	−0.19	−0.12	−0.16	−0.24	−0.24	−0.22	−0.11	−0.10 *
Testicular germ cell tumors	−0.17	−0.09	−0.14	−0.35 *	−0.08	−0.31	−0.31	−0.20 *	−0.18 *
Thyroid carcinoma	−0.01	−0.02	−0.03	−0.11 *	0.11	−0.21	−0.08	0.10	0.18 *
Thymoma	−0.34	0.09	0.46	−0.12	−0.07	−0.09	0.09	−0.08	−0.10
Uterine corpus endometrial carcinoma	−0.07	−0.01	−0.05	−0.15	−0.07	−0.08	−0.26 *	0.03 *	0.02
Uterine carcinosarcoma	−0.09	−0.02	−0.04	−0.08	0.21	0.21	−0.08	0.39	0.09

Red background—expression of CXCR2 ligand is positively correlated with the count of CD8^+^ T cells; blue background—CXCR2 ligand expression is negatively correlated with the count of CD8^+^ T cells; gray background—CXCR2 ligand expression is not significantly correlated with the count of CD8^+^ T cells; *—correlation analyzed by other algorithms indicated a different result.

**Table 13 ijms-24-13287-t013:** Correlation of CXCR2 ligand expression levels with the level of tumor infiltration by NK cells.

Name of the Cancer	*CXCL1*	*CXCL2*	*CXCL3*	*CXCL5*	*CXCL6*	*PPBP*	*CXCL8*	*CXCR1*	*CXCR2*
Adrenocortical carcinoma	−0.09	−0.22	−0.08	0.10	−0.01	−0.08	−0.06	−0.06	0.08
Bladder urothelial carcinoma	0.02	0.03	0.04	0.01	−0.02	−0.03	−0.01	−0.04	−0.03
Breast invasive carcinoma	−0.02	−0.06	−0.05	−0.04	0.02	−0.02	−0.04	0.00	−0.07
Cervical squamous cell carcinoma and endocervical adenocarcinoma	0.03	0.03	0.01	0.06	−0.01	0.02	0.07	0.02	0.00
Cholangiocarcinoma	−0.01	−0.22	−0.31	−0.05	0.10	−0.48 *	−0.18	−0.31	−0.35
Colon adenocarcinoma	−0.02	0.00	−0.01	−0.02	−0.07	−0.18 *	0.02	−0.08	−0.09
Lymphoid neoplasm diffuse large B-cell lymphoma	0.29	0.39	0.35	0.16	−0.15	0.14	0.24	0.26	0.37 *
Esophageal carcinoma	−0.04	0.01	−0.03	−0.14	−0.03	−0.09	−0.07	−0.16 *	−0.10
Glioblastoma multiforme	−0.06	−0.06	0.00	0.00	−0.06	0.02	−0.04	0.02	−0.07
Head and neck squamous cell carcinoma	−0.02	0.00	0.03	0.03	0.00	0.04 *	0.06	−0.02	−0.02
Kidney chromophobe	0.18	0.07	0.06	0.00	0.11	−0.05	0.10	−0.28 *	−0.12
Kidney renal clear cell carcinoma	−0.01	0.00	0.05	0.01	−0.04	−0.02	−0.05 *	0.01	−0.04
Kidney renal papillary cell carcinoma	−0.03 *	0.07	0.00 *	−0.06	−0.06	−0.16 *	−0.11	−0.14 *	−0.13 *
Acute myeloid leukemia	N/A	N/A	N/A	N/A	N/A	N/A	N/A	N/A	N/A
Brain lower grade glioma	0.04	0.03 *	0.11	0.10	−0.04	0.01	0.07	−0.02	−0.04
Liver hepatocellular carcinoma	0.03	0.03	0.05	0.06	0.08	−0.06	0.01	−0.11 *	−0.09
Lung adenocarcinoma	−0.01	0.01	0.00	−0.06	−0.07	−0.05	−0.04	−0.08	−0.04
Lung squamous cell carcinoma	−0.02	−0.06	−0.03	−0.12	−0.06	−0.11 *	−0.02	−0.03	0.04
Ovarian serous cystadenocarcinoma	0.01	−0.06	−0.04	−0.13	−0.03	−0.14	−0.03	−0.21 *	−0.09
Pancreatic adenocarcinoma	−0.15	−0.22 *	−0.22 *	−0.07	0.00	−0.13	−0.23 *	−0.17 *	−0.08
Pheochromocytoma and paraganglioma	0.01	0.00	0.04	−0.03	−0.11	−0.01	0.04	−0.02	−0.05
Prostate adenocarcinoma	−0.01	0.02	0.00	−0.01	−0.01	−0.02	0.03	−0.03	−0.05
Rectum adenocarcinoma	0.11	0.05	0.04	−0.02	0.05	0.16	0.09	−0.01	0.07
Sarcoma	−0.02	−0.07	−0.06	0.03	0.01	0.10	0.08	−0.02	0.03
Skin cutaneous melanoma	0.05	−0.02	−0.01	−0.08	−0.09	−0.08	0.02	−0.08	−0.08
Stomach adenocarcinoma	0.02	0.04	0.05	0.02	−0.03	0.03	0.06	0.05	0.05
Testicular germ cell tumors	−0.22	−0.22 *	−0.22 *	−0.27	−0.18 *	−0.13	−0.21 *	−0.09	−0.17 *
Thyroid carcinoma	0.06	0.01	0.04	−0.01	0.05 *	0.00	0.04	0.02	−0.01
Thymoma	0.15	−0.06	0.02 *	0.14	0.18	−0.01	0.12	0.18	−0.17
Uterine corpus endometrial carcinoma	−0.25 *	0.09	−0.02	−0.04	−0.25 *	0.11	−0.15	0.00	−0.09
Uterine carcinosarcoma	−0.20	−0.45 *	−0.035 *	−0.22	0.04	−0.03	−0.28 *	−0.02	0.04

Red background—expression of CXCR2 ligand is positively correlated with the count of NK cells; blue background—CXCR2 ligand expression is negatively correlated with the count of NK cells; gray background—CXCR2 ligand expression is not significantly correlated with the count of NK cells; *—correlation analyzed by other algorithms indicated a different result.

**Table 14 ijms-24-13287-t014:** Correlation of CXCR2 ligand expression levels with the level of tumor infiltration by conventional (myeloid) dendritic cells.

Name of the Cancer	*CXCL1*	*CXCL2*	*CXCL3*	*CXCL5*	*CXCL6*	*PPBP*	*CXCL8*	*CXCR1*	*CXCR2*
Adrenocortical carcinoma	0.01	0.18	0.16	0.02	0.15	0.21	−0.05	−0.05	0.14
Bladder urothelial carcinoma	0.16	0.22	0.20	0.22	0.08	0.11	0.08	0.13	−0.07
Breast invasive carcinoma	0.29	0.17	0.23	0.24	0.27	0.09	0.16	0.08	0.18
Cervical squamous cell carcinoma and endocervical adenocarcinoma	−0.16	−0.13	−0.16	−0.12	−0.07	−0.17	−0.30	−0.10	0.07
Cholangiocarcinoma	0.03	0.06	0.22	−0.09	0.00	0.29	0.07	0.05	0.16
Colon adenocarcinoma	0.15	0.07	0.06	0.25	0.31	0.12	0.44	0.49	0.47
Lymphoid neoplasm diffuse large B-cell lymphoma	−0.05	−0.19	−0.09	0.03	0.08	−0.35	−0.13	0.13	0.32
Esophageal carcinoma	−0.15	−0.12	−0.16	−0.04	−0.12	0.01	0.00	0.01	−0.06
Glioblastoma multiforme	0.10	−0.05	0.15	0.21	0.16	0.08	0.09	0.03	0.06
Head and neck squamous cell carcinoma	−0.05	0.04	0.05	0.01	−0.02	−0.02	−0.07	0.07	0.10
Kidney chromophobe	0.44	0.46	0.46	0.35	0.09	0.26	0.47	0.18	0.15
Kidney renal clear cell carcinoma	0.11 *	0.08	0.17	0.21	0.13	0.03	0.36	0.13	0.24
Kidney renal papillary cell carcinoma	0.32	0.31	0.33	0.32	0.27	0.29	0.35	0.29	0.34
Acute myeloid leukemia	N/A	N/A	N/A	N/A	N/A	N/A	N/A	N/A	N/A
Brain lower grade glioma	0.09	−0.05	−0.07	−0.06	0.02	−0.08	0.05	−0.06	−0.02 *
Liver hepatocellular carcinoma	0.13	0.04	0.14	0.06	0.03	−0.14 *	0.07	0.03	0.06
Lung adenocarcinoma	0.01	0.11	−0.01	0.01	0.03	0.25	−0.01	0.06	0.34
Lung squamous cell carcinoma	−0.04	0.16	0.13	0.25	0.04	0.18	0.15 *	0.25	0.20
Ovarian serous cystadenocarcinoma	0.22	0.17	0.17	0.11	0.20	−0.04	0.21	0.16	0.14
Pancreatic adenocarcinoma	0.18	0.23	0.17	0.14	0.12	0.33	0.27	0.32	0.47
Pheochromocytoma and paraganglioma	0.41	0.39	0.38	0.21	0.08	0.23	0.39	0.16	0.21
Prostate adenocarcinoma	0.20	0.16	0.18	0.16	0.13	0.08 *	0.16	0.16	0.12
Rectum adenocarcinoma	0.08	0.03	−0.01	0.16	0.18	0.02	0.30	0.37	0.34
Sarcoma	0.46	0.46	0.36	0.27	0.34	0.12	0.25	0.27	0.30
Skin cutaneous melanoma	−0.03	0.13	0.06	0.10	−0.04	−0.03	−0.04	0.06	0.12 *
Stomach adenocarcinoma	0.02	−0.09 *	−0.02	−0.03	−0.04 *	−0.06 *	0.09	0.23	0.23
Testicular germ cell tumors	0.37	0.42	0.35	0.20	0.33	0.01	0.35	0.20	0.30
Thyroid carcinoma	0.69	0.59	0.57	0.73	0.47	0.45	0.61	0.20	0.63
Thymoma	−0.54	0.08	0.23	−0.30 *	−0.31 *	−0.16	−0.10	−0.13	−0.23 *
Uterine corpus endometrial carcinoma	0.09	0.14	0.04	−0.02	−0.10	0.01	0.00	0.40	0.26
Uterine carcinosarcoma	0.21	0.40	0.33	0.09	0.15	0.25	0.19	0.36	0.17

Red background—expression of CXCR2 ligand is positively correlated with the count of conventional (myeloid) dendritic cells; blue background—CXCR2 ligand expression is negatively correlated with the count of conventional (myeloid) dendritic cells; gray background—CXCR2 ligand expression is not significantly correlated with the count of conventional (myeloid) dendritic cells; *—correlation analyzed by other algorithms indicated a different result.

**Table 15 ijms-24-13287-t015:** Correlation of CXCR2 ligand expression levels with the level of tumor infiltration by plasmacytoid dendritic cells.

Name of the Cancer	*CXCL1*	*CXCL2*	*CXCL3*	*CXCL5*	*CXCL6*	*PPBP*	*CXCL8*	*CXCR1*	*CXCR2*
Adrenocortical carcinoma	0.04	0.13	−0.16	−0.29	−0.05	0.12	0.03	0.24	0.19
Bladder urothelial carcinoma	0.18	0.18	0.18	0.16	0.01	0.02	0.09	0.01	−0.17
Breast invasive carcinoma	0.16	−0.05	0.11	0.13	0.10	0.01	0.11	0.00	0.06
Cervical squamous cell carcinoma and endocervical adenocarcinoma	−0.16	−0.07	−0.13	−0.12	−0.10	−0.12	−0.18	−0.03	0.05
Cholangiocarcinoma	−0.05	0.03	0.12	0.14	0.09	0.35	0.05	0.20	0.25
Colon adenocarcinoma	0.08	0.18	0.06	0.06	0.06	−0.16	0.06	0.09	0.10
Lymphoid neoplasm diffuse large B-cell lymphoma	0.10	0.26	0.09	0.15	−0.22	−0.05	−0.25	−0.08	−0.03
Esophageal carcinoma	0.13	0.17	0.19	0.08	0.02	0.04	0.05	0.17	−0.04
Glioblastoma multiforme	−0.10	0.01	−0.05	−0.01	−0.10	−0.05	−0.18	−0.17	0.05
Head and neck squamous cell carcinoma	0.01	0.01	0.00	−0.24	−0.13	−0.18	−0.20	0.07	0.04
Kidney chromophobe	0.06	0.03	0.11	−0.16	−0.16	−0.07	−0.01	0.20	0.03
Kidney renal clear cell carcinoma	0.05	−0.03	−0.06	0.04	0.03	−0.06	−0.02	−0.10	−0.06
Kidney renal papillary cell carcinoma	0.04	0.11	0.11	0.03	−0.01	−0.07	0.02	0.05	0.02
Acute myeloid leukemia	N/A	N/A	N/A	N/A	N/A	N/A	N/A	N/A	N/A
Brain lower grade glioma	0.00	−0.08	−0.02	−0.02	−0.05	0.06	−0.02	0.09	0.16
Liver hepatocellular carcinoma	−0.09	−0.05	−0.03	−0.14	−0.18	−0.05	−0.17	0.08	0.07
Lung adenocarcinoma	0.08	0.04	0.13	0.09	0.08	−0.17	0.15	0.03	−0.03
Lung squamous cell carcinoma	−0.16	−0.03	0.02	0.10	−0.18	−0.09	−0.02	−0.03	−0.08
Ovarian serous cystadenocarcinoma	0.17	0.01	0.06	0.02	0.00	−0.12	0.10	−0.08	−0.07
Pancreatic adenocarcinoma	−0.07	−0.07	0.04	0.18	−0.05	−0.09	−0.03	0.13	0.09
Pheochromocytoma and paraganglioma	−0.06	−0.07	−0.09	−0.06	−0.07	0.02	−0.07	−0.04	0.02
Prostate adenocarcinoma	0.05	0.05	0.06	0.04	−0.04	0.07	0.08	−0.02	−0.04
Rectum adenocarcinoma	0.08	0.22	0.10	0.07	0.12	−0.07	0.09	0.01	0.06
Sarcoma	0.11	0.23	0.26	0.15	0.10	0.02	0.02	0.07	0.06
Skin cutaneous melanoma	−0.03	0.04	−0.03	−0.05	−0.13	−0.22	−0.11	−0.10	−0.12
Stomach adenocarcinoma	0.22	0.10	0.26	0.07	−0.07	0.03	0.20	0.15	0.11
Testicular germ cell tumors	−0.15	−0.11	−0.14	−0.25	−0.16	−0.30	−0.20	−0.18	−0.04
Thyroid carcinoma	0.05	−0.03	0.00	−0.06	0.11	−0.12	−0.04	0.08	0.09
Thymoma	−0.55	0.05	0.11	−0.42	−0.46	0.02	−0.28	0.00	−0.18
Uterine corpus endometrial carcinoma	−0.05	0.02	−0.07	−0.10	−0.15	−0.07	−0.25	0.18	0.04
Uterine carcinosarcoma	−0.02	0.22	0.14	−0.07	−0.06	0.05	−0.08	0.03	−0.23

Red background—expression of CXCR2 ligand is positively correlated with the count of plasmacytoid dendritic cells; blue background—CXCR2 ligand expression is negatively correlated with the count of plasmacytoid dendritic cells; gray background—CXCR2 ligand expression is not significantly correlated with the count of plasmacytoid dendritic cells.

**Table 16 ijms-24-13287-t016:** Correlation of CXCR2 ligand expression levels with the count of endothelial cells in tumorigenesis.

Name of the Cancer	*CXCL1*	*CXCL2*	*CXCL3*	*CXCL5*	*CXCL6*	*PPBP*	*CXCL8*	*CXCR1*	*CXCR2*
Adrenocortical carcinoma	−0.09	0.03	−0.08	−0.16	−0.17	0.15	−0.26	0.17	0.06
Bladder urothelial carcinoma	−0.20	0.10	−0.06	−0.04	−0.18 *	0.03	−0.20	0.17	−0.06
Breast invasive carcinoma	−0.08	0.15	−0.05	−0.12	0.00	0.12	−0.12	0.16	0.12
Cervical squamous cell carcinoma and endocervical adenocarcinoma	−0.03	0.15	0.12 *	0.21	−0.03 *	0.15	0.00 *	0.22	−0.02
Cholangiocarcinoma	0.10	−0.04	−0.05	−0.08	−0.08	0.13	−0.14	−0.05	−0.09
Colon adenocarcinoma	0.06	−0.02	−0.08 *	0.24	0.25	0.17	0.28	0.42	0.38
Lymphoid neoplasm diffuse large B-cell lymphoma	0.19	0.34	0.09	0.24	−0.01	0.41	0.13	0.03	−0.29
Esophageal carcinoma	0.21	0.37	0.33	0.20	0.17	0.17 *	0.01	0.26	0.03
Glioblastoma multiforme	0.15	−0.10	−0.09	0.00	0.03	0.09	0.09	0.17	0.10
Head and neck squamous cell carcinoma	0.02	0.18	0.14	0.27	0.24	0.23	0.09 *	0.33	0.13
Kidney chromophobe	0.06	0.28	0.15	0.04	0.05	0.17	0.02	0.09	0.14 *
Kidney renal clear cell carcinoma	−0.04	0.08	0.05	−0.05	0.06	0.25	−0.04	0.17	0.09 *
Kidney renal papillary cell carcinoma	−0.23	−0.11 *	−0.16	−0.26	−0.34	0.00	0.00	0.09	0.03
Acute myeloid leukemia	N/A	N/A	N/A	N/A	N/A	N/A	N/A	N/A	N/A
Brain lower grade glioma	0.18	−0.14 *	−0.17	−0.29	−0.09	0.20	0.13	0.19	0.13
Liver hepatocellular carcinoma	−0.24 *	0.07	−0.28	−0.31	−0.21 *	0.08	−0.25 *	0.03	−0.03
Lung adenocarcinoma	0.08	0.17	0.18	0.16	0.01	0.28	0.05	0.43	0.31
Lung squamous cell carcinoma	−0.10	0.29	0.13	0.19	−0.05	0.26	−0.04	0.25	0.11
Ovarian serous cystadenocarcinoma	−0.04	0.06	0.00	−0.05	−0.04	0.16	−0.06	0.12 *	0.07 *
Pancreatic adenocarcinoma	0.12 *	0.21	−0.06	0.03	0.06 *	0.12 *	0.05	0.26	0.35
Pheochromocytoma and paraganglioma	0.02	0.05	0.01	−0.13	0.01	0.17	0.03	0.01	0.00
Prostate adenocarcinoma	0.01 *	0.06 *	−0.04	−0.05	0.02 *	0.14	−0.01 *	0.16	0.02 *
Rectum adenocarcinoma	0.24 *	0.18	0.09	0.39	0.35	0.06	0.38	0.42	0.36
Sarcoma	0.26	0.26	0.04	−0.05	0.13	0.24	0.15	0.38	0.31
Skin cutaneous melanoma	−0.23	0.04	−0.05	0.10 *	0.05	0.17	−0.07	0.10	0.10
Stomach adenocarcinoma	−0.02	−0.01	−0.08 *	0.14	0.16	0.01	0.04	0.16	0.15
Testicular germ cell tumors	0.45	0.47	0.43	0.54	0.30	0.47	0.42	0.50	0.42
Thyroid carcinoma	−0.38	−0.23	−0.31	−0.53	−0.18	−0.20	−0.40	0.10	−0.23
Thymoma	0.33	0.00	−0.32	0.17 *	0.20	0.28	−0.17	0.30	−0.15
Uterine corpus endometrial carcinoma	−0.27 *	−0.21 *	−0.26	−0.20	0.04	0.04	−0.18	−0.11	−0.18
Uterine carcinosarcoma	−0.24	0.02	−0.18	0.08	−0.09	−0.16	−0.20	0.12	0.09

Red background—expression of CXCR2 ligand is positively correlated with the count of endothelial cells; blue background—CXCR2 ligand expression is negatively correlated with the count of endothelial cells; gray background—CXCR2 ligand expression is not significantly correlated with the count of endothelial cells; *—correlation analyzed by other algorithms indicated a different result.

**Table 17 ijms-24-13287-t017:** Correlation of CXCR2 ligand expression levels with the count of macrophages in the tumor niche.

Name of the Cancer	*CXCL1*	*CXCL2*	*CXCL3*	*CXCL5*	*CXCL6*	*PPBP*	*CXCL8*	*CXCR1*	*CXCR2*
Adrenocortical carcinoma	−0.08 *	0.06 *	0.04	−0.04	−0.06	0.21 *	0.10	0.09	0.11
Bladder urothelial carcinoma	0.17	0.23	0.20	0.25	0.06	0.03	0.07	0.00	−0.26
Breast invasive carcinoma	0.01	−0.13	0.04	0.06	−0.01	0.07	0.19	0.08 *	0.30
Cervical squamous cell carcinoma and endocervical adenocarcinoma	−0.25	−0.21	−0.29	−0.19	−0.18 *	−0.17	−0.33	−0.12 *	0.07
Cholangiocarcinoma	−0.18	−0.46	−0.18	−0.22	−0.06	−0.09	−0.09	−0.01	0.12
Colon adenocarcinoma	0.08	−0.03	−0.02	0.21	0.23	0.14	0.35	0.43	0.40
Lymphoid neoplasm diffuse large B-cell lymphoma	0.30	0.16	0.22	0.33 *	−0.03	−0.06	0.31	0.01	0.22
Esophageal carcinoma	−0.11	−0.01	0.03	0.18	−0.06	0.08	0.06	0.12	−0.14 *
Glioblastoma multiforme	0.18	0.16	0.19 *	0.30	0.28	0.22	0.24	0.08	0.33
Head and neck squamous cell carcinoma	−0.12 *	0.02	0.01	−0.03	−0.13 *	−0.10	−0.09	0.02	−0.09
Kidney chromophobe	0.21	0.22	0.37	0.18	−0.14	0.14 *	0.30	0.18	0.18 *
Kidney renal clear cell carcinoma	0.03	−0.09	0.02	0.14	0.02	−0.10	0.24	−0.02	0.13
Kidney renal papillary cell carcinoma	0.14	0.23	0.17	0.10 *	0.09 *	0.09 *	0.20	0.16	0.21
Acute myeloid leukemia	N/A	N/A	N/A	N/A	N/A	N/A	N/A	N/A	N/A
Brain lower grade glioma	0.01	0.11	0.14	−0.08	0.14	0.02	0.17	0.08	0.50
Liver hepatocellular carcinoma	0.04	0.14	0.02	0.01	0.03	−0.02	0.05	0.04	0.08
Lung adenocarcinoma	−0.03	0.00	0.03	0.08	0.06 *	0.08	0.11	0.09 *	0.22
Lung squamous cell carcinoma	−0.26	0.08 *	0.01	0.16	−0.24	0.07	0.03	0.10	0.10 *
Ovarian serous cystadenocarcinoma	0.08	0.04	0.06	0.05	0.06	−0.14	0.15 *	0.13	0.30
Pancreatic adenocarcinoma	0.00	0.05	−0.03	0.19	0.13	0.04	0.14 *	0.19	0.33
Pheochromocytoma and paraganglioma	0.39	0.35	0.34	0.32	0.16	0.25	0.39	0.21	0.29
Prostate adenocarcinoma	0.06	0.07	0.09	0.09	0.04 *	0.05	0.08	0.10	0.00 *
Rectum adenocarcinoma	−0.02	−0.11	−0.13	0.04	0.02	0.15	0.32	0.25	0.19
Sarcoma	0.40	0.21	0.39	0.30	0.30	0.12	0.32	0.23	0.24
Skin cutaneous melanoma	−0.05 *	0.01	−0.02	−0.08	−0.13 *	−0.18	0.09	−0.12 *	−0.02
Stomach adenocarcinoma	0.10	−0.01	0.11 *	0.06	−0.08	−0.06	0.19	0.27	0.25
Testicular germ cell tumors	0.27 *	0.41	0.31	0.16	0.15	−0.03	0.48	0.13	0.23 *
Thyroid carcinoma	0.26	0.17	0.22	0.29	0.17	0.07	0.25	0.01	0.29
Thymoma	0.08	−0.04	−0.27	0.10	−0.06	0.26	−0.09	0.14	0.03
Uterine corpus endometrial carcinoma	−0.04	0.01	−0.13	−0.28 *	−0.14	−0.17	−0.05	0.13	−0.03
Uterine carcinosarcoma	0.13	0.37 *	0.27	0.20	0.01	0.07	0.13	0.36	0.10

Red background—expression of CXCR2 ligand is positively correlated with the count of macrophages; blue background—CXCR2 ligand expression is negatively correlated with the count of macrophages; gray background—CXCR2 ligand expression is not significantly correlated with the count of macrophages; *—correlation analyzed by other algorithms indicated a different result.

**Table 18 ijms-24-13287-t018:** Correlation of CXCR2 ligand expression levels with the count of M1 macrophages in the tumor niche.

Name of the Cancer	*CXCL1*	*CXCL2*	*CXCL3*	*CXCL5*	*CXCL6*	*PPBP*	*CXCL8*	*CXCR1*	*CXCR2*
Adrenocortical carcinoma	0.01	0.09	−0.05	−0.18	−0.10	0.10	0.05	0.12	0.15
Bladder urothelial carcinoma	0.13	0.25	0.19	0.22	0.02	0.02	0.02 *	0.03	−0.31
Breast invasive carcinoma	0.15	−0.02	0.15	0.18	0.11	0.09	0.18	0.01	0.14
Cervical squamous cell carcinoma and endocervical adenocarcinoma	−0.25	−0.18	−0.27	−0.16	−0.15 *	−0.14	−0.31	−0.08	0.04
Cholangiocarcinoma	−0.27	−0.27	−0.09	−0.10	−0.14	0.24	−0.08	0.01	0.12
Colon adenocarcinoma	0.07	−0.03	−0.04	0.24	0.25	0.18	0.39	0.50	0.44
Lymphoid neoplasm diffuse large B-cell lymphoma	0.38	0.26	0.20 *	0.30	−0.01	−0.02	0.39 *	0.00	0.19
Esophageal carcinoma	−0.03	0.11	0.09	0.20	0.03	0.09	0.10	0.16	−0.15
Glioblastoma multiforme	0.25 *	0.18	0.33 *	0.42	0.36	0.29	0.33	0.16	0.30
Head and neck squamous cell carcinoma	−0.12 *	0.04	0.01	−0.05	−0.11 *	−0.12	−0.13 *	0.03	−0.11 *
Kidney chromophobe	0.31	0.36	0.40	0.15	−0.06	0.18	0.32	0.27 *	0.19
Kidney renal clear cell carcinoma	0.09	−0.07	0.05	0.16	0.05	−0.07	0.24	−0.01	0.09
Kidney renal papillary cell carcinoma	0.17	0.25	0.21	0.07	0.10	0.10	0.22	0.15	0.20
Acute myeloid leukemia	N/A	N/A	N/A	N/A	N/A	N/A	N/A	N/A	N/A
Brain lower grade glioma	0.02	0.09	0.12	−0.17 *	0.12	0.02	0.17	0.07	0.48
Liver hepatocellular carcinoma	0.23	0.02	0.18	0.18	0.16	−0.08	0.18	0.05	0.06
Lung adenocarcinoma	0.02	−0.06	0.04	0.08	0.12	−0.02	0.16	0.09	0.11
Lung squamous cell carcinoma	−0.26	0.05	−0.01	0.15	−0.23 *	0.02	−0.01	0.08	0.05
Ovarian serous cystadenocarcinoma	0.14	0.08	0.11	0.08	0.11	−0.10	0.14	0.13	0.31
Pancreatic adenocarcinoma	−0.04	−0.01	−0.09	0.13 *	0.08	−0.04	0.05	0.09	0.20
Pheochromocytoma and paraganglioma	0.46	0.41	0.39	0.34	0.21	0.29	0.41	0.27	0.32
Prostate adenocarcinoma	0.10	0.11	0.12	0.03	0.06	0.11	0.00	0.10	0.05
Rectum adenocarcinoma	−0.02	−0.04	−0.12	0.17	0.12	0.09	0.31	0.31	0.27
Sarcoma	0.41	0.28	0.41	0.30	0.32	0.11	0.32	0.22	0.29
Skin cutaneous melanoma	−0.08	0.02	−0.03	−0.05	−0.11	−0.15	0.07	−0.09	0.02
Stomach adenocarcinoma	0.09	−0.01	0.08	0.05	−0.05	−0.01	0.17	0.27	0.23
Testicular germ cell tumors	0.43	0.55	0.44	0.35	0.28	0.11	0.59	0.29	0.40
Thyroid carcinoma	0.17	0.16	0.17	0.11	0.14	−0.03	0.12	0.11	0.27
Thymoma	0.10	−0.02	−0.27	0.09	−0.03	0.19	−0.07	0.14	0.16
Uterine corpus endometrial carcinoma	−0.05	−0.06	−0.17	−0.32 *	−0.10	−0.16	−0.14	0.04	−0.13
Uterine carcinosarcoma	0.02	0.14	0.06	0.08	0.05	−0.04	−0.10	0.33	0.18

Red background—expression of CXCR2 ligand is positively correlated with the count of M1 macrophages; blue background—CXCR2 ligand expression is negatively correlated with the count of M1 macrophages; gray background—CXCR2 ligand expression is not significantly correlated with the count of M1 macrophages; *—correlation analyzed by other algorithms indicated a different result.

**Table 19 ijms-24-13287-t019:** Correlation of CXCR2 ligand expression levels with the count of M2 macrophages in the tumor niche.

Name of the Cancer	*CXCL1*	*CXCL2*	*CXCL3*	*CXCL5*	*CXCL6*	*PPBP*	*CXCL8*	*CXCR1*	*CXCR2*
Adrenocortical carcinoma	0.00	0.13	0.05	0.18	−0.10	0.20	−0.05	0.00	0.12
Bladder urothelial carcinoma	−0.11	0.09	0.06	0.06	−0.12	−0.01	−0.20	−0.04	−0.15 *
Breast invasive carcinoma	−0.16	−0.10	−0.13	−0.16	−0.16	0.04	0.02	0.15	0.31
Cervical squamous cell carcinoma and endocervical adenocarcinoma	−0.24	−0.20	−0.42	−0.19	−0.18	−0.13	−0.39	−0.11	0.02
Cholangiocarcinoma	−0.31	−0.53	−0.20	−0.33	−0.21	−0.13	−0.24	−0.36 *	−0.22
Colon adenocarcinoma	0.00 *	−0.06	−0.41	0.03	0.03	0.13 *	0.12	0.16	0.12
Lymphoid neoplasm diffuse large B-cell lymphoma	0.11	0.05	0.04	0.21	0.07	−0.02	0.18	−0.06	0.02
Esophageal carcinoma	−0.01 *	0.23 *	−0.27	0.15	−0.08	0.14	−0.05	0.18 *	−0.12
Glioblastoma multiforme	−0.02	−0.05	−0.14	0.00	0.04	−0.03	−0.10	−0.03	0.23
Head and neck squamous cell carcinoma	−0.18	−0.07	−0.13	−0.12	−0.17	−0.09 *	−0.20	−0.04	−0.06
Kidney chromophobe	0.21	0.25	0.23	0.18	−0.07	0.22	0.17	0.30	0.11
Kidney renal clear cell carcinoma	0.03	−0.07	−0.03	0.00	−0.01	−0.05	0.11	0.01	0.12
Kidney renal papillary cell carcinoma	0.09	0.17	0.03	0.01	0.04	0.02	0.07	0.08	0.08
Acute myeloid leukemia	N/A	N/A	N/A	N/A	N/A	N/A	N/A	N/A	N/A
Brain lower grade glioma	0.07	−0.05	−0.18	0.01	−0.01	0.08	0.09	0.06	0.17
Liver hepatocellular carcinoma	−0.10	0.16	−0.17	−0.19	−0.05	0.02	−0.12 *	0.01	0.03
Lung adenocarcinoma	−0.15	0.06	−0.02	0.05	−0.10	0.19	−0.11	0.13	0.26
Lung squamous cell carcinoma	−0.27	0.13	−0.03	0.06	−0.24	0.12	−0.02	0.11	0.15
Ovarian serous cystadenocarcinoma	−0.14	−0.09	−0.11	−0.13 *	−0.12	−0.18	−0.03	0.02	0.23
Pancreatic adenocarcinoma	0.00	0.04	−0.03	0.16	0.04	0.04	0.05	0.27	0.39
Pheochromocytoma and paraganglioma	0.12	0.21	0.19	0.01	0.00	0.14	0.19	0.11	0.17
Prostate adenocarcinoma	−0.06	−0.01	−0.05	−0.08	−0.10 *	0.04	−0.05	0.00	−0.12
Rectum adenocarcinoma	−0.09	−0.23	−0.18	−0.11	−0.16	0.07	0.14	0.08	0.00
Sarcoma	0.15	0.19	0.09	0.04	0.04	−0.02	0.09 *	0.23	0.15
Skin cutaneous melanoma	−0.13	−0.17	−0.17	−0.25	−0.19 *	−0.16	−0.19	−0.12 *	−0.09
Stomach adenocarcinoma	0.02 *	−0.03 *	0.08 *	0.09	−0.09 *	−0.14	0.07	0.26	0.24
Testicular germ cell tumors	0.03	0.23	0.22	0.02 *	−0.04	−0.06	0.24	0.01 *	0.02 *
Thyroid carcinoma	−0.21	−0.21	−0.19	−0.17	−0.06	−0.18	−0.16	−0.07	−0.05
Thymoma	0.21	0.01	−0.31	0.15	0.06	0.22	−0.02	0.05 *	0.01 *
Uterine corpus endometrial carcinoma	−0.10	−0.09	−0.21	−0.31	−0.08	−0.13	−0.15	0.03	−0.06
Uterine carcinosarcoma	−0.23	0.00	−0.03	−0.11	−0.12	−0.06	−0.12	0.04	−0.10

Red background—expression of CXCR2 ligand is positively correlated with the count of M2 macrophages; blue background—CXCR2 ligand expression is negatively correlated with the count of M2 macrophages; gray background—CXCR2 ligand expression is not significantly correlated with the count of M2 macrophages; *—correlation analyzed by other algorithms indicated a different result.

**Table 20 ijms-24-13287-t020:** Frequency of mutations in CXCR2 ligand genes in selected cancers.

Name of the Cancer	Number of Cases Studied	*CXCL1*	*CXCL2*	*CXCL3*	*CXCL5*	*CXCL6*	*PPBP*	*CXCL8*	*CXCR1*	*CXCR2*
Adrenocortical carcinoma	92	1.1% T	1.1% A	1.1% A	1.1% A	-	1.1% A	-	-	-
Bladder urothelial carcinoma	411	1.9% A	1.2% A	1.2% A	1.5% A	1.9% A	1.7% A	2.2% A	1.2% D	1.2% D
Breast invasive carcinoma	1084	1.9% A	2.1% A	2.2% A	2.2% A	1.9% A	2.2% A	1.9% A	0.37% A 0.46% D	0.37% A 0.46% D
Cervical squamous cell carcinoma and endocervical adenocarcinoma	297	0.67% A 0.34% D	0.34% A 0.34% D	0.67% A 0.34% D	0.67% A 0.34% D	0.67% A 0.34% D	0.67% A 0.34% D	0.67% A 0.34% D 0.34% T	2.0% D	2.0% D
Cholangiocarcinoma	36	-	-	-	-	-	-	-	-	-
Colon adenocarcinoma	378	0.26% A 0.26% D	0.26% A 0.26% D 0.26% T	0.53% A 0.26% D	0.53% A 0.26% D	0.26% A 0.26% D	0.53% A 0.26% D	0.26% A 0.26% D	-	-
Lymphoid neoplasm diffuse large B-cell lymphoma	48	-	-	-	-	-	-	-	-	-
Esophageal carcinoma	182	2.7% A	2.7% A	3.3% A	3.3% A	2.7% A	3.3% A	2.7% A	0.55% D	0.55% A 0.55% D 1.6% T
Glioblastoma multiforme	585	0.17% A	0.17% A	0.17% A	0.17% A	0.17% A	0.17% A	0.17% A	0.17% A 0.17% D	0.17% A 0.17% D
Head and neck squamous cell carcinoma	523	0.76% A 0.19% T 0.19% CXCL1-AFP	0.57% A	0.57% A	0.57% A	0.76% A	0.57% A	0.76% A	1.3% D 0.19% T	1.3% D
Kidney chromophobe	65	-	-	-	-	-	-	-	-	-
Kidney renal clear cell carcinoma	512	0.39% A	0.20% A	0.20% A	0.20% A	0.39% A	0.20% A	0.39% A	0.20% A 0.20% D	0.20% A 0.20% D
Kidney renal papillary cell carcinoma	283	-	-	-	-	-	-	-	1.1% D	1.1% D
Acute myeloid leukemia	200	-	-	-	-	0.50% A	-	0.50% A	-	-
Brain lower grade glioma	514	0.19% D	0.19% D	0.19% D	0.19% D	0.19% D	0.19% D	0.19% D	0.19% A 0.19% D	0.19% A 0.19% D
Liver hepatocellular carcinoma	372	0.27% A 0.27% D	0.27% D	0.27% D	0.27% D	0.27% A 0.27% D	0.27% D	0.27% A 0.27% D	0.81% A	0.54% A
Lung adenocarcinoma	566	0.88% A	0.88% A 0.18% T	0.88% A	0.88% A	0.88% A	0.88% A 0.18% T	0.88% A	0.35% A	0.35% A 0.35% T
Lung squamous cell carcinoma	487	2.3% A	2.3% A	2.3% A	2.3% A	2.3% A	2.3% A	2.3% A	0.21% A 0.62% D 0.21% T	0.21% A 0.62% D
Ovarian serous cystadenocarcinoma	585	2.6% A 0.34% D	2.6% A 0.34% D	2.6% A 0.34% D	2.6% A 0.34% D	2.6% A 0.34% D	2.6% A 0.34% D	2.4% A 0.51% D	0.68% A 0.17% T	0.68% A
Pancreatic adenocarcinoma	184	0.54% A	0.54% A	0.54% A	0.54% A	0.54% A	0.54% A	0.54% A	1.1% A	1.1% A
Pheochromocytoma and Paraganglioma	178	0.56% A	0.56% A	1.1% A	0.56% A	0.56% A	0.56% A	0.56% A	-	-
Prostate adenocarcinoma	494	0.61% A 0.20% D	0.61% A 0.20% D	0.81% A 0.20% D	0.81% A 0.20% D	0.61% A 0.20% D	0.81% A 0.20% D	0.61% A 0.20% D	0.61% A	0.61% A
Rectum adenocarcinoma	155	-	-	-	-	-	-	-	-	-
Sarcoma	255	1.2% A 0.39% D	1.2% A 0.39% D	1.2% A 0.39% D	1.2% A 0.39% D	1.2% A 0.39% D	1.2% A 0.39% D	1.2% A 0.39% D 0.39% T	1.6% A 1.2% D	1.6% A 1.2% D
Skin cutaneous melanoma	442	0.23% A 0.23% D	0.23% A 0.23% D	0.23% A 0.23% D	0.23% A 0.23% D	0.23% A 0.23% D	0.23% A 0.23% D	0.23% A 0.23% D 0.23% T	0.23% A 0.23% T	0.23% A
Stomach adenocarcinoma	440	1.6% A	1.8% A	1.8% A	1.8% A	1.6% A	1.8% A	1.6% A	0.68% A	0.45% A
Testicular germ cell tumors	149	-	-	0.67% D	-	-	-	-	-	-
Thyroid carcinoma	500	-	-	-	-	-	-	-	-	-
Thymoma	123	-	-	-	-	-	-	-	-	-
Uterine corpus endometrial carcinoma	529	0.19% T	0.38% T	-	-	-	-	0.19% A 0.19% D 0.57% T	1.1% A 0.19% D 0.19% T	1.1% A 0.19% D 0.19% T
Uterine carcinosarcoma	57	-	-	-	-	-	-	-	-	-
Pan-cancer analysis	10,726	0.85% A 0.08% D 0.03% T	0.83% A 0.08% D 0.04% T	0.88% A 0.09% D	0.88% A 0.08% D 0.01% T	0.86% A 0.08% D	0.89% A 0.08% D 0.01% T	0.87% A 0.10% D 0.06% T	0.33% A 0.34% D 0.05% T	0.33% A 0.34% D 0.06% T

A—amplification; D—deletion; T—truncating mutation; -—no mutations found in the test sample; red background—more than 1.5% of tumor cases have amplifications of the gene for a particular CXCR2 ligand; blue background—more than 1.5% of tumor cases have a deletion of the gene for a particular CXCR2 ligand or receptor.

**Table 21 ijms-24-13287-t021:** Potential transcription factors regulating the expression of CXCR2 ligands.

Ligand CXCR2 Gene	Up to 200 bp Upstream of the Transcription Start Site	From 200 bp to 500 bp Upstream of the Transcription Start Site	500 to 1500 bp Upstream of the Transcription Start Site
*CXCL1*	AP-2 (−59), NF-κB (−77), Sp1 (−128), AP-2 (−133), TEF2-GT-I/EGR-1 (−163),	TRF-site-C (TREF1/2) (−265), CuE4.1 (−296), E1A (−299), ASP-CYP21/EGR-1 (−387), Fli-1-Ets-site-1 (−411), PEA-3 (−412), NF-E1 (−475),	FREAC-2/EGFR-downstream-enh (−659), ??(a) (−664), GATA-1 (−707), NF-E1.2 (−769), PEA-3 (−773), NF-IL6 (−803), C/EBPa (−817), M-Box (−953), TEF1 (−1057), Ets1 (−1076), NF-IL3 site (−1207), TFII-I (−1262), C/EBP (−1278), tPA-GC-box-I (−1292), E-box/NF-kappaE1 site (−1413)
*CXCL2*	c-Myb (−76), NF-κB (−77), pyrimidine-rich domain (−112), IRE-A/DSE/TRE (−127), ZIP-site (Sp1/EGR-1) (−137),	Sp1 (−252), epsilon-NRA-FP2 (−267), UIRR-GATA (−268), MT-I.4 (−318), (−474),	X2 (−535), poly CACA sequence (−612–−636), Wt1 (−624–−636), microE5 motif (−658), HNF-3 (−709), Sox2-POU-site-1 (−1019), NMP-2 (−1040), NFE4 site (−1186), NF-4FA (−1193), HoxA-5 (−1197), RIPE3b (−1392)
*CXCL3*	NF-κB (−77), Sp1 (−127), c-GC-box (−128), ZIP-site (Sp1/EGR-1) (−138),	epsilon-NRA-FP2 (−281), PEA-3 (−401), (−406), Ets-1-Octa-1 (−433),	FREAC-2/EGFR-downstream-enh (−669), ??(a) (−675), Fli-1-Ets-site-1/GATA-1 (−788), TIN-1 (NRE)/IL-1 response_eleme (−810), Tbx2 (−841), C/EBP-AT-Site-A (−968)
*CXCL5*	MZF-1 (−51), NF-AT (−63), TTF (−67), NF-κB (−89), C/EBP (−113), U-prosaposin (−130), H4TF1/IKAROS/LYF-1 (−132), Sp1 (−133), XRE (−133),	TFII-I (−255), mTDT-site D (−268), mTDT-site D (−278), GATA-1 (−350), NF-E4 site (−359), C/EBP (−421), NF-κB/ISGF1 (−495),	PEA-3 (−512), TATA box (−552), Ets-1-Octa-1 (−782), GATA-1 (−961), ?? (−1051), Stat3 (−1102), ??(a) (−1279), STAT5B (−1345), C/EBPa (−1349), Optimedin-Opts2 (Pax6)(−1408)
*CXCL6*	Kaiso-box (−56), NF-AT (−62), TTF (−66), NF-κB (−88), C/EBP (−112), ADA-NF2 (−132), Sp1 (−137), Ets1 (−144), E2A (−150),	Sp1 (−215), c-GC-box (−216), mTDT-site D (−267), AP-2 (−284), GATA-1 (−338), HoxTF-Hoxb-4 (NFY/YY1 motif) (−354), PEA-3 (−381), BAR1.2 (−438), C/EBP (−440),	Optimedin-Opts2 (Pax6) (−509), AP-1 (−514), DBP (−527), Optimedin-Opts2 (Pax6) (−628), AP-1 (−633), DBP (−646), C/EBP (−870), HNF-6 (−881), beta-RARE (−988), oct-element N1 (−1171), kappaY (Oct-1/2) (−1306)
*PPBP*	?? (−42), PEA-3 (−61), PU.1/Fli-1-Ets-site-1 (−78), EF-1A-site A (−81), PuF site (−128), ?? (−134),	NF-E1.2 (−217), PEA-3 (−221), kappaY (Oct-1/2) (−222), TBP (−323), CuE3.1 (−381), ISGF2 (−396), Ncx (−496),	epsilon-NRA-FP5 (−557), Ets-1 (−599), (−756), NF-E1.2 (−980), T3RE (−989), HBP1 (−1014), AGP-HA (−1123), Brn-3 (−1404)
*CXCL8*	C/EBP (−77), IL-8-NRE (NF-κB)/CSBP-1 (−87), NF-κB/NF-AT (−91), Oct-1 (−104), NF-E2-consensus (−136), AP-1 (−137),	Oct3/MEF2-like-sequence (−232), Fbx15-oct-site (−296), E-box (Hes-1) (−334), microE5 motif (−345), NF-E1.2 (−422), ERK2-CAAT-box (NF-Y/CBF) (−433), XOR-TATA-like element (NRE) (TFIID) (−440), Bcl-6 (NRE)/MEF2-like-sequence (−495),	Gfi-1 (NRE) (−551), Gfi-1 (NRE) (−610), PEA-3 (−841), TEF1 (−863), SF-1 (−886), ?? (−893), Tbx2/EGR-1 (−959), AP-2/Sp1 (−966), GATA-1 (−1409)

??—an unknown protein that binds to a sequence important in the regulation of expression; ??(a)—a binding site classified as essential for pro alpha 1 (I) collagen expression; CSBP-1—conserved sequence-binding protein 1; DBP—D-box binding PAR bZIP transcription factor; EGR-1—early growth response-1; H4TF1—histone H4 gene-specific transcription factor-1; HNF-3—hepatocyte nuclear factor 3; IRE A—insulin response element A; FREAC-2—forkhead-related activator 2; kappaY—high pyrimidine content motif; MZF-1—myeloid zinc finger 1; NF-AT—nuclear factor of activated T cells; NMP-2—nuclear matrix protein 2; NRE—negative regulatory element; Pax-8—paired box-containing 8; RARE—retinoic acid response element; RIPE3b—rat insulin enhancer-binding complex 3b2; UIRR—upstream interferon response region; X2—TNF-alpha enhancement of IFN-gamma-induced promoter activity; XRE—xenobiotic response element; T3RE—triiodothyronine (T3) response element; TRE—thyroid hormone response element; TBP—TATA-binding proteins; Tbx2—T-box 2; TIN-1—testis-specific factor TIN-1 (transcription inhibition); TTF—thyroid transcription factor; Wt1—Wilms tumor wt1 protein; ZIP site—site for the zinc finger proteins (Sp1 and EGR-1).

**Table 22 ijms-24-13287-t022:** CXCR2 ligands as a target for microRNAs with a target score between 85–100 for one of the CXCR2 ligands.

miRNA	Target Score for *CXCL1*	Target Score for *CXCL2*	Target Score for *CXCL3*	Target Score for *CXCL5*	Target Score for *CXCL6*	Target Score for *PPBP*	Target Score for *CXCL8*	Number of CXCR2 Ligands as a Target
miR-7-1-3p	<50	87	<50	63	<50	<50	<50	2
miR-7-2-3p	<50	87	<50	63	<50	<50	<50	2
let-7c-3p	<50	<50	93	<50	<50	<50	<50	1
miR-17-5p	<50	<50	<50	<50	88	<50	<50	1
miR-20a-5p	<50	<50	<50	<50	90	<50	<50	1
miR-20b-5p	<50	<50	<50	<50	90	<50	<50	1
miR-25-3p	<50	<50	<50	90	<50	<50	<50	1
miR-32-5p	<50	<50	<50	90	<50	<50	<50	1
miR-92a-3p	<50	<50	<50	90	<50	<50	<50	1
miR-92b-3p	<50	<50	<50	90	<50	<50	<50	1
miR-93-5p	<50	<50	<50	<50	88	<50	<50	1
miR-95-5p	87	99	<50	<50	<50	<50	<50	2
miR-106a-5p	<50	<50	<50	<50	88	<50	<50	1
miR-106b-5p	<50	<50	<50	<50	88	<50	<50	1
miR-140-3p	<50	<50	<50	<50	<50	<50	94	1
miR-153-5p	<50	<50	<50	<50	<50	<50	89	1
miR-190a-3p	<50	<50	<50	97	54	<50	<50	2
miR-192-5p	<50	91	<50	<50	<50	<50	<50	1
miR-194-5p	<50	<50	86	<50	<50	<50	<50	1
miR-215-5p	<50	91	<50	<50	<50	<50	<50	1
miR-302a-3p	86	<50	<50	<50	<50	<50	<50	1
miR-302b-3p	86	<50	<50	<50	<50	<50	<50	1
miR-302c-3p	86	<50	<50	<50	<50	<50	<50	1
miR-302d-3p	86	<50	<50	<50	<50	<50	<50	1
miR-302e	86	<50	<50	<50	<50	<50	<50	1
miR-335-3p	<50	<50	86	<50	<50	<50	<50	1
miR-363-3p	<50	<50	<50	90	<50	<50	<50	1
miR-367-3p	<50	<50	<50	90	<50	<50	<50	1
miR-376c-3p	51	<50	<50	85	<50	<50	<50	2
miR-380-3p	<50	<50	<50	85	<50	<50	<50	1
miR-466	<50	86	<50	<50	<50	<50	<50	1
miR-495-3p	<50	87	<50	51	<50	<50	<50	2
miR-519d-3p	<50	<50	<50	<50	90	<50	<50	1
miR-526b-3p	<50	<50	<50	<50	90	<50	<50	1
miR-532-5p	92	86	<50	<50	<50	<50	<50	2
miR-548p	<50	<50	<50	85	<50	<50	<50	1
miR-548t-3p	<50	<50	<50	<50	99	<50	<50	1
miR-548aa	<50	<50	<50	<50	99	<50	<50	1
miR-548ah-5p	<50	<50	<50	94	<50	<50	<50	1
miR-548ap-3p	<50	<50	<50	<50	99	<50	<50	1
miR-548at-5p	<50	<50	<50	<50	<50	<50	92	1
miR-548l	<50	<50	86	<50	<50	<50	<50	1
miR-548n	<50	<50	64	87	<50	<50	<50	2
miR-548o-3p	86	<50	<50	<50	70	<50	<50	2
miR-570-3p	96	68	<50	<50	<50	<50	<50	2
miR-629-5p	<50	<50	<50	<50	<50	88	<50	1
miR-642a-3p	<50	<50	92	56	<50	<50	<50	2
miR-656-3p	<50	<50	<50	93	<50	<50	<50	1
miR-889-3p	<50	<50	<50	89	<50	<50	76	2
miR-1266-3p	<50	92	<50	<50	<50	<50	<50	1
miR-1277-5p	<50	<50	<50	100	51	51	<50	3
miR-1323	87	<50	<50	<50	73	<50	<50	2
miR-2115-3p	<50	<50	<50	<50	<50	<50	85	1
miR-3123	<50	63	<50	<50	87	<50	<50	2
miR-3140-5p	<50	<50	<50	<50	<50	91	<50	1
miR-3148	<50	<50	<50	57	91	<50	<50	2
miR-3152-3p	<50	<50	<50	<50	<50	87	<50	1
miR-3163	<50	<50	65	89	<50	<50	68	3
miR-3606-5p	<50	<50	<50	<50	86	<50	<50	1
miR-3609	<50	<50	<50	93	<50	<50	<50	1
miR-3671	<50	<50	<50	<50	75	<50	92	2
miR-4291	<50	<50	90	<50	<50	<50	<50	1
miR-4312	<50	<50	<50	<50	<50	<50	85	1
miR-4436b-5p	<50	<50	<50	<50	<50	<50	87	1
miR-4524a-3p	<50	<50	<50	<50	<50	88	<50	1
miR-4687-5p	<50	<50	<50	88	<50	<50	<50	1
miR-4699-3p	<50	<50	<50	<50	<50	<50	88	1
miR-4753-3p	<50	<50	95	86	<50	<50	<50	2
miR-4776-3p	<50	<50	<50	99	<50	<50	<50	1
miR-4782-5p	<50	<50	<50	<50	<50	<50	85	1
miR-4789-3p	<50	87	79	<50	<50	<50	<50	2
miR-4795-3p	<50	<50	65	<50	<50	85	<50	2
miR-5009-3p	<50	<50	<50	<50	<50	94	<50	1
miR-5011-5p	<50	<50	<50	100	75	<50	<50	2
miR-5582-3p	<50	<50	90	<50	<50	<50	51	2
miR-5584-3p	88	73	<50	<50	<50	<50	<50	2
miR-5688	<50	87	<50	52	<50	<50	<50	2
miR-5692a	78	89	87	<50	<50	<50	100	4
miR-5692b	<50	<50	<50	99	<50	<50	52	2
miR-5692c	<50	<50	<50	99	<50	<50	52	2
miR-5706	<50	<50	<50	<50	<50	<50	85	1
miR-6074	<50	<50	<50	76	90	67	<50	3
miR-6853-3p	<50	<50	<50	89	<50	<50	<50	1
miR-6882-5p	<50	<50	<50	87	<50	<50	<50	1
miR-7161-5p	<50	<50	<50	87	<50	<50	<50	1
miR-8080	<50	<50	<50	87	<50	<50	<50	1
miR-12136	<50	<50	85	<50	<50	60	<50	2

Blue background—target score for the selected microRNA and CXCR2 ligand is above 50; yellow background—there are two relationships between a given microRNA and CXCR2 ligands with a target score above 50; bright orange background—there are three relationships between a given microRNA and CXCR2 ligands with a target score above 50; dark orange background—there are four relationships between a given microRNA and CXCR2 ligands with a target score above 50.

**Table 23 ijms-24-13287-t023:** The count of patients with a given cancer and the count of control groups for a given type of cancer.

Name of the Cancer	Abbreviation	Size of the Control Group	Number of Patients
Adrenocortical carcinoma	ACC	128	77
Bladder urothelial carcinoma	BLCA	28	404
Breast invasive carcinoma	BRCA	291	1085
Cervical squamous cell carcinoma and endocervical adenocarcinoma	CESC	13	306
Cholangiocarcinoma	CHOL	9	36
Colon adenocarcinoma	COAD	349	275
Lymphoid neoplasm diffuse large B-cell lymphoma	DLBC	337	47
Esophageal carcinoma	ESCA	286	182
Glioblastoma multiforme	GBM	207	163
Head and neck squamous cell carcinoma	HNSC	44	519
Kidney chromophobe	KICH	53	66
Kidney renal clear cell carcinoma	KIRC	100	523
Kidney renal papillary cell carcinoma	KIRP	60	286
Acute myeloid leukemia	LAML	70	173
Brain lower grade glioma	LGG	207	518
Liver hepatocellular carcinoma	LIHC	160	369
Lung adenocarcinoma	LUAD	347	483
Lung squamous cell carcinoma	LUSC	338	486
Ovarian serous cystadenocarcinoma	OV	88	426
Pancreatic adenocarcinoma	PAAD	171	179
Pheochromocytoma and Paraganglioma	PCPG	3	182
Prostate adenocarcinoma	PRAD	152	492
Rectum adenocarcinoma	READ	318	92
Sarcoma	SARC	2	262
Skin cutaneous melanoma	SKCM	558	461
Stomach adenocarcinoma	STAD	211	408
Testicular germ cell tumors	TGCT	165	137
Thyroid carcinoma	THCA	337	512
Thymoma	THYM	339	118
Uterine corpus endometrial carcinoma	UCEC	91	174
Uterine carcinosarcoma	UCS	78	57

## Data Availability

Not applicable.

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
