# Peer review of "Bioinformatic Analysis of the CXCR2 Ligands in Cancer Processes"

_ijms, 2023, doi:10.3390/ijms241713287_

Round 1

Reviewer 1 Report

A brief summary

In this study, the authors performed a bioinformatic analysis using the GEPIA, UALCAN, and TIMER2.0 databases to investigate the role of CXCR2 ligands in 31 different types of cancer. They found that the effect of CXCR2 ligands on prognosis depends on the type of cancer, and CXCR2 ligands are associated with EMT, angiogenesis, and recruiting neutrophils to the tumor microenvironment.

Specific comments

  1. The words including tumor, cancer, and miRNA are better removed from the keyword section.
  2. Based on which database did you report the increase or decrease in the expression of the selected genes. GEPIA, UALCAN, or common between both?
  3. The title of 2.3 (in some tumors, the expression of only one out of the 7 CXCR2 ligands positively correlated with deteriorated lymph node metastasis status) is better rewritten.
  4. I think that some of the manuscripts were written by AI (artificial intelligence) tools. If I am correct, it is better to rewrite those sections.

-

Author Response

Rev 1

  1. The words including tumor, cancer, and miRNA are better removed from the keyword section.

These three keywords have been removed from the keywords list. "Gro-alpha" has been added. This is another name for CXCL1.

  1. Based on which database did you report the increase or decrease in the expression of the selected genes. GEPIA, UALCAN, or common between both?

As stated in the methodology, the expression relationship of the CXCR2 ligand between healthy tissue and tumor was analyzed solely on the GEPIA portal. Unlike UALCAN, GEPIA additionally incorporates control values from the GTEx database. This is why it is considerably superior to other portals for comparing healthy tissue to a tumor.

  1. The title of 2.3 (in some tumors, the expression of only one out of the 7 CXCR2 ligands positively correlated with deteriorated lymph node metastasis status) is better rewritten.

The section title has been changed to: "Only in 5 out of 20 types of cancers, certain CXCR2 ligands may positively correlate with lymph node metastasis status."

  1. I think that some of the manuscripts were written by AI (artificial intelligence) tools. If I am correct, it is better to rewrite those sections.

The article was entirely written in Polish without the involvement of AI, then translated into English. Data acquisition from various portals was also conducted without the use of AI or any algorithms.

Reviewer 2 Report

Dear Authors, 

The manuscript by Korbecki et al includes an interesting approach unravelling the CXCR2 ligands and their actions in different functions and tumors.

The manuscript is written in excellent English which provides an advantage to the manuscript as it is quite extensive and overwhelmed with information.

The manuscript is structured in two large parts, the first is the bioinformatic analysis and the 2nd one is the discussion part which is like a mini review for the CXCR2 ligands.

As I mentioned the information included in the article are valuable.

My sole recommendation is regarding the presentation matter of the manuscript as the involved tables do not help the manuscript.

My suggestion will be to present the abbreviation of the tumors only once, and swipe the order of the presented data and the tables, meaning that the abbreviations of tumors will be as columns and the CXCR2 ligands as rows, if space is enough for the columns.

ACC

BLCA

BRCA

CESC

CXCL1

CXCL2

CXCL3

Wish you the best.

Author Response

Rev 2

My sole recommendation is regarding the presentation matter of the manuscript as the involved tables do not help the manuscript.

My suggestion will be to present the abbreviation of the tumors only once, and swipe the order of the presented data and the tables, meaning that the abbreviations of tumors will be as columns and the CXCR2 ligands as rows, if space is enough for the columns.

The article contains very large tables with dimensions of 10x32. We agree with the reviewer that such large tables can make data presentation difficult. However, when we performed the transposition of one table, it became too large to fit on a single sheet of paper. The sheet is oriented vertically, meaning more cells need to be represented as rows. We can carry out the transposition of the table and change the page orientation from vertical to horizontal. However, as a result of these two actions, we would obtain the same table with a different page layout. Most likely, during the final formatting stage of the article by the Editorial team, the tables will be arranged in such a way that each table occupies only one page.

Regarding the cancer abbreviations, I personally specialize in two types of cancers. For me, the abbreviations for these two types of cancers are clear and understandable, and I don't need to decode them in my work because I work with these abbreviations/types of cancers on a daily basis. The remaining 29 abbreviations are unfamiliar to me, and when I encounter them, I require explanations for the abbreviations. I believe that readers will likely face the same issue. Therefore, we have removed the cancer abbreviations from the tables while retaining the full names of the cancer types.

Reviewer 3 Report

In this manuscript by Korbecki et al, the authors have performed an extensive analyses of CXCR2 ligand expression patterns in a multitude of different tumors, plus assessed the promoter sequences of the genes encoding each legand for potential transcriptional regulator binding sites, compared protein structures, and identified potential miRNA regulatory sequences. The tables reviewing differences in expression patterns in different tumors, either increased or decreased vs. normal tissues, is impressive and comprehensive. The authors do an excellent job describing the necessary functions of the ligands and how their binding to CXCR2 activates signaling pathways. They also review correlations of ligand expression with a variety of markers, including proliferation markers and three types indicating EMT. These informative tables are followed by a series of correlations between ligand expression and a variety of cell types associated with CXCR2 ligand binding. The descriptions are extensive, and the in-depth discussion provides their interpretations of the correlations, which are interesting, albeit a bit lengthy. While there are some suggestions, in particular to shortening the manuscript as its length is arguably overwhelming, the work reveals very interesting correlations of expression patterns of the different CXCR2 ligands. The work adds to our knowledge of how CXCR2 ligand expression levels may play an important role in the tumor microenvironment or indicative of cancer progression and/or likelihood of leading to metastasis.

Introduction, since the bulk of the work is to compare the expression levels of multiple CXCR2 ligands, each of which have unique properties with respect to activating CXCR2 (among other receptors) and the resulting functions caused by the ligands, a table that summarizes the information provided in the introductory remarks (that includes both tissues of expression and functions activated) would be helpful while interpreting the abundant gene expression data.

Section 2.2, lines 170-177, the authors mention that the levels of ligand expression "affected the survival of patients....." and "effect on prognosis", but these are not data that indicate a cause-and-effect function - the survival might be associated with increased expression, but we cannot conclude that this actually affected the outcome of the cancer. Suggest rewording to be clear as to associations that can be concluded, vs. an actual cause of the tumor progression/survival. Also, the data does not indicate anti-tumor or pro-tumor effects, but its safe to write that it "suggests" – “indicate” is too strong.

Section 2.2, can the authors provide a comparison of tumor types with different expression (e.g., elevated) vs. prognosis? This would facilitate interpretation of the data.

Section 2.8, lines 297-299, the two sentences here might be combined, as they are too similar to easily distinguish - there was positive or negative correlation of the CXCR2 ligands with MDSC counts, so just state as such.

Section 2.17, the authors might provide a transition statement that introduces this comparison of CXCR2 ligand sequences - why is this presented? This can be brief, but as is, this is a bit confusing given all the expression pattern comparisons with tumors above, and then mechanisms regulating ligand gene expression in the subsequent section (the reviewer notes the comment about protein structure vs. mechanisms regulating gene expression in 2.18, which is excellent).

Section 3.1, can the authors provide any brief description, or at least examples, of how differences in gene regulators that control expression of the ligands correlate with differences in ligand gene regulation? How does expression of NF-kB or other regulators identified differ, which might explain the differences in ligand gene expression in the studied tumors?

Section 3.7 and throughout the Discussion, there are significant redundancies in the text regarding cell type functions and the correlations that are well-described in the Results section, which could be removed or shortened. For example, in lines 907-911, the authors do not need to re-explain M1 vs. M2 functions. With the already lengthy Discussion and abundant Results, some editing of redundancies would help to shorten the length of the manuscript.

Section 3.12, lines 1139-1155, again the properties of M1 vs. M2 cells are reviewed, which is unnecessary, so suggest removal or at least brevity while discussing the importance of these two cell types.

The quality of English is good, albeit there are some sentences that could be tightened or shortened.

Author Response

Rev3

Introduction, since the bulk of the work is to compare the expression levels of multiple CXCR2 ligands, each of which have unique properties with respect to activating CXCR2 (among other receptors) and the resulting functions caused by the ligands, a table that summarizes the information provided in the introductory remarks (that includes both tissues of expression and functions activated) would be helpful while interpreting the abundant gene expression data.

A table illustrating the location of the highest expression of a given ligand has been included. Regarding other functions, in experimental studies, not all CXCR2 ligands are examined, which is why knowledge about differences between them is currently quite limited and unsatisfactory.

Table 1. Location of the highest expression of each CXCR2 ligand.

Ligand CXCR2

Location of the highest expression

CXCL1

Appendix, bone marrow, gall bladder, small intestine, urinary bladder

CXCL2

Appendix, bone marrow, gall bladder, liver, lung

CXCL3

Appendix, bone marrow, colon, gall bladder, liver, lung, stomach, urinary bladder

CXCL5

Appendix, gall bladder, lung, lymph node, stomach, urinary bladder

CXCL6

Appendix, gall bladder, urinary bladder

PPBP

Bone marrow, spleen

CXCL8

Appendix, bone marrow, esophagus, gall bladder, liver, urinary bladder

Section 2.2, lines 170-177, the authors mention that the levels of ligand expression "affected the survival of patients....." and "effect on prognosis", but these are not data that indicate a cause-and-effect function - the survival might be associated with increased expression, but we cannot conclude that this actually affected the outcome of the cancer. Suggest rewording to be clear as to associations that can be concluded, vs. an actual cause of the tumor progression/survival. Also, the data does not indicate anti-tumor or pro-tumor effects, but its safe to write that it "suggests" – “indicate” is too strong.

We fully agree with the reviewer. The correlation between expression and prognosis is not a causal relationship but rather an indication of some potential association. As a result, the indicated section has been revised accordingly.

Correlation of CXCR2 ligand expression with prognosis

The correlation between CXCR2 ligand expression within tumors and patient prognosis was examined across 31 cancer types. However, it is essential to emphasize that this analysis only reflects correlation, which might indicate either anti-tumor or pro-tumor properties of CXCR2 ligands. It could also imply alternative associations, such as improved prognosis leading to distinct CXCR2 ligand expression, without the ligands participating in tumorigenesis. It is important to note that even the identification of correlation might not necessarily imply a causal relationship.  //   The analysis of 31 different types of cancer  revealed that in eight cancer types, the expression levels of more than one CXCR2 ligand are correlated with prognosis. Within almost all of these types, the correlation is either positive or negative exclusively. The exception is brain lower-grade glioma, where higher CXCL1 expression correlates with worse prognosis, while CXCL2 and CXCL5 expression correlates with better outcomes. The most frequent occurrence (7 out of 31) is the correlation of high CXCL8 expression with poorer prognoses. Conversely, the most commonly correlated (4 out of 31) with better prognoses is CXCL2 (Table 3).

Section 2.2, can the authors provide a comparison of tumor types with different expression (e.g., elevated) vs. prognosis? This would facilitate interpretation of the data.

Thirty-one tables have been added to the supplement, organized based on individual cancer types. We believe that readers would not only want to compare expression levels within tumors and the correlation of CXCR2 ligand expression with prognosis but also explore other aspects of tumor mechanisms.

Section 2.8, lines 297-299, the two sentences here might be combined, as they are too similar to easily distinguish - there was positive or negative correlation of the CXCR2 ligands with MDSC counts, so just state as such.

The first sentence has been removed.

Section 2.17, the authors might provide a transition statement that introduces this comparison of CXCR2 ligand sequences - why is this presented? This can be brief, but as is, this is a bit confusing given all the expression pattern comparisons with tumors above, and then mechanisms regulating ligand gene expression in the subsequent section (the reviewer notes the comment about protein structure vs. mechanisms regulating gene expression in 2.18, which is excellent).

A brief introduction has been added.

Above, an analysis was conducted to correlate the expression of CXCR2 ligands with prognosis, proliferation markers, epithelial-mesenchymal transition (EMT), and the recruitment of cells to the tumor microenvironment. This analysis revealed significant differences among various CXCR2 ligands. Within a given cancer type, many of these ligands exhibit opposing functions and properties in certain analyses. Consequently, in order to determine whether CXCR2 ligands also differ in terms of their sequences to the same extent, a comparison of sequence similarities between CXCR2 ligands was conducted.

Section 3.1, can the authors provide any brief description, or at least examples, of how differences in gene regulators that control expression of the ligands correlate with differences in ligand gene regulation? How does expression of NF-kB or other regulators identified differ, which might explain the differences in ligand gene expression in the studied tumors?

A brief description has been added.

Particularly, the overexpression or loss of expression of a regulatory element (such as a miRNA that suppresses expression or a transcription factor binding to enhancers or silencers) results in changes in the expression of genes regulated by these factors.

Section 3.7 and throughout the Discussion, there are significant redundancies in the text regarding cell type functions and the correlations that are well-described in the Results section, which could be removed or shortened. For example, in lines 907-911, the authors do not need to re-explain M1 vs. M2 functions. With the already lengthy Discussion and abundant Results, some editing of redundancies would help to shorten the length of the manuscript.

The sentence indicated by the reviewer has been removed.

Section 3.12, lines 1139-1155, again the properties of M1 vs. M2 cells are reviewed, which is unnecessary, so suggest removal or at least brevity while discussing the importance of these two cell types.

The paragraph indicated by the reviewer has been removed.
